# Differential Information Distribution: A Bayesian Perspective on Direct Preference Optimization

## Abstract

Direct Preference Optimization (DPO) has been widely used for aligning language models with human preferences in a supervised manner. However, several key questions remain unresolved: the rationale behind its log-ratio reward, how the statistical structure of preference datasets shapes its training dynamics, and how those dynamics impact downstream capabilities. We approach these questions from a Bayesian perspective, interpreting the goal of preference optimization as learning the differential information required to update a reference policy into a target policy. To formalize this view, we introduce the DIFFERENTIAL INFORMATION DISTRIBUTION (DID), defined as the distribution over samples that carry the Bayesian evidence required to update policies. We introduce three complementary insights by viewing preference optimization through the DID. First, we find that DPO's log-ratio reward is uniquely justified when preferences encode the Differential Information needed to update a reference policy into the target policy. Second, we discuss how commonly observed training dynamics in DPO, including changes in log-likelihood and policy exploration, stem from a power-law DID relationship. Finally, we analyze how training dynamics influence downstream performance using the entropy of DID, a principled measure of uncertainty in the learned information. We observe that learning high-entropy DID improves open-ended instruction-following, while low-entropy DID benefits knowledge-intensive QA. Taken together, our results show that DPO's reward design, training dynamics, and downstream capabilities all emerge as natural consequences of learning Differential Information, offering both a principled theoretical foundation and practical guidance for preference-based alignment.[1]

## 1 Introduction

Aligning language models to human preferences is essential for both safety and usefulness (Ouyang et al., 2022; Bai et al., 2022). Among various alignment methods, Direct Preference Optimization (DPO) (Rafailov et al., 2023) has gained popularity for its strong empirical performance, training stability, and computational efficiency (Xiao et al., 2024b; Liu et al., 2025b). Despite its widespread use, several fundamental questions remain: what justifies the log-ratio reward beyond its derivation from a KL-regularized RL objective, what statistical structure in preference datasets underlie DPO's training dynamics, and how these dynamics impact downstream capabilities.

To address these gaps, we propose a Bayesian perspective that interprets the goal of preference optimization as learning the information needed to update a reference policy into a target policy. We call this information "Differential Information" as it represents the difference in the information encoded by the two policies. To formalize this, we introduce the DIFFERENTIAL INFORMATION DISTRIBUTION (DID), defined as the distribution of samples containing the Bayesian evidence[2] for this policy update. The DID can be expressed as the normalized ratio of the two policy distributions (Theorem 2.2). By analyzing the DID in preference optimization, we show that DPO's key

---

[1]Model checkpoints and training/evaluation code will be released upon acceptance.

[2]We consider Bayesian evidence that is conditionally independent of the prior given a sample, ensuring that this evidence reflects a sample's intrinsic content rather than the distribution it was drawn from. This aligns with Shannon information being additive for *independent* events. See Section 2 and Appendix D for details.

components–reward parameterization, training dynamics, and learned capabilities–emerge naturally from this Bayesian perspective.

In Section 3, we present a Bayesian interpretation of the optimality of DPO's log-ratio reward parameterization. We first show how a preference data generation process naturally yields a preference distribution that encodes the Differential Information required to update a reference policy into a target policy. We then prove that the log-ratio reward is the *unique* Bradley-Terry reward that learns this target policy, revealing that DPO's design follows directly from Bayesian principles. We validate these findings through controlled Energy-Based Model experiments, where we simulate a preference dataset encoding Differential Information and demonstrate that only the log-ratio reward converges to the target policy while other objectives (*e.g.*, SimPO (Meng et al., 2024)) fail to do so. We also validate our analysis of the DID relationship in DPO on practical training settings.

In Section 4, we analyze the training dynamics of DPO based on a power-law relationship in the DID imposed by DPO. This DID relationship links the DPO-converged policy to the preference sampling distribution. Because the normalized ratio form of the DID is algebraically tied to the KL-divergence, it explains the consistent changes in the log-likelihood observed during DPO training. The DID power-law further clarifies how policy exploration is jointly influenced by the KL-penalty term and the sharpness of preference data, where sharpness reflects the sparsity (reciprocal of sampling temperature) of distributions over chosen and rejected responses. On both controlled setups and real-world datasets, we confirm these predictions and show that the identified properties of DPO persist under gradient-based stochastic optimization.

In Section 5, we investigate the empirical link between training dynamics and downstream capabilities by analyzing the Shannon entropy of the DID. Recent work finds that preventing log-likelihood displacement (LLD) improves factual accuracy (*e.g.*, MMLU (Hendrycks et al., 2021)) at the cost of open-ended tasks (*e.g.*, Wild-Bench (Lin et al., 2024a)) (Shi et al., 2024; Chen et al., 2024; Xiao et al., 2024a). We hypothesize that this trade-off reflects differences in the DID entropy. To test this, we compare standard DPO against a variant that prevents LLD. We train `Mistral7B-v0.3` (Jiang et al., 2024) and `Qwen3-4B` (Team, 2025) on Magpie-Pro and Magpie-G27 datasets (Xu et al., 2024b). Across all settings, preventing LLD consistently learns a low-entropy DID that improves factual QA, while allowing LLD learns a high-entropy DID that enhances open-ended generation. These results suggest that DID entropy could serve as a useful factor in characterizing whether a model's learned capabilities align with factual precision or open-ended generation.

Taken together, our Bayesian perspective unifies reward design, training dynamics, and learned capabilities of DPO. By explicitly linking preference data to the Differential Information they convey, our approach provides both theoretical grounding and practical guidance for designing and understanding preference-based alignment.

## 2 PRELIMINARIES

Let $\mathcal{Y}$ denote the sample space of all possible sentences. A policy (*i.e.*, language model) $\pi$ defines a probability distribution over $\mathcal{Y}$, and we assume full support, *i.e.*, $\pi(y) > 0$ for all $y \in \mathcal{Y}$. Let $\pi^*$ be the target policy we wish to learn and $\pi_{\mathrm{ref}}$ be a fixed reference policy.

Preferences are ordered pairs $(y_w, y_\ell)$, where $y_w$ is preferred to $y_\ell$, written as $y_w \succ y_\ell$. The Bradley-Terry (BT) model (Bradley & Terry, 1952; Luce et al., 1959) assigns the probability of such a preference under an implicit distribution $p^*$ as

$$p^*(y_w \succ y_\ell) \coloneqq \frac{p^*(y_w)}{p^*(y_w) + p^*(y_\ell)}.$$

If a latent reward $r : \mathcal{Y} \to \mathbb{R}$ induces a Boltzmann distribution $P(Y = y \mid r) \propto \exp(r(y))$, then the BT preference probability can be expressed via the logistic sigmoid $\sigma(x) = 1/(1 + \exp(-x))$ as

$$p(y_w \succ y_\ell \mid r) \coloneqq \sigma\big(r(y_w) - r(y_\ell)\big).$$

**DPO objective and log-ratio reward.** Direct Preference Optimization (DPO) (Rafailov et al., 2023) parameterizes a BT reward derived from the KL-regularized RL objective, the log-ratio be-

tween the learned policy and the reference: $r_{\mathrm{DPO}}(y) := \beta \log(\pi(y)/\pi_{\mathrm{ref}}(y))$, where $\beta > 0$ corresponds to the KL-penalty strength. Under this parameterization, the preference probability becomes

$$p(y_w \succ y_\ell \mid r_{\mathrm{DPO}}) := \sigma\left(\beta \log \frac{\pi(y_w)}{\pi_{\mathrm{ref}}(y_w)} - \beta \log \frac{\pi(y_\ell)}{\pi_{\mathrm{ref}}(y_\ell)}\right).$$

**Preference generation assumption.** Following prior works (Pan et al., 2025; Higuchi & Suzuki, 2025), we assume preferences $(y_w, y_\ell)$ are sampled independently from two distributions $\pi_w$ and $\pi_\ell$ respectively. Given an unordered pair of distinct responses $(y_1, y_2)$, the ground-truth probability that $y_1$ is preferred over $y_2$ is

$$p(y_1 \succ y_2) := P(y_1 \sim \pi_w,\ y_2 \sim \pi_\ell \mid \text{sampled pair is } (y_1, y_2))$$

$$= \frac{\pi_w(y_1)\pi_\ell(y_2)}{\pi_w(y_1)\pi_\ell(y_2) + \pi_w(y_2)\pi_\ell(y_1)}.$$

We denote by $\mathcal{D} = \{(y_w, y_\ell) \mid y_w \sim \pi_w,\ y_\ell \sim \pi_\ell\}$ a preference dataset consisting of such pairs. We assume $\pi_w \neq \pi_\ell$ and $\pi_{\mathrm{ref}} \neq \pi^*$ in the following discussions.

**Preference optimization as distribution matching.** Preference optimization maximizes the empirical likelihood of observed preferences under a BT model parameterized by a reward $r$. Under standard identifiability and coverage assumptions, maximizing the preference likelihood is equivalent to fitting the reward-induced Boltzmann distribution to the implicit preference distribution. We cite the following standard result from Dumoulin et al. (2023) (Proof in Appendix J.1).

**Theorem 2.1** (Preference vs. Distribution Matching (Dumoulin et al., 2023)). *Let $\mathcal{D} = \{(y_w, y_\ell)\}$ be a sufficiently large preference dataset where the sets of $y_w$ and $y_\ell$ cover $\mathcal{Y}$. Then preference optimization on $\mathcal{D}$ is equivalent to fitting the reward-induced distribution $P(Y = y \mid r)$ to the implicit preference distribution $p^*(y)$:*

$$\max_r \mathbb{E}_{(y_w, y_\ell) \sim \mathcal{D}} \left[\log \sigma(r(y_w) - r(y_\ell))\right] \iff \min_r \mathbb{D}_{\mathrm{KL}}\left[p^*(y) \| P(Y = y \mid r)\right].$$

**Differential Information Distribution.** We now introduce the DIFFERENTIAL INFORMATION DISTRIBUTION (DID) to formalize the information that drives the update from a prior belief $\pi_{\mathrm{ref}}$ into a posterior $\pi^*$, which we later use to interpret preference optimization. The term *differential* highlights that the difference in the information contained in $\pi^*$ and $\pi_{\mathrm{ref}}$ is precisely the additional Bayesian evidence required to update $\pi_{\mathrm{ref}}$ into $\pi^*$.

Suppose we start from a prior $\pi_{\mathrm{ref}}$ over sentences and then observe new Bayesian evidence $X$ that updates $\pi_{\mathrm{ref}}$ to $\pi^*$. We define the DID as the distribution over sentences that carry this incremental evidence. A central postulate is that $X$ is conditionally independent of the prior distribution given some sentence $y$. In other words, the probability that $y$ exhibits $X$ does not depend on whether $y$ was sampled from $\pi_{\mathrm{ref}}$. This ensures that the information $X$ attributed to each sentence reflects its intrinsic features. This also parallels how the difference in Shannon information, $-\log P_A - (-\log P_B) = -\log \frac{P_B}{P_A}$, equals the Shannon information of an *independent* event $X$ that updates $P_A$ into $P_B$ via Bayes' rule: $P_{A,X} = P_A P_X = P_B$. See Appendix D for details.

**Definition 2.2** (Differential Information Distribution). Let $\pi^*$ and $\pi_{\mathrm{ref}}$ be two probability distributions over $\mathcal{Y}$ with full support. Let $X$ be an event that satisfies the following:

$$\begin{cases} P(X \mid Y = y, \pi_{\mathrm{ref}}) = P(X \mid Y = y) & \text{(Conditional Independence)} \\ \pi^*(y) = P(Y = y \mid \pi_{\mathrm{ref}}, X) & \text{(Bayesian Update)} \end{cases}$$

Then, $P(Y = y \mid X)$ is defined as the *Differential Information Distribution* (DID) from $\pi_{\mathrm{ref}}$ to $\pi^*$.

Intuitively, if our initial belief regarding which sentence $y$ is the correct sentence follows the distribution $\pi_{\mathrm{ref}}(y)$, and then learn that "all sentences satisfying $X$ are correct", our updated belief becomes $\pi^*(y)$. The following theorem shows that the DID can be computed directly from the normalized likelihood ratio.

**Theorem 2.3** (Likelihood Ratio Representation of Differential Information Distribution). *For policies $\pi^*, \pi_{\mathrm{ref}}$ over $\mathcal{Y}$ with full support, the Differential Information Distribution (DID) from $\pi_{\mathrm{ref}}$ to $\pi^*$ is equivalent to the normalized ratio distribution:*

$$P(Y = y \mid X) = \frac{\pi^*(y)/\pi_{\mathrm{ref}}(y)}{Z} := q_{\pi^*/\pi_{\mathrm{ref}}}(y),$$

*where $Z = \sum_{y' \in \mathcal{Y}} \frac{\pi^*(y')}{\pi_{\mathrm{ref}}(y')}$ is the partition function.*

(Proof in Appendix D.3) Therefore, the normalized ratio distribution $q_{\pi^*/\pi_{\text{ref}}}$ can be understood as the Differential Information Distribution responsible for the update from $\pi_{\text{ref}}$ to $\pi^*$.

# 3 OPTIMALITY OF DPO'S LOG-RATIO REWARD

In this section, we begin our analysis of preference optimization through the lens of DIFFEREN-TIAL INFORMATION. We reframe the goal of preference optimization as learning the Differential Information that updates a reference policy $\pi_{\text{ref}}$ into a target policy $\pi^*$. We first show how a preference dataset naturally encodes that Differential Information (Theorem 3.1), then prove that DPO's log-ratio reward is the unique Bradley-Terry reward that learns $\pi^*$ from such data (Theorem 3.2). Finally, we validate these claims in a controlled Energy-Based Model experiment and also using real-world datasets (Section 3.3).

## 3.1 HOW PREFERENCES ENCODE DIFFERENTIAL INFORMATION

Since policy updates in preference optimization are driven by the preference distribution $p^*$ of a dataset, it is essential to characterize precisely *when* $p^*$ contains the Differential Information required to transform $\pi_{\text{ref}}$ into $\pi^*$. We find this condition is met when the Differential Information Distributions (DIDs) of the underlying policies are related by a power-law.

**Theorem 3.1** (Preferences Encoding Differential Information). *Consider a preference dataset* $\mathcal{D} = \{(y_w, y_\ell) \mid y_w \sim \pi_w, y_\ell \sim \pi_\ell\}$. *Let $\pi^*$ be the target policy. If the Differential Information Distribution between policies match up to an exponent $\beta > 0$:*

$$q_{\pi_w/\pi_\ell}(y) \propto q_{\pi^*/\pi_{\text{ref}}}(y)^\beta, \quad \forall y \in \mathcal{Y},$$

*then the preference probability $p^*(y_w \succ y_\ell)$ can be expressed as preferences induced by the DID:*

$$p^*(y_w \succ y_\ell) = \sigma\left(\beta \log q_{\pi^*/\pi_{\text{ref}}}(y_w) - \beta \log q_{\pi^*/\pi_{\text{ref}}}(y_\ell)\right).$$

(Proof in Appendix J.2) The condition requires that the Differential Information that updates the rejected response distribution $\pi_\ell$ into the chosen one $\pi_w$ should align with the DID from $\pi_{\text{ref}}$ to $\pi^*$, up to an exponent $\beta > 0$. When this holds, the dataset's preference distribution carries the Bayesian evidence needed to update $\pi_{\text{ref}}$ into $\pi^*$.

## 3.2 BRADLEY-TERRY REWARD FOR LEARNING DIFFERENTIAL INFORMATION

As we have characterized when a preference dataset encodes the Differential Information necessary for learning $\pi^*$, we now ask which functional form of the reward parameterization $r(y)$ recovers $\pi^*$. We find that the log-ratio form used by DPO is the unique functional form (up to a constant) that makes $\pi^*$ the global optimizer of the training objective.

**Theorem 3.2** (Optimal Reward for Learning Differential Information). *Let $\mathcal{D}$ be a preference dataset satisfying Theorem 3.1, encoding the Differential Information required to learn the target policy $\pi^*$. Then, for some constant $C$, we have*

$$\pi^* = \arg\max_\pi \mathbb{E}_{(y_w, y_\ell) \sim \mathcal{D}} \left[\log \sigma(r(y_w) - r(y_\ell))\right] \iff r(y) = \beta \log \frac{\pi(y)}{\pi_{\text{ref}}(y)} + C.$$

(Proof in Appendix J.3) This justifies DPO's log-ratio structure: if preference captures the Differential Information needed to improve $\pi_{\text{ref}}$ toward $\pi^*$, then using the log-ratio reward is not merely a heuristic choice, but the only functional form that ensures preference optimization recovers $\pi^*$. Our derivation recovers the result of Rafailov et al. (2023), originally motivated by the KL-regularized RL objective. This highlights the Bayesian structure of DPO in learning Differential Information.

The key results of Theorems 3.1 and 3.2 can be summarized into the following relationship.

**Corollary 3.2.1** (DID Power-Law of DPO). *Consider a preference dataset $\mathcal{D} = \{(y_w, y_\ell) \mid y_w \sim \pi_w, y_\ell \sim \pi_\ell\}$ and a policy $\pi^*$ obtained as a stationary point of preference optimization using the log-ratio reward $r = \beta \log(\pi/\pi_{\text{ref}})$ on $\mathcal{D}$. Then, a power-law relationship between the DID of policies must hold:*

$$q_{\pi_w/\pi_\ell}(y) \propto q_{\pi^*/\pi_{\text{ref}}}(y)^\beta, \quad \forall y \in \mathcal{Y}.$$

(Proof in Appendix J.4) Corollary 3.2.1 can be read two ways: either datasets that satisfy the DID power-law lead DPO to recover $\pi^*$; conversely, if DPO has converged to some $\pi^*$ then the dataset's sampling distributions must satisfy this power-law relation.[3]

### 3.3 EXPERIMENTS

We first validate our theoretical findings in a controlled setup using Energy-Based Models.

**Setup.** We define policies $\pi_\theta(i) = \exp(\theta_i) / \sum_j \exp(\theta_j)$ for class $i \in \{1, \ldots, K\}$ and $\theta \in \mathbb{R}^K$. The logits of the reference policy $\pi_{\text{ref}}$ are sampled from a normal distribution: $\theta_{\text{ref}} \sim \mathcal{N}(0, I)$. Next, to construct the target policy $\pi^*$, we set the target logits $\theta^* = \theta_{\text{ref}} / \tau$ with $0 < \tau < 1$ for reinforcing and $\tau > 1$ for smoothing. The logits of $\pi_\ell$ are set as $\theta_\ell = 2\theta_{\text{ref}} - \theta^*$, which aligns the DID between policies: $q_{\pi_{\text{ref}}/\pi_\ell} = q_{\pi^*/\pi_{\text{ref}}}$. Finally, preference pairs $(y_w, y_\ell)$ are constructed by sampling $y_w \sim \pi_{\text{ref}}$ and $y_\ell \sim \pi_\ell$, and labeled as $y_w \succ y_\ell$ (Hyper-parameters in Appendix M.1).

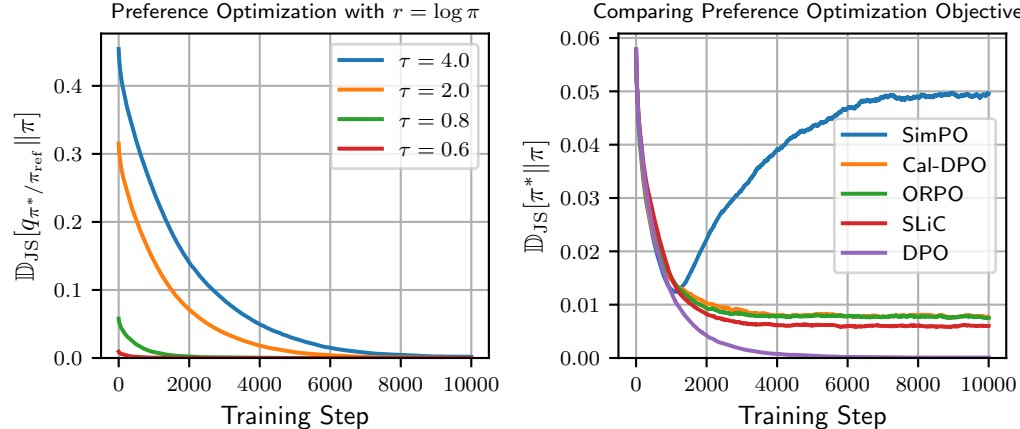

Figure 1: *Left*: Optimization using $r = \log \pi$ on preference data satisfying the DID power-law. The Jensen-Shannon Divergence $\mathbb{D}_{\text{JS}}[q_{\pi^*/\pi_{\text{ref}}} \| \pi]$ converges to 0, confirming Theorem 3.1 that the preference encodes Differential Information. *Right*: Comparison of $\mathbb{D}_{\text{JS}}[\pi^* \| \pi]$ using different objectives on the same data with $\tau = 4$. Standard DPO ($r = \log(\pi/\pi_{\text{ref}})$, purple) uniquely converges to $\pi^*$, consistent with Theorem 3.2.

**Do preferences encode Differential Information?** Theorem 3.1 predicts that the preference distribution encodes the Differential Information required to learn the target policy: $p^* = q_{\pi^*/\pi_{\text{ref}}}$. According to Theorem 2.1, a policy optimized using $r = \log \pi$ converges to the underlying preference distribution $p^*$ (Dumoulin et al., 2023; Xu et al., 2024a; Liu et al., 2024b). Therefore, we optimize a policy $\pi$ with $r = \log \pi$ and measure the Jensen-Shannon (JS) divergence between $q_{\pi^*/\pi_{\text{ref}}}$ and $\pi$.

Figure 1 shows that the JS divergence consistently converges to zero, meaning that the policy trained to directly fit $p^*$ converges to the DID $q_{\pi^*/\pi_{\text{ref}}}$. This confirms Theorem 3.1 in that **sampling chosen and rejected samples each from a distribution satisfying the DID power-law yields a preference distribution encoding the Differential Information required to learn the target policy.**

**Is the log-ratio reward optimal?** To test Theorem 3.2, we compare optimization with the log-ratio reward $r = \log(\pi/\pi_{\text{ref}})$ against several alternative objectives: SLiC (Zhao et al., 2023b), ORPO (Hong et al., 2024), SimPO (Meng et al., 2024), and Cal-DPO (Xiao et al., 2024a). All methods are trained on the same synthetic dataset and we compare $\mathbb{D}_{\text{JS}}[\pi^* \| \pi]$ for each method.

Figure 9 shows that the DPO log-ratio objective is the *only* method that consistently minimizes $\mathbb{D}_{\text{JS}}[\pi^* \| \pi]$ across various $\tau$ values, converging to $\pi^*$. This empirical result supports Theorem 3.2,

---

[3]Corollary 3.2.1 directly yields a closed-form expression of the optimal DPO dataset (Appendix F).

highlighting that **when preferences encode Differential Information, the log-ratio reward succeeds in recovering the target policy while other objectives fail to do so.**

**Does the DID power-law hold in real-world DPO training?** Corollary 3.2.1 predicts that a DPO-converged policy $\pi^*$ should satisfy the DID power-law: $q_{\pi_w/\pi_\ell}(y) \propto q_{\pi^*/\pi_{\mathrm{ref}}}(y)^\beta$. In particular, the DID power-law implies that when the reference policy is fine-tuned on the chosen responses ($\pi_{\mathrm{ref}} = \pi_w$), the average reward-margin of the converged $\pi^*$ should be invariant to the choice of $\beta > 0$ used for DPO training (proof in Appendix G):

$$\mathbb{E}_{y_w \sim \pi_{\mathrm{ref}}}[\beta \log \frac{\pi^*(y_w)}{\pi_{\mathrm{ref}}(y_w)}] - \mathbb{E}_{y_\ell \sim \pi_\ell}[\beta \log \frac{\pi^*(y_\ell)}{\pi_{\mathrm{ref}}(y_\ell)}] = \mathbb{D}_{\mathrm{KL}}\left[\pi_{\mathrm{ref}} \| \pi_\ell\right] + \mathbb{D}_{\mathrm{KL}}\left[\pi_\ell \| \pi_{\mathrm{ref}}\right].$$

To test this on real data, we fine-tuned `Mistral7B-v0.3` and `Qwen3-4B` on the Magpie-Pro and Magpie-G27 instruction-following datasets (Appendix M.2). The reference policy was first fine-tuned for one epoch on the chosen responses and then trained with DPO for one epoch (1,530 steps) using $\beta \in \{0.05, 0.1, 0.2\}$. Final reward- and log-margins were measured over the last 50 steps.

Figure 10 shows that the log-margins of the DPO-converged policy differ substantially across $\beta$, approximately scaling inversely with $\beta$ as predicted. In contrast, Figure 11 shows that the reward-margins remain within a narrow, stable range. **These results provide strong empirical support that the DID power-law of Corollary 3.2.1 holds robustly in practical DPO training setups.** To our knowledge, prior gradient-based analyses (Pal et al., 2024; Feng et al., 2024; Razin et al., 2024) cannot predict such log/reward-margin trends at convergence, which demonstrates the explanatory strength of our DID-based analysis.

## 4 TRAINING DYNAMICS OF DPO

The power-law structure of the Differential Information Distribution (DID) identified in the previous section (Corollary 3.2.1) determines how DPO updates policies during training. This provides a unified lens through which we can understand the training dynamics of DPO. We discuss how the power-law DID relationship proves general guarantees on the log-likelihood change in DPO (Theorem 4.1). Next, we analyze which factors impact policy exploration during DPO training, based on the power-law DID relationship (Theorem 4.2).

### 4.1 LOG-LIKELIHOOD CHANGE

Corollary 3.2.1 explains the characteristic log-likelihood shifts observed during DPO training. The DID power-law, which DPO must satisfy at convergence, clarifies how the preference sampling distributions and the converged policy are linked together. Combined with Jensen's and Gibbs' inequalities, this relationship predicts the asymmetric shifts in the log-likelihoods of chosen and rejected responses. Unlike prior analyses (Feng et al., 2024; Razin et al., 2024; Cho et al., 2025), our statements derived from this principle place no restrictions on step sizes, gradients, or parameterization methods. We further extend beyond the in-distribution regime (*i.e.*, samples $\pi_{\mathrm{ref}}$ was trained on) considered in some previous work (Rafailov et al., 2024).

**Theorem 4.1** (Log-Likelihood Change of DPO). *Consider a preference dataset $\mathcal{D} = \{(y_w, y_\ell) \mid y_w \sim \pi_{\mathrm{ref}}, y_\ell \sim \pi_\ell\}$, and $\pi^*$ obtained by preference optimization on $\mathcal{D}$ using the log-ratio reward $r = \beta \log \pi/\pi_{\mathrm{ref}}$. Then, for any $\beta > 0$, $\pi^*$ must decrease the average log-likelihood of $y_\ell$:*

$$\mathbb{E}_{y_\ell \sim \pi_\ell}\left[\log \pi^*(y_\ell)\right] < \mathbb{E}_{y_\ell \sim \pi_\ell}\left[\log \pi_{\mathrm{ref}}(y_\ell)\right].$$

*Conversely, if $\pi_{\mathrm{ref}}$ was fine-tuned on $y_\ell$ (i.e., $\pi_{\mathrm{ref}} = \pi_\ell$), then, for any $\beta \geq 1$, $\pi^*$ must increase the average log-likelihood of $y_w$:*

$$\mathbb{E}_{y_w \sim \pi_w}\left[\log \pi^*(y_w)\right] > \mathbb{E}_{y_w \sim \pi_w}\left[\log \pi_{\mathrm{ref}}(y_w)\right].$$

See Appendix J.5 for proof. The theorem captures a basic asymmetric effect of DPO: when $y_w$ is sampled from $\pi_{\mathrm{ref}}$, the converged policy $\pi^*$ must decrease $\log \pi(y_\ell)$, and when $y_\ell \sim \pi_{\mathrm{ref}}$, $\pi^*$ must increase $\log \pi(y_w)$ for $\beta \geq 1$. The theorem therefore connects the power-law DID to concrete, observable training dynamics and clarifies how and why likelihoods change during DPO training.[4]

---

[4]See Appendix G for a discussion on how the log-margin $\log \pi(y_w) - \log \pi(y_\ell)$ evolves during DPO training, and how it implies an information-theoretic triangle inequality that must hold at convergence.

## 4.2 POLICY EXPLORATION

The characteristics of policy exploration in DPO can also be explained using the power-law DID relationship. In particular, this DID structure allows DPO to adaptively adjust its KL-divergence from the reference policy $\pi_{\text{ref}}$ based on the sharpness of the preference data. Intuitively, when preference labels are weak (*i.e.*, $p^*(y_1 \succ y_2)$ being close to 0.5), the power-law DID relationship constrains the DPO-converged policy $\pi^*$ to remain close to $\pi_{\text{ref}}$. Conversely, when preference probabilities are stronger (*i.e.*, $p^*(y_1 \succ y_2)$ being close to 0 or 1), the same DID relationship drives the converged policy farther away from $\pi_{\text{ref}}$, under the same KL-penalty $\beta$. This trade-off is formalized below (proof in Appendix J.6).

**Theorem 4.2** (Adaptive Policy Exploration of DPO). *Let $\mathcal{D} = \{(y_w, y_\ell) \mid y_w \sim \pi_{\text{ref}}, \ y_\ell \sim \pi_\ell\}$ be a preference dataset with an implicit Bradley-Terry preference distribution $p^*_{\mathcal{D}}$. Consider another dataset $\mathcal{D}' = \{(y_w, y_\ell)\}$ whose implicit Bradley-Terry distribution $p^*_{\mathcal{D}'}$ is a "sharpened" version of $p^*_{\mathcal{D}}$, in the sense that there exists $\alpha > 1$ such that for all pairs $(y_w, y_\ell) \in \mathcal{Y} \times \mathcal{Y}$,*

$$p^*_{\mathcal{D}'}(y_w \succ y_\ell) = \frac{\left(p^*_{\mathcal{D}}(y_w)\right)^\alpha}{\left(p^*_{\mathcal{D}}(y_w)\right)^\alpha + \left(p^*_{\mathcal{D}}(y_\ell)\right)^\alpha} = \sigma\left(\alpha \log p^*_{\mathcal{D}}(y_w) - \alpha \log p^*_{\mathcal{D}}(y_\ell)\right).$$

*For the same reference policy $\pi_{\text{ref}}$ and any $\beta > 0$, let $\pi^*_{\mathcal{D}}$ and $\pi^*_{\mathcal{D}'}$ denote the policies obtained by preference optimization on $\mathcal{D}$ and $\mathcal{D}'$, respectively, using the log-ratio reward $r = \beta \log \pi/\pi_{\text{ref}}$. Then the strengthened dataset $\mathcal{D}'$ induces a strictly larger divergence from the reference:*

$$\mathbb{D}_{\text{KL}}\left[\pi_{\text{ref}}\|\pi^*_{\mathcal{D}'}\right] \ > \ \mathbb{D}_{\text{KL}}\left[\pi_{\text{ref}}\|\pi^*_{\mathcal{D}}\right].$$

**Remark 1.** *By Theorem 3.1, decreasing the data sampling temperature by a factor of $\alpha > 1$, i.e., drawing $y_w$ and $y_\ell$ from $\pi_w(y)^\alpha$ and $\pi_\ell(y)^\alpha$, amplifies preference strength by the same factor $\alpha$.*

**Remark 2.** *To recover $\pi^*_{\mathcal{D}}$ by optimizing a stronger dataset $\mathcal{D}'$ using the log-ratio reward $r' = \beta' \log \frac{\pi}{\pi_{\text{ref}}}$, one must increase the KL-penalty strength such that $\beta' = \beta \cdot \alpha$ for $\alpha > 1$.*

Thus, the effective KL-budget of DPO depends not only on $\beta$, but also on the strength of the preference data. Datasets with weak or noisy preference labels (*i.e.*, $p^*(y_1 \succ y_2) \approx 0.5$) constrain DPO to remain near $\pi_{\text{ref}}$, where large deviations cannot be justified by the evidence. In contrast, stronger preference labels act as a divisor on the log-ratio reward (Remark 2), enabling larger departures from $\pi_{\text{ref}}$. Therefore, the DID power-law allows DPO to balance conservatism and exploration, by linking policy deviation directly to the strength or quality of preference data.[5]

## 4.3 EXPERIMENTS

We first validate Theorems 4.1 and 4.2 using the Energy-Based Model (EBM) experiment from Section 3.3. All policies (reference, rejected, chosen) are parameterized by independent logits drawn from a normal distribution. The dataset is constructed from preference pairs sampled as in Section 3.3, and we train policies using the DPO objective with varying $\beta$ values.

**Verifying log-likelihood change.** To test Theorem 4.1, we consider two cases: (1) chosen responses sampled from the reference ($y_w \sim \pi_{\text{ref}}$), and (2) rejected responses sampled from the reference ($y_\ell \sim \pi_{\text{ref}}$). We then track the change in average log-likelihoods under DPO training.

Figure 2 confirms the predictions of Theorem 4.1. When $y_w \sim \pi_{\text{ref}}$, the converged policy $\pi^*$ decreases $\mathbb{E}[\log \pi(y_\ell)]$ relative to $\pi_{\text{ref}}$. Conversely, when $y_\ell \sim \pi_{\text{ref}}$, DPO increases $\mathbb{E}[\log \pi(y_w)]$ for $\beta \geq 1$. The same trend can also be observed in real-world settings using `Qwen3-4B` (Figure 3). We find that the direction of log-likelihood change of DPO ($\beta = 1.0$) is consistent with Theorem 4.1 on three datasets (Magpie-Pro, Magpie-G27, Ultra-Feedback; 50k samples each).

These results show that **the consistent log-likelihood shifts observed in DPO can be rigorously explained and precisely predicted based on the power-law DID relationship.**

---

[5]This is consistent with empirical trends from Wu et al. (2024) where easily discriminated preference pairs require higher $\beta$ values to prevent overfitting, while noisy pairs require lower $\beta$ values for aggressive updates.

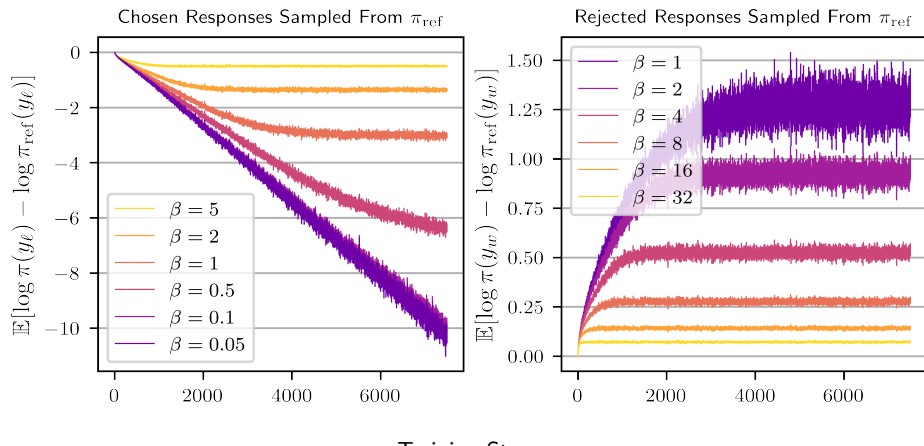

Figure 2: Log-likelihood change during DPO training. When chosen responses $y_w$ are sampled from $\pi_{\text{ref}}$, the log-likelihood of rejected responses $y_\ell$ decreases relative to $\pi_{\text{ref}}$ (left plot). When rejected samples $y_\ell$ are sampled from $\pi_{\text{ref}}$, the log-likelihood of chosen responses increases for $\beta \geq 1$ (right plot). This confirms the predicted change in log-likelihood of DPO (Theorem 4.1).

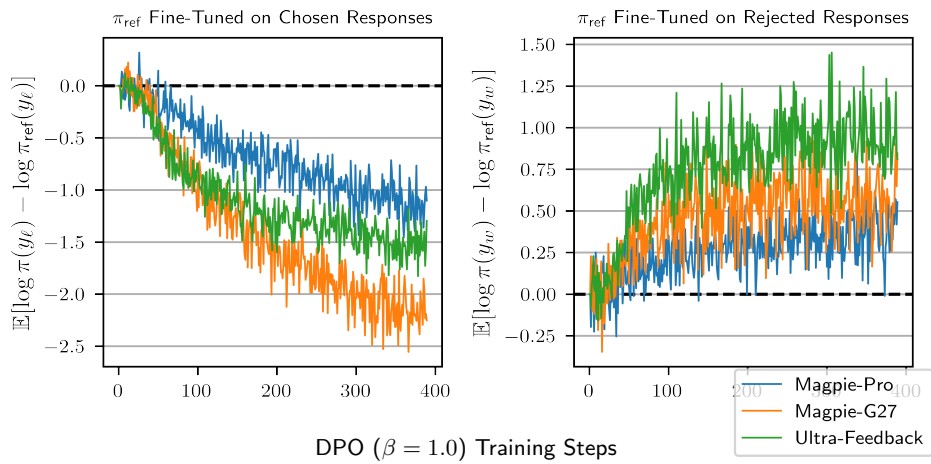

Figure 3: Log-likelihood change of real-world DPO training. On three datasets (Magpie-Pro, Magpie-G27, Ultra-Feedback; 50k samples each) and using `Qwen3-4B`, the reference policy $\pi_{\text{ref}}$ is first fine-tuned for one epoch on either chosen or rejected responses, followed by one epoch of DPO ($\beta = 1.0$). Consistent with Theorem 4.1, DPO decreases the log-probability of rejected responses when $\pi_{\text{ref}}$ is fine-tuned on chosen responses (left), and increases the log-probability of chosen responses when $\pi_{\text{ref}}$ is fine-tuned on rejected responses (right).

**Verifying policy exploration.** To test Theorem 4.2, we generate preference datasets with varying sharpness. Specifically, we compare a fixed dataset $\mathcal{D}$ with a sharpened version $\mathcal{D}'$ that halves the sampling temperature of preference pairs (Remark 1), increasing the preference strength to $\alpha = 2$. We train policies on both datasets using the DPO objective with various $\beta$ values, and track the KL-divergence $\mathbb{D}_{\text{KL}}\left[\pi_{\text{ref}}\|\pi\right]$ throughout the training process.

Figure 4 shows that a stronger preference signal ($\mathcal{D}', \alpha = 2$) consistently yields larger divergence from $\pi_{\text{ref}}$ than a weaker one ($\mathcal{D}, \alpha = 1$), for the same $\beta$. Moreover, increasing the KL-penalty to $\beta' = 2\beta$ for the sharpened dataset $\mathcal{D}'$ results in a converged policy that matches the KL-divergence of the original dataset $\mathcal{D}$, consistent with Remark 2. This confirms that **policy exploration in DPO is jointly governed by the KL-penalty weight $\beta$ and the implicit strength of preference data.**

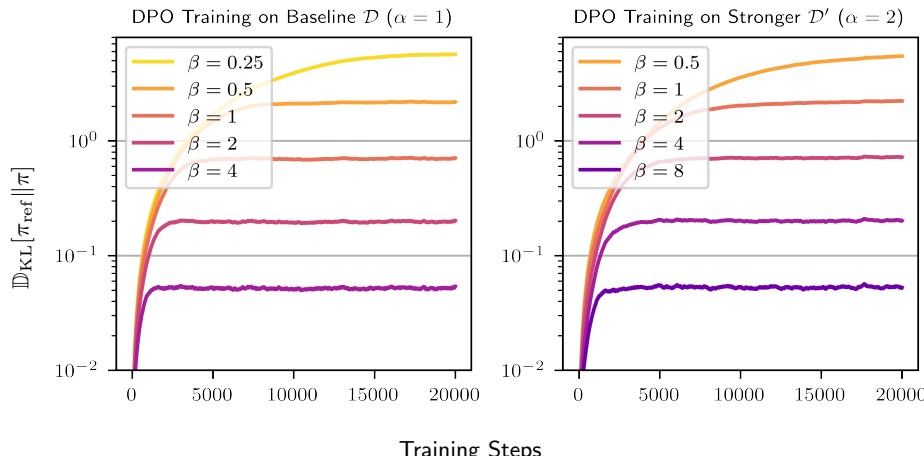

Figure 4: Policy exploration of DPO. Compared to the original dataset $\mathcal{D}$, halving the sampling temperature to form $\mathcal{D}'$ strengthens preferences ($\alpha = 2$) and increases the KL-divergence from $\pi_{\text{ref}}$ under the same KL-penalty $\beta$, consistent with Theorem 4.2. Increasing the KL-penalty to $2\beta$ when training on $\mathcal{D}'$ restores the divergence to the level obtained with $\mathcal{D}$ using $\beta$, in line with Remark 2.

## 5 TRAINING DYNAMICS AND LEARNED CAPABILITIES

In this section, we examine how log-likelihood dynamics shape downstream performance by analyzing the properties of the learned Differential Information. We first study the empirical link between the *log-likelihood displacement* (LLD) phenomenon and downstream capabilities (Section 5.1). We then show how the Shannon entropy of the Differential Information Distribution can help explain the observed trade-off, where different training dynamics lead to distinct capabilities (Section 5.2).

### 5.1 A CASE STUDY ON LOG-LIKELIHOOD DISPLACEMENT

*Log-likelihood displacement* (LLD) refers to the phenomenon where the log-likelihood of chosen response decreases during DPO training, even as alignment improves (Rafailov et al., 2024; Razin et al., 2024; Shi et al., 2024). Preventing LLD has been shown to improve performance on benchmarks such as MMLU (Hendrycks et al., 2021), which requires verifiable, ground-truth answers (Shi et al., 2024; Chen et al., 2024; Xiao et al., 2024a). While prior work investigates the cause of LLD through sample similarity or gradient dynamics (Pal et al., 2024; Razin et al., 2024; Feng et al., 2024), it doesn't fully explain why preventing LLD results in learning different capabilities.

To investigate this gap, we conduct a case study on how LLD affects downstream performance. We compare standard DPO with an another method utilizing projected gradient descent to prevent LLD while still optimizing the DPO objective, which we term **DPO-PG** (Appendix K). All runs start from the same $\pi_{\text{ref}}$ fine-tuned on chosen responses. We train `Mistral7B-v0.3` (Jiang et al., 2024) and `Qwen3-4B` (Team, 2025) on two instruction-following datasets (Magpie-Pro and Magpie-G27), and evaluate on open-ended instruction-following (Arena-Hard (Li* et al., 2024), Wild-Bench (Lin et al., 2024a)) and a suite of eight knowledge-intensive QA tasks (details in Appendix M.2).

As shown in Table 1, across both model architectures, preventing LLD (DPO-PG) consistently yields the strongest performance on knowledge-intensive QA, at the expense of open-ended instruction-following. Standard DPO shows the opposite pattern, excelling on open-ended tasks but underperforming on factual QA. The same trend appears on Magpie-G27 (Table 4, Figure 16). This raises the question: *why is LLD associated with a trade-off between factual QA and open-ended tasks?*

### 5.2 CONNECTING POLICY DYNAMICS WITH LEARNED CAPABILITIES

We hypothesize that this trade-off reflects the properties of the information learned from training. To measure this, we use the Shannon entropy of the DID which quantifies the uncertainty or con-

Table 1: Impact of log-likelihood displacement (LLD) on downstream capabilities using the Magpie-Pro dataset. For open-ended instruction-following, we report Arena-Hard-v0.1 win-rate (AH, [%]) and Wild-Bench-v2 ELO score (WB). For knowledge-intensive QA, we report mean reciprocal rank across 8 QA benchmarks (QA). DID entropy (Ent., [nats]) is estimated via importance sampling (Appendix D.5). Compared to standard DPO, preventing LLD (DPO-PG) learns a low-entropy DID, which enhances factual accuracy but reduces performance on open-ended tasks.

| | | Mistral7B-v0.3 | | | | Qwen3-4B | | | |
|--------|--------|--------|---------|-----------|---------|--------|---------|-----------|---------|
| Method | $\beta$ | Ent. | AH ($\uparrow$) | WB ($\uparrow$) | QA ($\uparrow$) | Ent. | AH ($\uparrow$) | WB ($\uparrow$) | QA ($\uparrow$) |
| DPO | 1.0 | 1123.2 | 19.1 | 1141.5 | 0.53 | 9158.4 | 30.7 | 1134.4 | 0.49 |
| | 0.2 | 1303.4 | 18.5 | 1145.3 | 0.26 | 4765.9 | 28.1 | 1146.2 | 0.34 |
| | 0.1 | 1253.9 | **23.4** | **1146.6** | 0.28 | 7663.3 | 27.5 | 1148.2 | 0.27 |
| | 0.05 | 970.2 | 22.4 | 1145.3 | 0.30 | 6801.2 | **43.7** | **1164.5** | 0.24 |
| DPO-PG | | 495.1 | 19.6 | 1129.9 | **0.92** | 388.2 | 37.4 | 1148.5 | **0.94** |

centration of Differential Information that drives policy updates. The DID entropy is defined as

$$H(q_{\pi^*/\pi_{\text{ref}}}) = -\sum_{y \in \mathcal{Y}} q_{\pi^*/\pi_{\text{ref}}}(y) \log q_{\pi^*/\pi_{\text{ref}}}(y).$$

Intuitively, low DID entropy indicates that Differential Information is concentrated on a narrow subset of samples, while high entropy suggests it is distributed more broadly. Appendices D.4 and D.5 detail how DID entropy reflects the properties of Differential Information and how it can be estimated using importance sampling.

As shown in Table 1, the DID entropy (column "Ent.") is observed to be significantly lower when preventing LLD (DPO-PG) compared to standard DPO. We hypothesize that LLD reflects changes in the output distribution that increase the entropy of the learned DID. Since chosen responses $y_w$ typically lie in high-probability regions of the reference policy, decreasing $\log \pi(y_w)$ would smooth these probability peaks, yielding a more high-entropy DID. Conversely, increasing $\log \pi(y_w)$ sharpens these peaks, concentrating probability mass and reducing DID entropy (Appendix H).

Comparing the DID entropy and downstream performance in Table 1, we observe that factual QA performance is associated with low-entropy DID, while open-ended task performance is associated with high-entropy DID. This aligns with intuition: factual queries (*e.g.*, "What is the capital of France?") admit only a narrow set of correct answers (Lee et al., 2023; Xiang et al., 2025), concentrating Bayesian evidence on a small subset of $\mathcal{Y}$. In contrast, open-ended prompts (*e.g.*, "Write a story about a dragon.") admit a wide variety of valid responses (Li et al., 2025; Gu et al., 2024), dispersing Bayesian evidence more broadly across $\mathcal{Y}$. These results suggest that learning low-entropy DID enhances factual precision, while learning high-entropy DID improves open-ended tasks.

In summary, we observe that preventing LLD induces the model to learn a low-entropy DID, which improves accuracy on factual QA tasks. In contrast, allowing LLD results in learning a high-entropy DID that enhances open-ended tasks. These observations suggest that log-likelihood dynamics reflect the type of information learned during alignment.

## 6 CONCLUSION

We introduced a Bayesian perspective on Direct Preference Optimization (DPO) through the lens of DIFFERENTIAL INFORMATION DISTRIBUTION (DID). We showed that DPO's log-ratio reward is the unique Bradley-Terry reward that learns the target policy when preferences encode Differential Information. We further demonstrated that DPO's characteristic training dynamics (log-likelihood shifts and adaptive policy exploration) stem from a power-law DID relationship. We finally introduced DID entropy as a principled measure of uncertainty in the learned information, clarifying the trade-off between log-likelihood displacement and downstream performance: high-entropy DID smooths the output distribution and aids open-ended instruction-following, while low-entropy DID concentrates probability mass and benefits knowledge-intensive QA. Together, our findings provide both a principled theoretical foundation and practical guidance for preference-based alignment.

## 7 REPRODUCIBILITY STATEMENT

To ensure the reproducibility of our work, we provide comprehensive details on our theoretical and empirical findings. For our theoretical results, detailed proofs for all theorems and corollaries are available in Appendix J. For our empirical validation, Appendix M contains a full description of our experimental setups. Specifically, Appendix M.1 details the setup and hyper-parameters for our controlled experiments using Energy-Based Models. Appendix M.2 describes the details for preparing the Magpie-G27 dataset, the training configurations for both DPO and our DPO-PG method, and the evaluation protocols for the real-data experiments in Section 5.2. In the supplementary material, we include the training code for the EBM experiments, raw evaluation results for the real-data experiments, and a reference Pytorch implementation of the DPO-PG method (Appendix K). We plan to release all model checkpoints and the complete code for training and evaluation upon acceptance to ensure direct replication of our findings.

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

CONTENTS OF APPENDIX

## A   LLM USAGE DISCLOSURE

In accordance with the ICLR 2026 policy,[6] we disclose that large language models were used for minor writing assistance and polishing. All research ideations, technical derivations, and experiments were planned and carried out by the human authors.

## B   LIMITATIONS

While our perspective offers novel insights, we acknowledge limitations for future work. First, Theorem 2.1, established from prior work (Dumoulin et al., 2023), assumes sufficient data coverage and train-test generalization. Second, the connection between DID entropy and policy dynamics (Claim H.1) is qualitative and based on information-theoretic intuition (Appendix H); despite experimental support (Section 5.2), a formal treatment would strengthen this aspect of our work.

**Assumptions on the Reference Policy.**   Our general Bayesian formulation of Direct Preference Optimization (DPO) in Section 3 applies to any reference policy with full-support. However, our analytical derivation of training dynamics in Section 4 assumes the setting where the reference policy $\pi_{\text{ref}}$ matches either the chosen or rejected response distribution (*i.e.*, $\pi_{\text{ref}} = \pi_w$ or $\pi_{\text{ref}} = \pi_\ell$). This assumption reflects standard practice: fine-tuning $\pi_{\text{ref}}$ on chosen/rejected responses (Rafailov et al., 2023; 2024) or using on-policy generations for preference data (Meng et al., 2024; Guo et al., 2024b). Importantly, our theoretical predictions match the empirical patterns observed in real-life DPO training (Figures 3, 10, 11), confirming that our analysis accurately captures practical behavior. While scenarios may arise where $\pi_{\text{ref}}$ differs substantially from the underlying preference distributions, extending our analysis to such cases and studying their convergence properties is a promising direction for future work. For such alternative choices of reference policy or different preference-sampling assumptions, the power-law DID result in Corollary 3.2.1 still holds, as long as closed-form expressions for the reference policy or the preference-sample distribution under $\pi_w$ and $\pi_\ell$ can be obtained. This allows the mathematical analysis in Section 4 to be applied in a similar manner.

**Real-World Optimality of DPO.**   It is worth noting that Theorem 3.2 should not be interpreted as claiming universal optimality of DPO on real data; rather, it highlights DPO's unique optimality in learning Differential Information, which may help explain its widespread adoption across domains. While verifying DPO's optimality on real-world data is infeasible without access to the global optimal policy, **we do not claim that DPO is optimal for arbitrary datasets**. In our framework, $\pi^*$ represents a target policy we aim to learn, which is not necessarily the globally optimal policy. Furthermore, Theorem 3.2 establishes DPO's optimality within a Bayesian framework where preferences encode the *Differential Information* required to update $\pi_{\text{ref}}$ into $\pi^*$. When real-world data deviates from this assumption, DPO is not expected to recover a globally optimal policy. Instead, it converges to a policy satisfying the power-law DID relationship (Corollary 3.2.1). If this converged policy empirically improves over $\pi_{\text{ref}}$, it indicates that the preference data encodes the Differential Information necessary for policy improvement. DPO's strong empirical performance across diverse domains (Xiao et al., 2024b) suggests that this condition often holds in practice.

## C   RELATED WORK

**Direct Preference Optimization.**   Direct Preference Optimization (DPO) (Rafailov et al., 2023) is widely used to align LMs with human preferences in a supervised manner (Xiao et al., 2024b; Liu et al., 2025b). Recent research investigates the theoretical foundations of preference optimization, connecting it to distribution matching (Korbak et al., 2022a; Dumoulin et al., 2023; Xu et al., 2024a; Liu et al., 2024b; Ji et al., 2024), and analyzing the optimization dynamics of log-likelihood

---

[6]https://iclr.cc/Conferences/2026/AuthorGuide

displacement (Pal et al., 2024; Feng et al., 2024; Mao et al., 2024). While Chen et al. (2024) reinterpret the DPO objective from a noise contrastive estimation perspective, their approach relies on the optimal policy of the KL-regularized RL objective and leaves its justification open for discussion.

We complement prior work by offering a Bayesian perspective on the justification for the reward parameterization of DPO, linking its optimality to the Differential Information captured by the preference data. This perspective explains the training properties of DPO, and also yields a novel interpretation of log-likelihood displacement, relating it to the entropy of the learned DID.

**Bayesian perspective of KL-regularized RL.** A prior work done by Korbak et al. (2022b) interprets the optimal policy of the KL-regularized RL objective $\pi^*(y) \propto \pi_{\mathrm{ref}}(y) \exp(r(y)/\beta)$ from a Bayesian perspective, showing how the reward-induced distribution $P(Y = y \mid r) \propto \exp(r(y))$ can be viewed as carrying the Bayesian-evidence towards a target policy. Because DPO learns the same optimal policy using supervised learning, there exists an inherent connection between this view and our DID perspective. Our work builds on that connection but provides a Bayesian account of DPO, characterizing the statistical structure of preference datasets, the optimality of the DPO log-ratio reward, and the resulting policy-dynamics phenomena.

## D    Interpretation of Differential Information Distribution

This section provides a probabilistic interpretation of the DIFFERENTIAL INFORMATION DISTRIBUTION (DID). Our goal is to illustrate the intuition that the DID $q_{\pi^*/\pi_{\mathrm{ref}}}$ represents the distribution over samples $y$ that carry the Differential Information needed to update the reference policy $\pi_{\mathrm{ref}}$ into the target policy $\pi^*$ through Bayesian conditioning.

### D.1    Information as an abstract event

We begin by establishing a Bayesian framework to reason about information associated with sentences. Consider the sample space $\mathcal{Y}$ of all possible sentences, assuming a uniform prior distribution $P(Y = y) = 1/|\mathcal{Y}|$.

Now, consider an abstract "event" or "property" $X$ that can be associated with sentences. This event $X$ represents some specific characteristic or information content. We can quantify the association between a sentence $y$ and the property $X$ using the conditional probability $P(X \mid Y = y)$. This term represents the likelihood that a given sentence $y$ possesses the property $X$. For instance:

1. If $P(X \mid Y = y)$ measures the probability of "$y$ being a mathematically correct sentence", then the probabilities will be either 0 or 1.
   - $P(X \mid Y =$"1+1=2"$) = 1$
   - $P(X \mid Y =$"1+0=1"$) = 1$
   - $P(X \mid Y =$"2+2=5"$) = 0$

2. If $P(X \mid Y = y)$ measures the probability of "$y$ being a safe sentence", then the probabilities can be in the range of $0 \leq P(X \mid Y = y) \leq 1$.
   - $P(X \mid Y =$"`Apples are red.`"$) = 0.99$
   - $P(X \mid Y =$"`Alcohol is good for relaxation.`"$) = 0.3$
   - $P(X \mid Y =$"`Let's promote violence!`"$) = 0$

A crucial assumption in our analysis is that the property $X$ is inherent to the sentence $y$ itself, regardless of which language model might have generated it. For instance, the mathematical correctness or safeness of a sentence should not depend on whether it came from `Mistral7B-v0.3` or `Qwen3-4B`; it's a property of the content in $y$ itself.

Formally, this means we assume that the event $X$ is conditionally independent of the generating model (*e.g.*, $\pi_{\mathrm{ref}}$) given the sentence $Y = y$:

$$P(X \mid Y = y, \pi_{\mathrm{ref}}) = P(X \mid Y = y).$$

This is equivalent to stating that the joint probability factors as

$$P(X, \pi_{\mathrm{ref}} \mid Y = y) = P(X \mid Y = y)P(\pi_{\mathrm{ref}} \mid Y = y).$$

This assumption allows us to treat $P(X \mid Y = y)$ as a property purely of the sentence $y$ and the abstract information $X$.

Note that similar independence assumptions are standard in adjacent areas. For example, in Controlled Text Generation, *Weighted Decoding* methods (Holtzman et al., 2018; Krause et al., 2021; Liu et al., 2021; Yang & Klein, 2021) factor the controlled distribution into the base model's prior distribution and a control-dependent likelihood, effectively mirroring our conditional independence formulation.

### D.2 Interpreting $P(Y = y \mid X)$

Given the likelihood $P(X \mid Y = y)$ that a sentence $y$ possesses property $X$, what does the distribution $P(Y = y \mid X)$ represent? This is the distribution over sentences for which the property $X$ holds. If $X$ represents "mathematical correctness", then sampling from $P(Y = y \mid X)$ would yield mathematically correct statements.

We can derive this distribution using Bayes' theorem and our uniform prior $P(Y = y) = 1/|\mathcal{Y}|$.

$$
\begin{aligned}
P(Y = y \mid X) &= \frac{P(X \mid Y = y)P(Y = y)}{P(X)} \\
&= \frac{P(X \mid Y = y)P(Y = y)}{\sum_{y' \in \mathcal{Y}} P(X \mid Y = y')P(Y = y')} \\
&= \frac{P(X \mid Y = y)(1/|\mathcal{Y}|)}{\sum_{y' \in \mathcal{Y}} P(X \mid Y = y')(1/|\mathcal{Y}|)} \\
&= \frac{P(X \mid Y = y)}{\sum_{y' \in \mathcal{Y}} P(X \mid Y = y')} \\
&\propto P(X \mid Y = y).
\end{aligned}
$$

This confirms the intuition: the probability of sampling a sentence $y$ that holds $X$ is directly proportional to the likelihood that sentence $y$ possesses the property $X$. Sentences that strongly exhibit property $X$ (*i.e.*, high $P(X \mid Y = y)$) are more likely to be sampled from $P(Y = y \mid X)$.

### D.3 Information difference between policies

We now focus on comparing two language models, $\pi^*$ and $\pi_{\mathrm{ref}}$, both assumed to have full support over $\mathcal{Y}$. We are interested in the difference in the information contained in these two models. We characterize this information difference as the Bayesian evidence required to update $\pi_{\mathrm{ref}}$ into $\pi^*$. We represent such information by an abstract event $X$ which we will call the DIFFERENTIAL INFORMATION that updates $\pi_{\mathrm{ref}}$ into $\pi^*$. We seek an $X$ such that conditioning $\pi_{\mathrm{ref}}(y)$ on $X$ yields $\pi^*(y)$. Formally, given $\pi_{\mathrm{ref}}(y) = P(Y = y \mid \pi_{\mathrm{ref}})$, we want $X$ to satisfy

$$
\pi^*(y) = P(Y = y \mid \pi_{\mathrm{ref}}, X).
$$

Furthermore, we maintain our key assumption that this information $X$ is intrinsic to the sentences, meaning it is conditionally independent of the prior $\pi_{\mathrm{ref}}$ given the sentence $y$:

$$
P(X \mid Y = y, \pi_{\mathrm{ref}}) = P(X \mid Y = y).
$$

In other words, the probability that a sentence $y$ holds the information $X$ does not depend on whether it was sampled from $\pi_{\mathrm{ref}}$.

Before proceeding, we should confirm that such an event $X$ can always be constructed. The following lemma guarantees its existence.

**Lemma D.1** (Existence of Differential Information). *For any two probability distributions $\pi^*, \pi_{\mathrm{ref}}$ with full support on $\mathcal{Y}$, there exists an event $X$ such that*

$$
\begin{cases}
P(X \mid Y = y, \pi_{\mathrm{ref}}) = P(X \mid Y = y) & \textit{(Conditional Independence)} \\
\pi^*(y) = P(Y = y \mid \pi_{\mathrm{ref}}, X) & \textit{(Bayesian Update)}
\end{cases}
$$

*Proof.* Define $X$ as a random variable that satisfies the conditional independence property $P(X \mid Y = y, \pi_{\mathrm{ref}}) = P(X \mid Y = y)$. We need to show that we can define $P(X \mid Y = y)$ such that the Bayesian update rule holds.

First, choose a base probability $P(X \mid \pi_{\mathrm{ref}})$ such that $0 < P(X \mid \pi_{\mathrm{ref}}) < 1/\max_{y'}\left[\frac{\pi^*(y')}{\pi_{\mathrm{ref}}(y')}\right]$. This ensures that the resulting conditional probability $P(X \mid Y = y)$ defined below is valid (*i.e.*, $0 \leq P(X \mid Y = y) \leq 1$). Now, define the likelihood of $X$ given $y$ as

$$P(X \mid Y = y) := \frac{P(X \mid \pi_{\mathrm{ref}})\pi^*(y)}{\pi_{\mathrm{ref}}(y)}.$$

Note that since $\pi_{\mathrm{ref}}$ has full support, we have $\pi_{\mathrm{ref}}(y) > 0$. We must check if $P(X \mid Y = y) \leq 1$. This holds because by our choice of $P(X \mid \pi_{\mathrm{ref}})$, we have

$$P(X \mid Y = y) = P(X \mid \pi_{\mathrm{ref}})\frac{\pi^*(y)}{\pi_{\mathrm{ref}}(y)}$$

$$\leq P(X \mid \pi_{\mathrm{ref}})\max_{y'}\left[\frac{\pi^*(y')}{\pi_{\mathrm{ref}}(y')}\right] < 1.$$

Now, using Bayes' rule we verify the Bayesian update condition:

$$P(Y = y \mid X, \pi_{\mathrm{ref}}) = \frac{P(X \mid Y = y, \pi_{\mathrm{ref}})P(Y = y \mid \pi_{\mathrm{ref}})}{P(X \mid \pi_{\mathrm{ref}})} \quad \text{(Bayes' Rule)}$$

$$= \frac{P(X \mid Y = y)\pi_{\mathrm{ref}}(y)}{P(X \mid \pi_{\mathrm{ref}})} \quad \text{(Conditional Independence)}$$

$$= \frac{\left(\frac{P(X \mid \pi_{\mathrm{ref}})\pi^*(y)}{\pi_{\mathrm{ref}}(y)}\right)\pi_{\mathrm{ref}}(y)}{P(X \mid \pi_{\mathrm{ref}})} \quad \text{(Definition of } P(X \mid Y = y))$$

$$= \frac{P(X \mid \pi_{\mathrm{ref}})\pi^*(y)}{P(X \mid \pi_{\mathrm{ref}})}$$

$$= \pi^*(y).$$

Thus, we have constructed an event $X$ satisfying both conditions. $\qquad\square$

This lemma confirms that it is always possible to conceptualize the transformation from $\pi_{\mathrm{ref}}$ to $\pi^*$ as a Bayesian update based on some underlying information $X$ that satisfies our conditional independence assumption. We defined such $X$ as the Differential Information that updates $\pi_{\mathrm{ref}}$ to $\pi^*$. Now, we connect this concept directly to the Differential Information Distribution (DID). The following theorem demonstrates that the distribution over samples conditioned on this Differential Information $X$ is precisely the normalized ratio distribution $q_{\pi^*/\pi_{\mathrm{ref}}}$.

**Theorem** (Likelihood Ratio Representation of Differential Information Distribution). *For policies $\pi^*, \pi_{\mathrm{ref}}$ over $\mathcal{Y}$ with full support, the Differential Information Distribution (DID) from $\pi_{\mathrm{ref}}$ to $\pi^*$ is equivalent to the normalized ratio distribution:*

$$P(Y = y \mid X) = \frac{\pi^*(y)/\pi_{\mathrm{ref}}(y)}{Z} := q_{\pi^*/\pi_{\mathrm{ref}}}(y),$$

*where $Z = \sum_{y' \in \mathcal{Y}} \frac{\pi^*(y')}{\pi_{\mathrm{ref}}(y')}$ is the partition function.*

*Proof.* Let $X$ be the event that satisfies Lemma D.1. The Bayes' Theorem states that

$$\pi^*(y) = P(Y = y \mid \pi_{\mathrm{ref}}, X)$$

$$= \frac{P(X \mid Y = y, \pi_{\mathrm{ref}})P(Y = y \mid \pi_{\mathrm{ref}})}{P(X \mid \pi_{\mathrm{ref}})}$$

$$= \frac{P(X \mid Y = y)P(Y = y \mid \pi_{\mathrm{ref}})}{P(X \mid \pi_{\mathrm{ref}})}.$$

We thus have $\frac{\pi^*(y)}{\pi_{\text{ref}}(y)} = \frac{P(X|Y=y)}{P(X|\pi_{\text{ref}})}$. Now, consider the following relationship:

$$\frac{\pi^*(y)}{\pi_{\text{ref}}(y)Z} = \frac{\pi^*(y)/\pi_{\text{ref}}(y)}{\sum_{y'\in\mathcal{Y}} \pi^*(y')/\pi_{\text{ref}}(y')}$$

$$= \frac{P(X \mid Y=y)/P(X \mid \pi_{\text{ref}})}{\sum_{y'\in\mathcal{Y}} P(X \mid Y=y')/P(X \mid \pi_{\text{ref}})}$$

$$= \frac{P(X \mid Y=y)}{\sum_{y'\in\mathcal{Y}} P(X \mid Y=y')}.$$

Since $P(Y = y)$ is a uniform distribution, we arrive at the relationship:

$$P(Y=y \mid X) = \frac{P(X \mid Y=y)P(Y=y)}{\sum_{y'\in\mathcal{Y}} P(X \mid Y=y')P(Y=y')}$$

$$= \frac{P(X \mid Y=y)}{\sum_{y'\in\mathcal{Y}} P(X \mid Y=y')}$$

$$= \frac{\pi^*(y)}{\pi_{\text{ref}}(y)Z}$$

$$= q_{\pi^*/\pi_{\text{ref}}}(y).$$

$\square$

**Therefore, sampling a sentence from the normalized ratio distribution $q_{\pi^*/\pi_{\text{ref}}}$ is equivalent to sampling a sentence that carries the Differential Information required to update $\pi_{\text{ref}}$ into $\pi^*$ via Bayes' rule.**

### D.4 UNCERTAINTY OF DIFFERENTIAL INFORMATION

The normalized ratio form of the DID $q_{\pi^*/\pi_{\text{ref}}}$ naturally admits an information-theoretic characterization. In particular, we can measure the uncertainty of the Differential Information by the Shannon entropy:

$$H(q_{\pi^*/\pi_{\text{ref}}}) = -\sum_y q_{\pi^*/\pi_{\text{ref}}}(y) \log q_{\pi^*/\pi_{\text{ref}}}(y).$$

This entropy quantifies how broadly the Bayesian evidence required to update $\pi_{\text{ref}}$ into $\pi^*$ is distributed across the sample space $\mathcal{Y}$.

A low-entropy DID $H(q_{\pi^*/\pi_{\text{ref}}})$ describes a deterministic Bayesian evidence that drives the update from $\pi_{\text{ref}}$ to $\pi^*$. Intuitively, if only a few samples hold that Bayesian evidence, then the DID $q_{\pi^*/\pi_{\text{ref}}}$ will be highly concentrated on a few samples that have a large enough value of $P(X \mid Y = y)$ (*i.e.*, the probability of $y$ having $X$, Appendix D.2). Therefore, the policy update from $\pi_{\text{ref}}$ to $\pi^*$ can effectively be explained by a few characteristic samples. This corresponds to information that is specific and localized, such as factual knowledge (*e.g.*, the birthplace of George Washington) where only a narrow subset of $\mathcal{Y}$ strongly supports the relevant property.

Conversely, a high-entropy DID $H(q_{\pi^*/\pi_{\text{ref}}})$ describes an uncertain Bayesian evidence that drives the update from $\pi_{\text{ref}}$ to $\pi^*$. If the evidence is spread across many possible samples, then the DID $q_{\pi^*/\pi_{\text{ref}}}$ will also be more spread-out and flatter. No single sample dominantly holds a high enough value of $P(X \mid Y = y)$, and the policy update requires a Bayesian evidence from a wide variety of samples. This corresponds to information that is general and broadly distributed, such as open-ended instruction-following (*e.g.*, writing a story about dragons) where many different completions may plausibly express the property.

**Therefore, the DID entropy provides a principled measure of how uncertain or spread-out the Differential Information is across the sample space.**

### D.5 ESTIMATION OF DID ENTROPY

To measure the Shannon entropy of the Differential Information Distribution (DID):

$$H(q_{\pi/\pi_{\text{ref}}}) = -\sum_y q_{\pi/\pi_{\text{ref}}}(y) \log q_{\pi/\pi_{\text{ref}}}(y)$$

$$= -\mathbb{E}_{y \sim q_{\pi/\pi_{\text{ref}}}} \left[ \log \frac{\pi(y)}{\pi_{\text{ref}}(y) Z} \right]$$

$$= \log Z - \mathbb{E}_{y \sim q_{\pi/\pi_{\text{ref}}}} [\log \frac{\pi(y)}{\pi_{\text{ref}}(y)}],$$

we can first estimate the log-partition function $\log Z = \log \sum_{y \in \mathcal{Y}} \frac{\pi(y)}{\pi_{\text{ref}}(y)} = \log \mathbb{E}_{y \sim \pi}[\frac{1}{\pi_{\text{ref}}(y)}]$, and then estimate the remainder term via self-normalized importance sampling. For Tables 1 and 4, we estimate the two terms in the following steps:

- To estimate $\log Z$, we sample $K = 32$ completions from $\pi$ and use the log-sum-exp trick to directly estimate $\log Z \approx \log \sum_{i=1}^{K} \exp(-\log \pi_{\text{ref}}(y_i)) - \log K$.

- To estimate $\mathbb{E}_{y \sim q_{\pi/\pi_{\text{ref}}}}[\log \frac{\pi(y)}{\pi_{\text{ref}}(y)}]$, we draw 32 samples from $\pi$ and re-weight them by $1/\pi_{\text{ref}}(y)$, which is proportional to the importance weight $q_{\pi/\pi_{\text{ref}}}(y)/\pi(y)$.

The 95% confidence intervals for the estimated DID entropy values are listed in Table 5. Overall, the difference in DID entropy across methods are statistically significant.

Note that naive auto-regressive sampling from the token-level ratio distribution is ineffective due to out-of-distribution prefixes, leading to degenerate outputs. While this method is sound for tractable output spaces, it does not scale to LLMs. This is the very reason behind our importance-sampling based approach for estimating the DID which is proportional to the sequence-level probability ratio. Alternative estimators (*e.g.*, token-level or length-normalized DID entropy) have an unclear theoretical interpretation and is not straightforward to empirically estimate. For instance, consider the length-normalized DID entropy. Normalizing the $\log Z$ term or the log-ratio term $\mathbb{E}_{y \sim q_{\pi/\pi_{\text{ref}}}}[\log \frac{\pi(y)}{\pi_{\text{ref}}(y)}]$ by token length has no clear interpretation within the DID framework and, most crucially, does not accurately measure the uncertainty in the DID defined over token *sequences*.

## E EMPIRICAL COMPARISON OF DPO AND REFERENCE-FREE REWARDS

In this section, we test whether real-world preference datasets align more closely with the DPO log-ratio reward $r = \beta \log (\pi/\pi_{\text{ref}})$ or with the reference-free reward $r = \beta \log \pi$ commonly used in SimPO (Meng et al., 2024) or CPO (Xu et al., 2024a).

For a Bradley-Terry preference distribution $p^*$ and the target policy $\pi^*$, the reference-free reward assumes that preferences directly encode the target policy:

$$p^*(y) \propto \pi^*(y)^{\beta}.$$

In contrast, the DPO log-ratio reward assumes that preferences encode *Differential Information* needed to update $\pi_{\text{ref}}$ into $\pi^*$:

$$p^*(y) \propto q_{\pi^*/\pi_{\text{ref}}}(y)^{\beta}.$$

**Experimental Setup** We trained `Mistral-7B-v0.3` on both reward forms using the Magpie-Pro and Ultra-Feedback datasets. For evaluation, we measured the average reward of sampled completions on 1,000 unseen prompts across a broad range of hyper-parameters, $\beta \in \{0.2, 0.1, 0.05, 0.02\}$, and over three training epochs. The reward scores were estimated using an off-the-shelf reward model `Skywork-Reward-V2-Llama-3.1-8B` (Liu et al., 2025a).

**Results** Table 2 shows the expected reward improvements in each setting, computed as the difference between the average reward of a policy and that under the reference policy. Formally, for a reward model $R : \mathcal{Y} \to \mathbb{R}$, we report $\mathbb{E}_{y \sim \pi}[R(y)] - \mathbb{E}_{y_{\text{ref}} \sim \pi_{\text{ref}}}[R(y_{\text{ref}})]$ to highlight the relative

Table 2: Comparison of expected reward improvements over $\pi_{\text{ref}}$ for policies trained with the DPO log-ratio reward $r = \beta \log(\pi/\pi_{\text{ref}})$ versus the reference-free reward $r = \beta \log \pi$. The largest reward improvement within the same reward parameterization form is highlighted in bold. The log-ratio reward consistently produces policies with higher expected rewards, indicating that instruction-following preferences primarily encode Differential Information rather than the target policy itself.

| Dataset | Reward | $\beta$ | Epoch-1 | Epoch-2 | Epoch-3 |
|---|---|---|---|---|---|
| Ultra-Feedback | $r = \beta \log \pi/\pi_{\text{ref}}$ | 0.2 | 5.08 | 5.73 | 6.81 |
| | | 0.1 | 6.31 | 6.04 | 6.28 |
| | | 0.05 | 7.53 | 7.95 | 8.15 |
| | | 0.02 | **8.20** | 8.07 | 6.04 |
| | $r = \beta \log \pi$ | 0.2 | 2.65 | 2.60 | **3.77** |
| | | 0.1 | 1.83 | 1.13 | 1.92 |
| | | 0.05 | 0.34 | 0.54 | 1.50 |
| | | 0.02 | 0.50 | -1.39 | -2.37 |
| Magpie-Pro | $r = \beta \log \pi/\pi_{\text{ref}}$ | 0.2 | 6.30 | 6.25 | 5.62 |
| | | 0.1 | **7.41** | 6.67 | 6.17 |
| | | 0.05 | 7.17 | 6.85 | 6.13 |
| | | 0.02 | 7.27 | 6.78 | 2.60 |
| | $r = \beta \log \pi$ | 0.2 | 1.11 | 0.80 | 0.19 |
| | | 0.1 | 1.81 | 1.81 | 2.93 |
| | | 0.05 | -1.01 | 2.13 | -4.52 |
| | | 0.02 | **3.41** | -0.87 | -5.67 |

improvement over $\pi_{\text{ref}}$. Overall, the DPO log-ratio reward consistently produces responses with higher expected reward than the reference policy, even outperforming the reference-free reward. **This provides empirical evidence that real-world preference data frequently encodes the *Differential Information* needed to improve $\pi_{\text{ref}}$ towards $\pi^*$, rather than directly encoding the target policy $\pi^*$ itself.**

# F  OPTIMAL DATASET FOR DPO

A central design choice when building DPO datasets is *how* to sample the chosen and rejected responses. Prior work has advocated opposing strategies: *strong contrasts* that maximize quality gaps (Meng et al., 2024; Xu et al., 2024b) versus *fine-grained distinctions* with minimal differences (Lin et al., 2024b; Tunstall et al., 2023; Guo et al., 2024a). We resolve this tension by showing that what matters is not *absolute* gap size but the *Differential Information* encoded by the pair $(y_w, y_\ell)$. In particular, the optimal rejection distribution should make the dataset's Differential Information distribution reflect the Differential Information between the reference and target policies. Using Corollary 3.2.1 we obtain the following closed-form characterization:

**Theorem F.1** (Optimal Distribution of Chosen and Rejected Responses)**.** *Given a preference dataset* $\mathcal{D} = \{(y_w, y_\ell) \mid y_w \sim \pi_w, y_\ell \sim \pi_\ell\}$, *if* $\pi_{\text{ref}} = \pi_w$, *then preference optimization on* $\mathcal{D}$ *using the log-ratio reward* $r = \beta \log \pi/\pi_{\text{ref}}$ *converges to* $\pi^*$ *if and only if the rejected sample distribution* $\pi_\ell$ *satisfies*

$$\pi_\ell(y) \propto \pi_{\text{ref}}(y) \left( \frac{\pi_{\text{ref}}(y)}{\pi^*(y)} \right)^\beta, \quad \forall y \in \mathcal{Y}.$$

*Likewise, if* $\pi_{\text{ref}} = \pi_\ell$, *then optimizing* $\mathcal{D}$ *using the log-ratio reward converges to* $\pi^*$ *if and only if the chosen sample distribution* $\pi_w$ *satisfies*

$$\pi_w(y) \propto \pi_{\text{ref}}(y) \left( \frac{\pi^*(y)}{\pi_{\text{ref}}(y)} \right)^\beta, \quad \forall y \in \mathcal{Y}.$$

Intuitively, Theorem F.1 states that the correct construction of preference data depends on matching the dataset's DID to the log-ratio reward used in DPO. Thus both "strong" and "fine-grained" constructions can be optimal, given that the DID from $\pi_\ell$ to $\pi_w$ aligns with the DID from $\pi_{\text{ref}}$ to $\pi^*$, up to the exponent $\beta > 0$.

*Proof.* This directly follows from Corollary 3.2.1. For any general preference dataset $\mathcal{D} = \{(y_w, y_\ell) \mid y_w \sim \pi_w, y_\ell \sim \pi_\ell\}$, the Bradley-Terry preference distribution $p^*$ must exactly follow $q_{\pi_w/\pi_\ell}$. Corollary 3.2.1 states that a power-law DID structure involving the converged policy $\pi^*$ must hold:

$$q_{\pi_w/\pi_\ell}(y) \propto q_{\pi^*/\pi_{\mathrm{ref}}}(y)^\beta, \quad \forall y \in \mathcal{Y}.$$

When $\pi_{\mathrm{ref}} = \pi_w$, for all $y \in \mathcal{Y}$, we have

$$q_{\pi_{\mathrm{ref}}/\pi_\ell}(y) \propto q_{\pi^*/\pi_{\mathrm{ref}}}(y)^\beta \iff \pi_\ell(y) \propto \pi_{\mathrm{ref}}(y) \left( \frac{\pi_{\mathrm{ref}}(y)}{\pi^*(y)} \right)^\beta.$$

Conversely, when $\pi_{\mathrm{ref}} = \pi_\ell$, for all $y \in \mathcal{Y}$, we have

$$q_{\pi_w/\pi_{\mathrm{ref}}}(y) \propto q_{\pi^*/\pi_{\mathrm{ref}}}(y)^\beta \iff \pi_w(y) \propto \pi_{\mathrm{ref}}(y) \left( \frac{\pi^*(y)}{\pi_{\mathrm{ref}}(y)} \right)^\beta.$$

$\square$

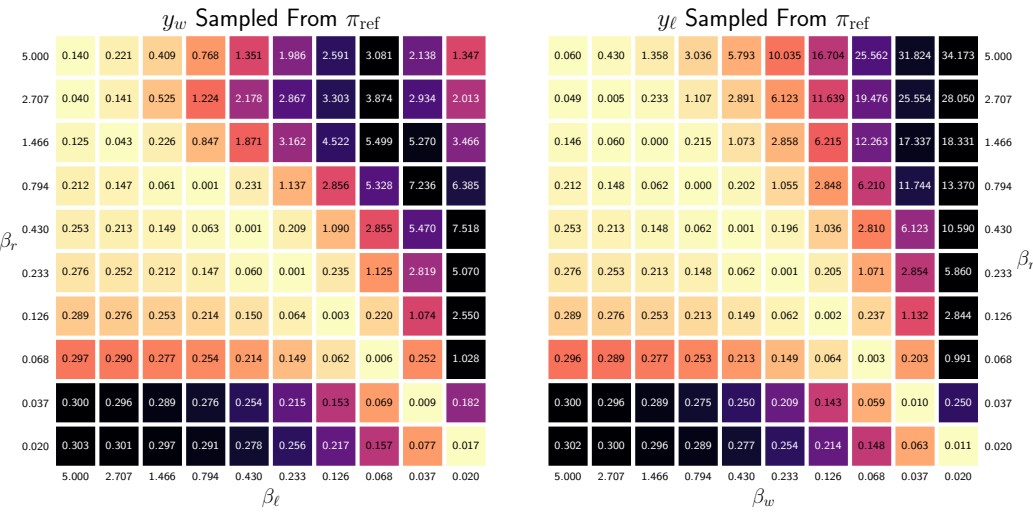

Figure 5: Convergence quality (Jensen-Shannon divergence) between the target $\pi^*$ and the converged policy $\pi$ under varying dataset exponents $\beta_\ell$ and $\beta_w$ (controlling $\pi_\ell$ and $\pi_w$ respectively), and reward scale $\beta_r$. Consistent with Theorem F.1, the best convergence occurs near the diagonal $\beta_\ell = \beta_r$ and $\beta_w = \beta_r$.

We can validate Theorem F.1 using the EBM experiments described in Section 3.3. To test Theorem F.1 we disentangle the exponent used to construct rejected samples from the scaling factor used in the DPO reward. We sample chosen responses $y_w$ from $\pi_{\mathrm{ref}}$ and draw rejected responses $y_\ell$ from $\pi_\ell(y) \propto \pi_{\mathrm{ref}}(y) \left( \frac{\pi_{\mathrm{ref}}(y)}{\pi^*(y)} \right)^{\beta_\ell}$, while training with reward $r(y) = \beta_r \log \frac{\pi(y)}{\pi_{\mathrm{ref}}(y)}$. We also test the setup where $y_\ell$ comes from $\pi_{\mathrm{ref}}$ and $y_w$ is drawn from $\pi_w(y) \propto \pi_{\mathrm{ref}}(y) \left( \frac{\pi^*(y)}{\pi_{\mathrm{ref}}(y)} \right)^{\beta_w}$. Sweeping $\beta \in [0.02, 5]$, we measure $\mathbb{D}_{\mathrm{JS}}[\pi^* \| \pi]$ for the converged policy. Figure 5 shows the minimum divergence concentrated along $\beta_\ell = \beta_r$ and $\beta_w = \beta_r$, showing that the optimal DPO dataset requires the DID from $\pi_\ell$ to $\pi_w$ to align with that from $\pi_{\mathrm{ref}}$ to $\pi^*$, up to a positive exponent $\beta$.

## G  LOG-MARGIN DYNAMICS OF DPO

Based on the power-law DID relationship in DPO (Corollary 3.2.1), we can prove how a policy ordering $\pi^* \succ \pi_{\mathrm{ref}} \succ \pi_\ell$ must exist based on increasing log-margins:

**Theorem G.1** (Log-Margin Ordered Policies of DPO). *Under the same setup as Theorem F.1, if $\pi_{\mathrm{ref}} = \pi_w$, then the following ordering of policies based on increasing log-margins must hold:*

$$\mathbb{E}_{y_w \sim \pi_{\mathrm{ref}}} \left[ \log \pi^*(y_w) \right] - \mathbb{E}_{y_\ell \sim \pi_\ell} \left[ \log \pi^*(y_\ell) \right] > \mathbb{E}_{y_w \sim \pi_{\mathrm{ref}}} \left[ \log \pi_{\mathrm{ref}}(y_w) \right] - \mathbb{E}_{y_\ell \sim \pi_\ell} \left[ \log \pi_{\mathrm{ref}}(y_\ell) \right]$$
$$> \mathbb{E}_{y_w \sim \pi_{\mathrm{ref}}} \left[ \log \pi_\ell(y_w) \right] - \mathbb{E}_{y_\ell \sim \pi_\ell} \left[ \log \pi_\ell(y_\ell) \right].$$

*Proof.* Since we assume that $\pi_{\mathrm{ref}} = \pi_w$, we have $\pi^*(y) \propto \pi_{\mathrm{ref}}(y) \cdot (q_{\pi_{\mathrm{ref}}/\pi_\ell}(y))^{\frac{1}{\beta}}$. Therefore, it follows that

$$\mathbb{E}_{y_w \sim \pi_{\mathrm{ref}}} \left[ \log \pi^*(y_w) \right] - \mathbb{E}_{y_\ell \sim \pi_\ell} \left[ \log \pi^*(y_\ell) \right] - \left( \mathbb{E}_{y_w \sim \pi_{\mathrm{ref}}} \left[ \log \pi_{\mathrm{ref}}(y_w) \right] - \mathbb{E}_{y_\ell \sim \pi_\ell} \left[ \log \pi_{\mathrm{ref}}(y_\ell) \right] \right)$$

$$= \frac{1}{\beta} \mathbb{E}_{y_w \sim \pi_{\mathrm{ref}}} \left[ \log \pi_{\mathrm{ref}}(y_w) - \log \pi_\ell(y_w) \right] - \frac{1}{\beta} \mathbb{E}_{y_\ell \sim \pi_\ell} \left[ \log \pi_{\mathrm{ref}}(y_\ell) - \log \pi_\ell(y_\ell) \right]$$

$$= \frac{1}{\beta} \mathbb{D}_{\mathrm{KL}} \left[ \pi_{\mathrm{ref}} \| \pi_\ell \right] + \frac{1}{\beta} \mathbb{D}_{\mathrm{KL}} \left[ \pi_\ell \| \pi_{\mathrm{ref}} \right] > 0.$$

Thus we have proven the top inequality (1). Next, the bottom inequality (2) can be shown by the following fact:

$$\mathbb{D}_{\mathrm{KL}} \left[ \pi_{\mathrm{ref}} \| \pi_\ell \right] > 0 > -\mathbb{D}_{\mathrm{KL}} \left[ \pi_\ell \| \pi_{\mathrm{ref}} \right]$$
$$\Rightarrow$$
$$\mathbb{E}_{y_w \sim \pi_{\mathrm{ref}}} \left[ \log \pi_{\mathrm{ref}}(y_w) - \log \pi_\ell(y_w) \right] > \mathbb{E}_{y_\ell \sim \pi_\ell} \left[ \log \pi_{\mathrm{ref}}(y_\ell) - \log \pi_\ell(y_\ell) \right].$$

$\square$

This directly yields an information-theoretic triangle inequality within the DPO framework.

**Corollary G.1.1** (Information-Theoretic Triangle Inequality of DPO). *Under the conditions of Theorem F.1, the following inequality holds:*

$$\mathbb{D}_{\mathrm{KL}} \left[ \pi_{\mathrm{ref}} \| \pi_\ell \right] + \mathbb{D}_{\mathrm{KL}} \left[ \pi_\ell \| \pi^* \right] > \mathbb{D}_{\mathrm{KL}} \left[ \pi_{\mathrm{ref}} \| \pi^* \right].$$

*Proof.* From Theorem G.1, it directly follows that

$$\mathbb{E}_{y_w \sim \pi_{\mathrm{ref}}} \left[ \log \pi^*(y_w) \right] - \mathbb{E}_{y_\ell \sim \pi_\ell} \left[ \log \pi^*(y_\ell) \right] > \mathbb{E}_{y_w \sim \pi_{\mathrm{ref}}} \left[ \log \pi_\ell(y_w) \right] - \mathbb{E}_{y_\ell \sim \pi_\ell} \left[ \log \pi_\ell(y_\ell) \right]$$
$$\iff$$
$$\mathbb{D}_{\mathrm{KL}} \left[ \pi_{\mathrm{ref}} \| \pi_\ell \right] - \mathbb{D}_{\mathrm{KL}} \left[ \pi_{\mathrm{ref}} \| \pi^* \right] > -\mathbb{D}_{\mathrm{KL}} \left[ \pi_\ell \| \pi^* \right]$$
$$\iff$$
$$\mathbb{D}_{\mathrm{KL}} \left[ \pi_{\mathrm{ref}} \| \pi_\ell \right] + \mathbb{D}_{\mathrm{KL}} \left[ \pi_\ell \| \pi^* \right] > \mathbb{D}_{\mathrm{KL}} \left[ \pi_{\mathrm{ref}} \| \pi^* \right].$$

$\square$

Although the KL-divergence does not generally satisfy a triangle inequality, Corollary G.1.1 shows that DPO enforces this specific triangle inequality for the trio $(\pi_{\mathrm{ref}}, \pi_\ell, \pi^*)$.

Corollary G.1.1 establishes a fundamental lower bound in the information "cost" (KL-divergence) of learning $\pi^*$ by contrasting $\pi_{\mathrm{ref}}$ against $\pi_\ell$. It shows that the cost of updating $\pi^*$ back into $\pi_{\mathrm{ref}}$ via $\pi_\ell$ must be larger than that of directly updating $\pi^*$ into $\pi_{\mathrm{ref}}$.

# H  LOG-LIKELIHOOD DISPLACEMENT AND DID ENTROPY

In this section, we provide a qualitative argument regarding the relationship between log-likelihood displacement (LLD) and DID entropy discussed in Section 5.2. In particular, we present the following informal claim.

**Informal Claim H.1.** *Consider a policy $\pi$ derived from $\pi_{\mathrm{ref}}$ such that $\mathbb{D}_{\mathrm{KL}} \left[ \pi \| \pi_{\mathrm{ref}} \right]$ is bounded. Assume that for any $y' \in \{ y \in \mathcal{Y} \mid \pi_{\mathrm{ref}}(y) \approx 0 \}$, we also have $\pi(y') \approx 0 \approx q_{\pi/\pi_{\mathrm{ref}}}(y')$.*

- *If $\pi$ is obtained by **reinforcing** $\pi_{\mathrm{ref}}$ (concentrating probability mass on modes of $\pi_{\mathrm{ref}}$), we expect the DID to be deterministic, corresponding to learning a lower-entropy Differential Information Distribution: $H(q_{\pi/\pi_{\mathrm{ref}}}) < H(\pi_{\mathrm{ref}})$.*

- *If $\pi$ is obtained by **smoothing** $\pi_{\text{ref}}$ (spreading probability mass more broadly), we expect the DID to be stochastic, corresponding to learning a higher-entropy Differential Information Distribution: $H(q_{\pi/\pi_{\text{ref}}}) > H(\pi_{\text{ref}})$.*

Our assumptions is as follows:

1. For any $y' \in \{y \in \mathcal{Y} \mid \pi_{\text{ref}}(y) \approx 0\}$, we have $\pi(y') \approx 0 \approx q_{\pi/\pi_{\text{ref}}}(y')$.

2. There is some reasonable upper-bound $c > 0$ such that $\mathbb{D}_{\text{KL}}\left[\pi_{\text{ref}}\|\pi\right] < c$.

The first condition assumes that $\pi_{\text{ref}}$ is "reasonably" trained, in that for "meaningless" $y'$ such that $\pi_{\text{ref}}(y') \approx 0$, we also have $\pi(y') \approx 0 \approx q_{\pi/\pi_{\text{ref}}}(y')$. The second condition states that $\pi_{\text{ref}}$ and $\pi$ should not differ significantly, such that $\mathbb{D}_{\text{KL}}\left[\pi_{\text{ref}}\|\pi\right]$ is bounded.

We now consider each cases of policy reinforcing and smoothing, and infer the relationship between $H(q_{\pi/\pi_{\text{ref}}})$ and $H(\pi_{\text{ref}})$.

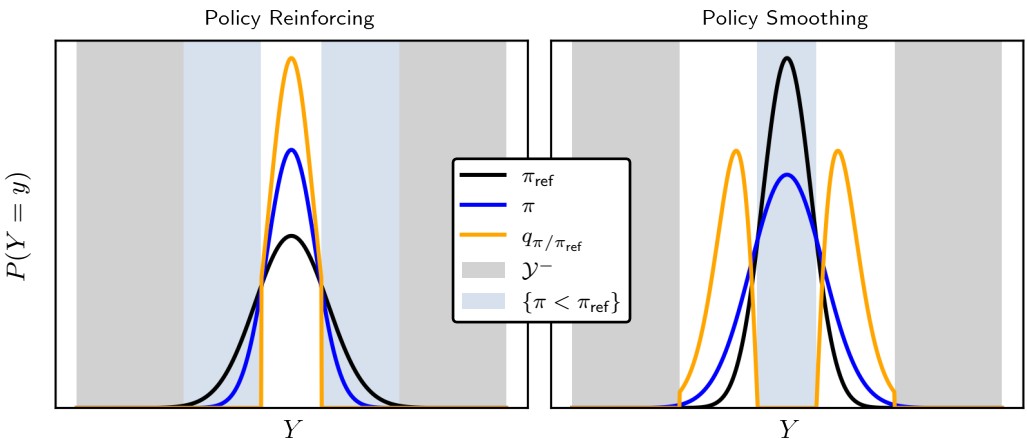

Figure 6: Illustration of policy reinforcement (*left*) and smoothing (*right*). The gray region corresponds to $\mathcal{Y}^- = \{y' \in \mathcal{Y} \mid \pi_{\text{ref}}(y') \approx 0\}$, and the light-blue region $\{\pi < \pi_{\text{ref}}\}$ corresponds to $\{\tilde{y} \in \mathcal{Y} \mid \pi(\tilde{y}) < \pi_{\text{ref}}(\tilde{y})\}$. *This plot serves only as an illustrative example and does not represent the true DID $q_{\pi^*/\pi_{\text{ref}}}$.*

**Case 1: Policy reinforcing.** We first consider the case when the policy $\pi$ reinforces its distribution with respect to the reference policy $\pi_{\text{ref}}$. If $\pi$ reinforces the distribution of $\pi_{\text{ref}}$, then under the assumption of $\pi(y') \approx 0 \approx q_{\pi/\pi_{\text{ref}}}(y')$, samples with $\pi(\tilde{y}) < \pi_{\text{ref}}(\tilde{y})$ should satisfy $\frac{\pi(\tilde{y})}{\pi_{\text{ref}}(\tilde{y})} < 1 \approx \frac{\pi(y')}{\pi_{\text{ref}}(y')}$. Since $q_{\pi/\pi_{\text{ref}}}(\tilde{y}) < q_{\pi/\pi_{\text{ref}}}(y') \approx 0$, we expect $q_{\pi/\pi_{\text{ref}}}(y)$ to concentrate its probability mass towards samples with $\pi(y) > \pi_{\text{ref}}(y)$ and sufficient probability of $\pi_{\text{ref}}(y) > 0$. Thus, the number of samples $y$ with sufficiently large $q_{\pi/\pi_{\text{ref}}}(y)$ is expected to be far less than the number of samples with sufficiently large $\pi_{\text{ref}}(y)$. As a result, we expect the relationship: $H(q_{\pi/\pi_{\text{ref}}}) < H(\pi_{\text{ref}})$. We visualize this intuition as the left plot in Figure 6.

**Case 2: Policy smoothing.** Now, consider the case when the policy $\pi$ smooths its distribution with respect to $\pi_{\text{ref}}$. A key relation between $H(q_{\pi/\pi_{\text{ref}}})$ and $H(\pi_{\text{ref}})$ is the following:

$$H(q_{\pi/\pi_{\text{ref}}}) - H(\pi_{\text{ref}}) =$$

$$\mathbb{D}_{\text{KL}}\left[q_{\pi/\pi_{\text{ref}}}\|\pi\right] - \mathbb{D}_{\text{KL}}\left[q_{\pi/\pi_{\text{ref}}}\|\pi_{\text{ref}}\right] + \mathbb{D}_{\text{KL}}\left[\pi_{\text{ref}}\|q_{\pi/\pi_{\text{ref}}}\right] - \mathbb{D}_{\text{KL}}\left[\pi_{\text{ref}}\|\pi\right].$$

Since we have assumed that $\pi$ and $\pi_{\text{ref}}$ do not diverge significantly, we mainly expect the last two terms to dominate:

$$\left|\mathbb{D}_{\text{KL}}\left[q_{\pi/\pi_{\text{ref}}}\|\pi\right] - \mathbb{D}_{\text{KL}}\left[q_{\pi/\pi_{\text{ref}}}\|\pi_{\text{ref}}\right]\right| < \left|\mathbb{D}_{\text{KL}}\left[\pi_{\text{ref}}\|q_{\pi/\pi_{\text{ref}}}\right] - \mathbb{D}_{\text{KL}}\left[\pi_{\text{ref}}\|\pi\right]\right|.$$

See the right plot in Figure 6 for a visual intuition. When $\pi$ smooths its distribution with respect to $\pi_{\text{ref}}$, we can expect $\mathbb{D}_{\text{KL}}\left[\pi_{\text{ref}}\|q_{\pi/\pi_{\text{ref}}}\right] > \mathbb{D}_{\text{KL}}\left[\pi_{\text{ref}}\|\pi\right]$. This results in the relationship: $H(q_{\pi/\pi_{\text{ref}}}) > H(\pi_{\text{ref}})$.

# I   DID ENTROPY AND PERFORMANCE TRENDS OF SIMPO

Table 3: Performance trends of `Qwen3-4B` trained on Magpie-Pro with the addition of SimPO. The DID entropy ($H(q_{\pi^*/\pi_{\mathrm{ref}}})$, [nats]) is estimated by importance sampling (Appendix D.5). We report the win-rate [%] for Arena-Hard-v0.1 and ELO score for Wild-Bench-v2. For knowledge-intensive QA, we report mean reciprocal rank across 8 QA benchmarks (QA). Overall, SimPO closely follows DPO's behavior in learning high-entropy DID and achieving strong performance on open-ended tasks, while exhibiting lower factual precision (QA) than DPO-PG.

| Method | $H(q_{\pi^*/\pi_{\mathrm{ref}}})$ | Arena-Hard-v0.1 ($\uparrow$) | Wild-Bench-v2 ($\uparrow$) | QA ($\uparrow$) |
|---|---|---|---|---|
| DPO ($\beta = 1.0$) | 9158.4 | 30.7 | 1134.4 | 0.37 |
| DPO ($\beta = 0.2$) | 4765.9 | 28.1 | 1146.2 | 0.30 |
| DPO ($\beta = 0.1$) | 7663.3 | 27.5 | 1148.2 | 0.23 |
| DPO ($\beta = 0.05$) | 6801.2 | **43.7** | 1164.5 | 0.20 |
| **SimPO** ($\beta = 10, \gamma = 5$) | 1071.4 | 42.8 | **1177.5** | 0.50 |
| DPO-PG | 388.2 | 37.4 | 1148.5 | **0.85** |

This section provides additional analysis on whether the Differential Information Distribution (DID) entropy trends identified in Section 5 extend to alternative objectives such as SimPO (Meng et al., 2024). Before presenting the results, we briefly note that comparisons with other LLD-mitigation objectives (DPOP (Pal et al., 2024), Cal-DPO (Xiao et al., 2024a)) were attempted but was shown to be infeasible at our scale: for 4-7B models trained on 100K preference samples over 5 epochs, these objectives exhibited high sensitivity to hyper-parameters and failure to mitigate LLD (see Appendix K). This instability was one of the motivations behind DPO-PG (Appendix K), which stably avoids LLD via projected gradient descent without introducing new hyper-parameters.

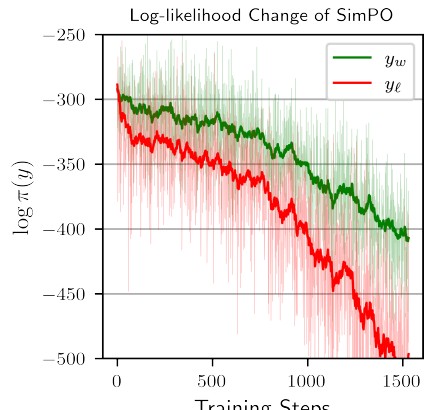

Figure 7: Likelihood change of SimPO when training `Qwen3-4B` on the Magpie-Pro dataset. SimPO also decreases the likelihood of chosen responses (LLD).

**Experimental Setup.**   We trained `Qwen3-4B` using the SimPO objective on the Magpie-Pro dataset. During the experiments, SimPO also exhibited instability and was prone to over-optimization. Due to limited compute, we trained SimPO for 2 epochs and saved model checkpoints every 20,000 samples (10 checkpoints total) with a learning rate of 8e-7. Following the same model-selection protocol used for DPO and DPO-PG (Appendix M.2), we selected the best-performing checkpoint for evaluation.

**Results.**   Figure 7 shows that SimPO also exhibits LLD, similar to the likelihood trends of DPO (Figure 14). Table 3 reports the DID entropy and downstream performance trends from Table 1 with the addition of SimPO. As with DPO, SimPO learns a high-entropy DID with strong performance on open-ended tasks (Arena-Hard-v0.1 and Wild-Bench-v2), at the cost of weaker factual accuracy (QA) compared to DPO-PG.[7]

These findings demonstrate that our DID entropy based analysis provides a comprehensive explanation of training dynamics and downstream behavior across diverse preference-learning methods.

---

[7] QA MRR values differ slightly from Table 1 because MRR is a relative metric which shifts when additional methods (SimPO) are included.

## J DERIVATIONS AND PROOFS

In this section we provide the detailed proofs supporting our theoretical findings.

### J.1 PROOF FOR EQUIVALENCE OF PREFERENCE OPTIMIZATION

**Theorem** (Preference vs. Distribution Matching (Dumoulin et al., 2023)). *Let $\mathcal{D} = \{(y_w, y_\ell)\}$ be a sufficiently large preference dataset where the sets of $y_w$ and $y_\ell$ cover $\mathcal{Y}$. Then preference optimization on $\mathcal{D}$ is equivalent to fitting the reward-induced distribution $P(Y = y \mid r)$ to the implicit preference distribution $p^*(y)$:*

$$\max_r \mathbb{E}_{(y_w, y_\ell) \sim \mathcal{D}} \left[ \log \sigma(r(y_w) - r(y_\ell)) \right] \iff \min_r \mathbb{D}_{\mathrm{KL}} \left[ p^*(y) \| P(Y = y \mid r) \right].$$

We restate the proof in Dumoulin et al. (2023) for reference.

*Proof.* Recall from Section 2 that we model the ground truth probability of $y_1$ being preferred over $y_2$ as

$$p^*(y_1 \succ y_2) = \frac{\pi_w(y_1)\pi_\ell(y_2)}{\pi_w(y_1)\pi_\ell(y_2) + \pi_w(y_2)\pi_\ell(y_1)}.$$

Now, for a sufficiently large preference dataset $\mathcal{D}$, we can show that preference optimization is equivalent to minimizing the KL-divergence between the preference distributions. First, observe the following relationship:

$$\mathbb{E}_{(y_w, y_\ell) \sim \mathcal{D}} \left[ \log \sigma(r(y_w) - r(y_\ell)) \right] = \sum_{(y_w, y_\ell) \in \mathcal{Y} \times \mathcal{Y}} \pi_w(y_w)\pi_\ell(y_\ell) \log \sigma(r(y_w) - r(y_\ell))$$

$$= \sum_{(y_w, y_\ell) \in \mathcal{I}} \left( \pi_w(y_w)\pi_\ell(y_\ell) + \pi_w(y_\ell)\pi_\ell(y_w) \right) \left[ \frac{\pi_w(y_w)\pi_\ell(y_\ell)}{\pi_w(y_w)\pi_\ell(y_\ell) + \pi_w(y_\ell)\pi_\ell(y_w)} \log \sigma(r(y_w) - r(y_\ell)) \right.$$

$$\left. + \frac{\pi_w(y_\ell)\pi_\ell(y_w)}{\pi_w(y_w)\pi_\ell(y_\ell) + \pi_w(y_\ell)\pi_\ell(y_w)} \log \sigma(r(y_\ell) - r(y_w)) \right]$$

$$= -\frac{1}{2} \mathbb{E}_{(y_w, y_\ell) \sim \mathcal{D}} \left[ - p^*(y_w \succ y_\ell) \log p(y_w \succ y_\ell \mid r) - p^*(y_\ell \succ y_w) \log p(y_\ell \succ y_w \mid r) \right]$$

$$= -\frac{1}{2} \mathbb{E}_{(y_w, y_\ell) \sim \mathcal{D}} \left[ \mathbb{D}_{\mathrm{KL}} \left[ p^*(y_w \succ y_\ell) \| p(y_w \succ y_\ell \mid r) \right] \right] + C,$$

where $\mathcal{I} = \{(y_i, y_j) \in \mathcal{Y} \times \mathcal{Y} : i > j\}$ is the set of ordered distinct pairs $(y_i, y_j)$, and $C$ is a constant term independent of $r$. Therefore, preference optimization is equivalent to minimizing the KL-divergence between preference distributions:

$$\arg\max_r \mathbb{E}_{(y_w, y_\ell) \sim \mathcal{D}} \left[ \log \sigma(r(y_w) - r(y_\ell)) \right]$$

$$= \arg\min_r \mathbb{E}_{(y_w, y_\ell) \sim \mathcal{D}} \left[ \mathbb{D}_{\mathrm{KL}} \left[ p^*(y_w \succ y_\ell) \| p(y_w \succ y_\ell \mid r) \right] \right].$$

Now, for any two reward parameterizations $r_1$ and $r_2$, $\mathbb{D}_{\mathrm{KL}} \left[ p(y_w \succ y_\ell \mid r_1) \| p(y_w \succ y_\ell \mid r_2) \right]$ is minimized to 0 if and only if $r_1(y) = r_2(y) + C$ for all $y \in \mathcal{Y}$ and for some constant $C$. If we let $r_1(y) = \log p^*(y)$, we have $p(y_w \succ y_\ell \mid r_1) = p^*(y_w \succ y_\ell)$. Next, set $r(y) = r_2(y)$ and the following holds:

$$\mathbb{E}_{(y_w, y_\ell) \sim \mathcal{D}} \left[ \mathbb{D}_{\mathrm{KL}} \left[ p(y_w \succ y_\ell \mid r_1) \| p(y_w \succ y_\ell \mid r_2) \right] \right] = 0 \iff$$

$$\mathbb{E}_{(y_w, y_\ell) \sim \mathcal{D}} \left[ \mathbb{D}_{\mathrm{KL}} \left[ p^*(y_w \succ y_\ell) \| p(y_w \succ y_\ell \mid r) \right] \right] = 0 \iff$$

$$\forall y \in \mathcal{Y} : \log p^*(y) = r(y) + C \iff$$

$$\forall y \in \mathcal{Y} : p^*(y) \propto \exp(r(y)) \iff$$

$$\forall y \in \mathcal{Y} : p^*(y) = P(Y = y \mid r) \iff$$

$$\mathbb{D}_{\mathrm{KL}} \left[ p^*(y) \| P(Y = y \mid r) \right] = 0.$$

Therefore, for any reward parameterization $r : \mathcal{Y} \to \mathbb{R}$, the preference optimization objective is optimized only when the reward induced distribution $P(Y = y \mid r) := \frac{\exp(r(y))}{\sum_{y' \in \mathcal{Y}} \exp(r(y'))}$ is exactly the same as the ground truth preference distribution $p^*(y)$. $\qquad \square$

## J.2 PROOF FOR PREFERENCES ENCODING DIFFERENTIAL INFORMATION

**Theorem** (Preferences Encoding Differential Information). *Consider a preference dataset $\mathcal{D} = \{(y_w, y_\ell) \mid y_w \sim \pi_w, y_\ell \sim \pi_\ell\}$. Let $\pi^*$ be the target policy. If the Differential Information Distribution between policies match up to an exponent $\beta > 0$:*

$$q_{\pi_w/\pi_\ell}(y) \propto q_{\pi^*/\pi_{\text{ref}}}(y)^\beta, \quad \forall y \in \mathcal{Y},$$

*then the preference probability $p^*(y_w \succ y_\ell)$ can be expressed as preferences induced by the DID:*

$$p^*(y_w \succ y_\ell) = \sigma\left(\beta \log q_{\pi^*/\pi_{\text{ref}}}(y_w) - \beta \log q_{\pi^*/\pi_{\text{ref}}}(y_\ell)\right).$$

*Proof.* The relationship follows by directly applying the power-law DID relationship to the ground-truth preference probability.

$$
\begin{aligned}
p^*(y_1 \succ y_2) &= \frac{\pi_w(y_1)\pi_\ell(y_2)}{\pi_w(y_1)\pi_\ell(y_2) + \pi_w(y_2)\pi_\ell(y_1)} \\
&= \frac{\frac{\pi_w(y_1)}{\pi_\ell(y_1)}}{\frac{\pi_w(y_1)}{\pi_\ell(y_1)} + \frac{\pi_w(y_2)}{\pi_\ell(y_2)}} \\
&= \sigma\left(\log \frac{\pi_w(y_1)}{\pi_\ell(y_1)} - \log \frac{\pi_w(y_2)}{\pi_\ell(y_2)}\right) \\
&= \sigma\left(\log q_{\pi_w/\pi_\ell}(y_1) - \log q_{\pi_w/\pi_\ell}(y_2)\right) \\
&= \sigma\left(\beta \log q_{\pi^*/\pi_{\text{ref}}}(y_1) - \beta \log q_{\pi^*/\pi_{\text{ref}}}(y_2)\right).
\end{aligned}
$$

$\square$

## J.3 PROOF FOR OPTIMAL REWARD FOR LEARNING DIFFERENTIAL INFORMATION

**Theorem** (Optimal Reward for Learning Differential Information). *Let $\mathcal{D}$ be a preference dataset satisfying Theorem 3.1, encoding the Differential Information required to learn the target policy $\pi^*$. Then, for some constant $C$, we have*

$$\pi^* = \arg\max_\pi \mathbb{E}_{(y_w,y_\ell)\sim\mathcal{D}}\left[\log \sigma(r(y_w) - r(y_\ell))\right] \iff r(y) = \beta \log \frac{\pi(y)}{\pi_{\text{ref}}(y)} + C.$$

*for some constant $C$.*

*Proof.* The equivalence between preference optimization and distribution matching (Theorem 2.1) yields the following relationship:

$$
\begin{aligned}
\mathbb{E}_{(y_w,y_\ell)\sim\mathcal{D}}\left[\mathbb{D}_{\text{KL}}\left[p^*(y_w \succ y_\ell)\|p(y_w \succ y_\ell \mid r)\right]\right] = 0 &\iff \\
\mathbb{E}_{(y_w,y_\ell)\sim\mathcal{D}}\left[\mathbb{D}_{\text{KL}}\left[p(y_w \succ y_\ell \mid r^*)\|p(y_w \succ y_\ell \mid r)\right]\right] = 0 &\iff \\
\mathbb{D}_{\text{KL}}\left[P(Y = y \mid r^*)\|P(Y = y \mid r)\right] = 0,
\end{aligned}
$$

where $r^* = \beta \log \frac{\pi^*}{\pi_{\text{ref}}}$. Now, observe the following relationship:

$$
\begin{aligned}
\forall y \in \mathcal{Y}, \pi^*(y) = \pi(y) &\iff \\
\forall y \in \mathcal{Y}, q_{\pi^*/\pi_{\text{ref}}}(y) = q_{\pi/\pi_{\text{ref}}}(y) &\iff \\
\forall y \in \mathcal{Y}, q_{\pi^*/\pi_{\text{ref}}}(y)^\beta = q_{\pi/\pi_{\text{ref}}}(y)^\beta &\iff \\
\mathbb{D}_{\text{KL}}\left[q_{\pi^*/\pi_{\text{ref}}}(y)^\beta\|q_{\pi/\pi_{\text{ref}}}(y)^\beta\right] = 0 &\iff \\
\mathbb{D}_{\text{KL}}\left[P(Y = y \mid r^*)\|q_{\pi/\pi_{\text{ref}}}(y)^\beta\right] = 0,
\end{aligned}
$$

where the last line follows from the fact that $r^* = \beta \log \frac{\pi^*}{\pi_{\text{ref}}}$.

Therefore, in order to have the following equivalence:

$$\mathbb{E}_{(y_w,y_\ell)\sim\mathcal{D}}\left[p^*(y_w \succ y_\ell)\|p(y_w \succ y_\ell \mid r)\right] = 0 \iff \pi^* = \pi,$$

we must have $\mathbb{D}_{\mathrm{KL}} \left[ P(Y = y \mid r) \| q_{\pi/\pi_{\mathrm{ref}}}(y)^\beta \right] = 0$. In other words, we require

$$\mathbb{D}_{\mathrm{KL}} \left[ P(Y = y \mid r) \| q_{\pi/\pi_{\mathrm{ref}}}(y)^\beta \right] = 0 \iff$$

$$\forall y \in \mathcal{Y}, P(Y = y \mid r) = q_{\pi/\pi_{\mathrm{ref}}}(y)^\beta \iff$$

$$\forall y \in \mathcal{Y}, \exp(r(y)) \propto (\frac{\pi(y)}{\pi_{\mathrm{ref}}(y)})^\beta \iff$$

$$\forall y \in \mathcal{Y}, r(y) = \beta \log \frac{\pi(y)}{\pi_{\mathrm{ref}}(y)} + C,$$

for some constant $C$. $\qquad\square$

### J.4 PROOF FOR POWER-LAW STRUCTURE OF DPO

**Corollary** (DID Power-Law of DPO). *Consider a preference dataset $\mathcal{D} = \{(y_w, y_\ell) \mid y_w \sim \pi_w, y_\ell \sim \pi_\ell\}$ and a policy $\pi^*$ obtained as a stationary point of preference optimization using the log-ratio reward $r = \beta \log(\pi/\pi_{\mathrm{ref}})$ on $\mathcal{D}$. Then, a power-law relationship between the DID of policies must hold:*

$$q_{\pi_w/\pi_\ell}(y) \propto q_{\pi^*/\pi_{\mathrm{ref}}}(y)^\beta, \quad \forall y \in \mathcal{Y}.$$

*Proof.* According to Theorem 2.1, the converged policy $\pi^*$ obtained by optimizing $\mathcal{D}$ with $r_{\mathrm{DPO}} = \beta \log \pi/\pi_{\mathrm{ref}}$ must follow $\pi^*(y) \propto \pi_{\mathrm{ref}}(y) \cdot (p^*(y))^{\frac{1}{\beta}}$ due to the following:

$$p^*(y) = P(Y = y \mid r_{\mathrm{DPO}}) \propto q_{\pi^*/\pi_{\mathrm{ref}}}(y)^\beta, \quad \forall y \in \mathcal{Y}$$

$$\iff (p^*(y))^{\frac{1}{\beta}} \propto q_{\pi^*/\pi_{\mathrm{ref}}}(y), \quad \forall y \in \mathcal{Y}$$

$$\iff \pi^*(y) \propto \pi_{\mathrm{ref}}(y) \cdot (p^*(y))^{\frac{1}{\beta}}. \quad \forall y \in \mathcal{Y}$$

Meanwhile, it can also be shown that $p^* = q_{\pi_w/\pi_\ell}$. This because the reward $r' = \log \pi_w/\pi_\ell$ perfectly fits the ground-truth preference distribution. For all $y_1, y_2 \in \mathcal{Y} \times \mathcal{Y}$,

$$p^*(y_1 \succ y_2) = \frac{\pi_w(y_1)\pi_\ell(y_2)}{\pi_w(y_1)\pi_\ell(y_2) + \pi_w(y_2)\pi_\ell(y_1)}$$

$$= \sigma \left( \log \frac{\pi_w(y_1)}{\pi_\ell(y_1)} - \log \frac{\pi_w(y_2)}{\pi_\ell(y_2)} \right)$$

$$= \sigma \left( r'(y_1) - r'(y_2) \right)$$

$$\Rightarrow p^*(y) = P(Y = y \mid r') = q_{\pi_w/\pi_\ell}(y), \quad \forall y \in \mathcal{Y} \quad \text{(Theorem 2.1)}.$$

Since $\pi^*(y) \propto \pi_{\mathrm{ref}}(y) \cdot (p^*(y))^{\frac{1}{\beta}}$ and $p^* = q_{\pi_w/\pi_\ell}$, the power-law DID relationship $q_{\pi_w/\pi_\ell}(y) \propto q_{\pi^*/\pi_{\mathrm{ref}}}(y)^\beta$ follows directly.

Note that this result recovers the findings of Pan et al. (2025), where the authors derive the power-law DID relationship from the functional derivative of the DPO loss. In contrast, our proof takes an alternative approach by leveraging the distribution matching result of Theorem 2.1 (Dumoulin et al., 2023). $\qquad\square$

### J.5 PROOF FOR LOG-LIKELIHOOD CHANGES IN DPO

**Theorem** (Log-Likelihood Change of DPO). *Consider a preference dataset $\mathcal{D} = \{(y_w, y_\ell) \mid y_w \sim \pi_{\mathrm{ref}}, y_\ell \sim \pi_\ell\}$, and $\pi^*$ obtained by preference optimization on $\mathcal{D}$ using the log-ratio reward $r = \beta \log \pi/\pi_{\mathrm{ref}}$. Then, for any $\beta > 0$, $\pi^*$ must decrease the average log-likelihood of $y_\ell$:*

$$\mathbb{E}_{y_\ell \sim \pi_\ell} \left[ \log \pi^*(y_\ell) \right] < \mathbb{E}_{y_\ell \sim \pi_\ell} \left[ \log \pi_{\mathrm{ref}}(y_\ell) \right].$$

*Conversely, if $\pi_{\mathrm{ref}}$ was fine-tuned on $y_\ell$ (i.e., $y_\ell \sim \pi_{\mathrm{ref}}$), then, for any $\beta \geq 1$, $\pi^*$ must increase the average log-likelihood of $y_w$:*

$$\mathbb{E}_{y_w \sim \pi_w} \left[ \log \pi^*(y_w) \right] > \mathbb{E}_{y_w \sim \pi_w} \left[ \log \pi_{\mathrm{ref}}(y_w) \right].$$

*Proof.*

**Case $\pi_{\mathrm{ref}} = \pi_w$:** Assume $\beta > 0$. Let $Z = \sum_{y \in \mathcal{Y}} \pi_{\mathrm{ref}}(y) \cdot (\frac{\pi_{\mathrm{ref}}(y)}{\pi_\ell(y)})^{\frac{1}{\beta}}$. It can be shown that $\log Z > 0$ due to the following:

$$\log Z = \log \sum_{y \in \mathcal{Y}} \pi_{\mathrm{ref}}(y) \cdot (\frac{\pi_{\mathrm{ref}}(y)}{\pi_\ell(y)})^{\frac{1}{\beta}}$$

$$= \log \sum_{y \in \mathcal{Y}} \pi_\ell(y) \cdot (\frac{\pi_{\mathrm{ref}}(y)}{\pi_\ell(y)})^{1 + \frac{1}{\beta}}$$

$$= \log \mathbb{E}_{y \sim \pi_\ell} \left[ (\frac{\pi_{\mathrm{ref}}(y)}{\pi_\ell(y)})^{1 + \frac{1}{\beta}} \right]$$

$$> \log \left( \mathbb{E}_{y \sim \pi_\ell} \left[ \frac{\pi_{\mathrm{ref}}(y)}{\pi_\ell(y)} \right] \right)^{1 + \frac{1}{\beta}} \quad \text{(Jensen's Inequality)}$$

$$= (1 + \frac{1}{\beta}) \log 1 = 0.$$

Since $\pi^*(y) \propto \pi_{\mathrm{ref}}(y) \cdot p^*(y)^{\frac{1}{\beta}}$ and $p^* = q_{\pi_w/\pi_\ell} = q_{\pi_{\mathrm{ref}}/\pi_\ell}$, it follows that $\pi^*(y) \propto \pi_{\mathrm{ref}}(y) \cdot (q_{\pi_{\mathrm{ref}}/\pi_\ell}(y))^{\frac{1}{\beta}}$. Therefore, we have

$$\mathbb{E}_{y_\ell \sim \pi_\ell} [\log \pi^*(y_\ell) - \log \pi_{\mathrm{ref}}(y_\ell)]$$

$$= \frac{1}{\beta} \sum_{y \in \mathcal{Y}} \pi_\ell(y) \log \frac{\pi_{\mathrm{ref}}(y)}{\pi_\ell(y) Z}$$

$$= -\frac{1}{\beta} \mathbb{D}_{\mathrm{KL}} [\pi_\ell \| \pi_{\mathrm{ref}}] - \log Z < 0 \quad \because \mathbb{D}_{\mathrm{KL}} [\pi_\ell \| \pi_{\mathrm{ref}}] > 0 \text{ and } \log Z > 0.$$

**Case $\pi_{\mathrm{ref}} = \pi_\ell$:** Assume $\beta \geq 1$ and $\pi_{\mathrm{ref}} \neq \pi_w$. Let $Z = \sum_{y \in \mathcal{Y}} \pi_{\mathrm{ref}}(y) \cdot (\frac{\pi_w(y)}{\pi_{\mathrm{ref}}(y)})^{\frac{1}{\beta}}$. It can be shown that $\log Z < 0$ due to the following:

$$\log Z = \log \sum_{y \in \mathcal{Y}} \pi_{\mathrm{ref}}(y) \cdot (\frac{\pi_w(y)}{\pi_{\mathrm{ref}}(y)})^{\frac{1}{\beta}}$$

$$= \log \sum_{y \in \mathcal{Y}} \pi_w(y)^{\frac{1}{\beta}} \cdot (\pi_{\mathrm{ref}}(y))^{1 - \frac{1}{\beta}}$$

$$< \log \left( \left( \sum_{y \in \mathcal{Y}} \pi_w(y) \right)^{\frac{1}{\beta}} \cdot \left( \sum_{y \in \mathcal{Y}} \pi_{\mathrm{ref}}(y) \right)^{1 - \frac{1}{\beta}} \right) \quad \text{(Hölder's Inequality)}$$

$$= \log (1 \cdot 1) = 0.$$

Since $\pi^*(y) \propto \pi_{\mathrm{ref}}(y) \cdot p^*(y)^{\frac{1}{\beta}}$ and $p^* = q_{\pi_w/\pi_\ell} = q_{\pi_w/\pi_{\mathrm{ref}}}$, it follows that $\pi^*(y) \propto \pi_{\mathrm{ref}}(y) \cdot (q_{\pi_w/\pi_{\mathrm{ref}}}(y))^{\frac{1}{\beta}}$. Therefore, we have

$$\mathbb{E}_{y_w \sim \pi_w} [\log \pi^*(y_\ell) - \log \pi_{\mathrm{ref}}(y_\ell)]$$

$$= \frac{1}{\beta} \sum_{y \in \mathcal{Y}} \pi_w(y) \log \frac{\pi_w(y)}{\pi_{\mathrm{ref}}(y) Z}$$

$$= \frac{1}{\beta} \mathbb{D}_{\mathrm{KL}} [\pi_w \| \pi_{\mathrm{ref}}] - \log Z > 0 \quad \because \mathbb{D}_{\mathrm{KL}} [\pi_w \| \pi_{\mathrm{ref}}] > 0 \text{ and } \log Z < 0.$$

$\square$

## J.6 PROOF FOR PREFERENCE DATA STRENGTH

**Theorem** (Adaptive Policy Exploration of DPO). *Let $\mathcal{D} = \{(y_w, y_\ell) \mid y_w \sim \pi_{\mathrm{ref}}, \ y_\ell \sim \pi_\ell\}$ be a preference dataset with an implicit Bradley-Terry preference distribution $p_{\mathcal{D}}^*$. Consider another dataset $\mathcal{D}' = \{(y_w, y_\ell)\}$ whose implicit Bradley-Terry distribution $p_{\mathcal{D}'}^*$ is a "sharpened" version of $p_{\mathcal{D}}^*$, in the sense that there exists $\alpha > 1$ such that for all pairs $(y_w, y_\ell) \in \mathcal{Y} \times \mathcal{Y}$,*

$$p_{\mathcal{D}'}^*(y_w \succ y_\ell) = \frac{\left(p_{\mathcal{D}}^*(y_w)\right)^\alpha}{\left(p_{\mathcal{D}}^*(y_w)\right)^\alpha + \left(p_{\mathcal{D}}^*(y_\ell)\right)^\alpha} = \exp\left(\alpha \log p_{\mathcal{D}}^*(y_w) - \alpha \log p_{\mathcal{D}}^*(y_\ell)\right).$$

*For the same reference policy $\pi_{\mathrm{ref}}$ and any $\beta > 0$, let $\pi_{\mathcal{D}}^*$ and $\pi_{\mathcal{D}'}^*$ denote the policies obtained by preference optimization on $\mathcal{D}$ and $\mathcal{D}'$, respectively, using the log-ratio reward $r = \beta \log \pi / \pi_{\mathrm{ref}}$. Then the strengthened dataset $\mathcal{D}'$ induces a strictly larger divergence from the reference:*

$$\mathbb{D}_{\mathrm{KL}}\left[\pi_{\mathrm{ref}} \| \pi_{\mathcal{D}'}^*\right] > \mathbb{D}_{\mathrm{KL}}\left[\pi_{\mathrm{ref}} \| \pi_{\mathcal{D}}^*\right].$$

*Proof.* Let us denote $Z_{\mathcal{D}} = \sum_y \pi_{\mathrm{ref}}(y)\left(\frac{\pi_{\mathrm{ref}}(y)}{\pi_\ell(y)}\right)^{\frac{1}{\beta}}$ and $Z_{\mathcal{D}'} = \sum_y \pi_{\mathrm{ref}}(y)\left(\frac{\pi_{\mathrm{ref}}(y)}{\pi_\ell(y)}\right)^{\frac{\alpha}{\beta}}$. Observe the following:

$$\pi_{\mathcal{D}}^*(y) = \frac{\pi_{\mathrm{ref}}(y) \cdot \left(\frac{\pi_{\mathrm{ref}}(y)}{\pi_\ell(y)}\right)^{\frac{1}{\beta}}}{Z_{\mathcal{D}}}, \quad \pi_{\mathcal{D}}^*(y) = \frac{\pi_{\mathrm{ref}}(y) \cdot \left(\frac{\pi_{\mathrm{ref}}(y)}{\pi_\ell(y)}\right)^{\frac{\alpha}{\beta}}}{Z_{\mathcal{D}'}}.$$

Therefore, we can express the difference in the KL-divergence as

$$\mathbb{D}_{\mathrm{KL}}\left[\pi_{\mathrm{ref}} \| \pi_{\mathcal{D}'}^*\right] - \mathbb{D}_{\mathrm{KL}}\left[\pi_{\mathrm{ref}} \| \pi_{\mathcal{D}}^*\right] = \sum_{y \in \mathcal{Y}} \pi_{\mathrm{ref}}(y) \log \frac{\pi_{\mathrm{ref}}(y)}{\pi_{\mathcal{D}'}^*(y)} - \sum_{y \in \mathcal{Y}} \pi_{\mathrm{ref}}(y) \log \frac{\pi_{\mathrm{ref}}(y)}{\pi_{\mathcal{D}}^*(y)}$$

$$= \sum_{y \in \mathcal{Y}} \pi_{\mathrm{ref}}(y) \log \frac{\pi_{\mathcal{D}}^*(y)}{\pi_{\mathcal{D}'}^*(y)}$$

$$= \log \frac{Z_{\mathcal{D}'}}{Z_{\mathcal{D}'}} + \frac{1-\alpha}{\beta} \mathbb{D}_{\mathrm{KL}}\left[\pi_{\mathrm{ref}} \| \pi_\ell\right].$$

Now, let $r(y) = \frac{\pi_{\mathrm{ref}}(y)}{\pi_\ell(y)}$ and $X = \log r(y)$. Also, define the cumulant-generating function $K(t)$:

$$K(t) = \log \mathbb{E}_{\pi_{\mathrm{ref}}}[e^{tX}] = \log \sum_{y \in \mathcal{Y}} \pi_{\mathrm{ref}}(y) r(y)^t.$$

Then, we have the following:

$$\log \frac{Z_{\mathcal{D}'}}{Z_{\mathcal{D}}} = K(\frac{\alpha}{\beta}) - K(\frac{1}{\beta}), \quad \mathbb{D}_{\mathrm{KL}}\left[\pi_{\mathrm{ref}} \| \pi_\ell\right] = \mathbb{E}_{\pi_{\mathrm{ref}}}[X] = K'(0),$$

where $K'(t) = \frac{d}{dt} K(t)$.

Therefore, we obtain the following expression:

$$\mathbb{D}_{\mathrm{KL}}\left[\pi_{\mathrm{ref}} \| \pi_{\mathcal{D}'}^*\right] - \mathbb{D}_{\mathrm{KL}}\left[\pi_{\mathrm{ref}} \| \pi_{\mathcal{D}}^*\right] = K(\frac{\alpha}{\beta}) - K(\frac{1}{\beta}) + \frac{1-\alpha}{\beta} K'(0).$$

Since the cumulant-generating function $K(t)$ is convex and twice differentiable, we have

$$K(\frac{\alpha}{\beta}) \geq K(\frac{1}{\beta}) + \left(\frac{\alpha}{\beta} - \frac{1}{\beta}\right) K'(\frac{1}{\beta})$$

$$\iff K(\frac{\alpha}{\beta}) - K(\frac{1}{\beta}) \geq \left(\frac{\alpha}{\beta} - \frac{1}{\beta}\right) K'(\frac{1}{\beta}).$$

Meanwhile, since $K'(t)$ is non-decreasing due to convexity, we have $K'(\frac{1}{\beta}) > K'(0)$. Therefore, we arrive at the final relationship:

$$\mathbb{D}_{\mathrm{KL}}\left[\pi_{\mathrm{ref}} \| \pi_{\mathcal{D}'}^*\right] - \mathbb{D}_{\mathrm{KL}}\left[\pi_{\mathrm{ref}} \| \pi_{\mathcal{D}}^*\right] \geq \frac{\alpha-1}{\beta}\left(K'(\frac{1}{\beta}) - K'(0)\right) > 0,$$

where the strict inequality comes from $\pi_{\mathrm{ref}} \neq \pi_\ell$. $\qquad\square$

# K  DPO-PROJECTED GRADIENT (DPO-PG)

While several variants of DPO have been proposed to address log-likelihood displacement (Pal et al., 2024; Xiao et al., 2024a), we observed that these methods exhibit instability when scaled to large datasets (approximately 100,000 samples) and trained over multiple epochs (*e.g.*, 5 epochs in our experiments of Section 5.2). A proper alternative that prevents LLD should increase $\log \pi(y_w)$ while reducing the DPO loss to a comparable extent. Without achieving a comparable reduction in the DPO loss, it becomes difficult to argue that this method has properly learned the underlying preference distribution.

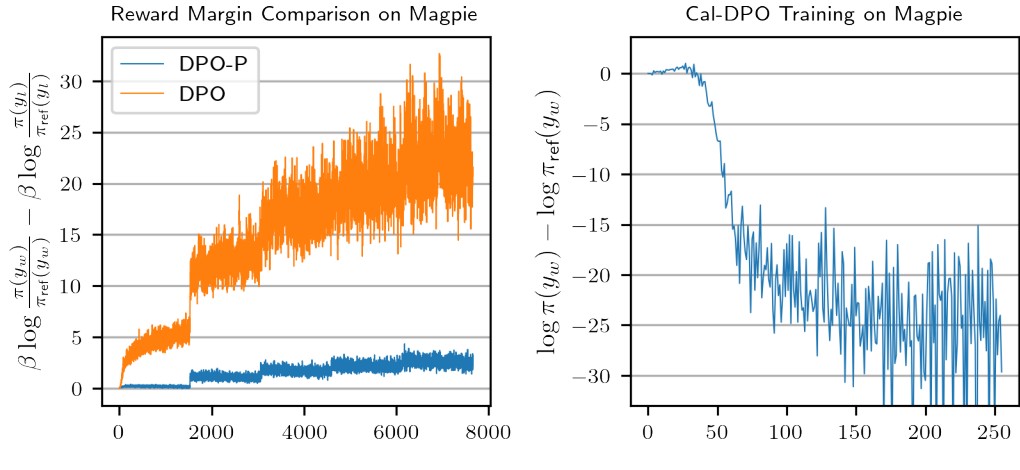

Figure 8: Testing DPOP (Pal et al., 2024) and Cal-DPO (Xiao et al., 2024a) on Magpie dataset. We found that DPOP fails to optimize the log-margin as effectively as vanilla DPO. Meanwhile, we found that Cal-DPO is unstable at preventing log-likelihood displacement.

Despite extensive experiments with various hyper-parameters, we failed to find a setting for both DPOP (Pal et al., 2024) and Cal-DPO (Xiao et al., 2024a) which met this criterion reliably (Figure 8). This motivated us to design a new method that reliably prevents log-likelihood displacement while ensuring optimization of the DPO loss. The result is DPO-PG, a method grounded in projected gradient descent.

As its name implies, DPO-Projected Gradient (DPO-PG) leverages projected gradient descent (Boyd & Vandenberghe, 2014) to reinforce the policy distribution while optimizing the DPO objective. Specifically, it increases $\log \pi(y_w)$ while maintaining or decreasing $\log \pi(y_\ell)$. Due to the log-margin term in the DPO loss, DPO-PG is guaranteed to reduce the DPO loss under sufficiently small step sizes (Corollary K.5.1).

The primary advantage of using DPO-PG over other DPO variants (*e.g.*, DPOP (Pal et al., 2024), Cal-DPO (Xiao et al., 2024a)) is that DPO-PG can reliably optimize the DPO loss while increasing both $\log \pi(y_w)$ and the log-margin $\log \pi(y_w) - \log \pi(y_\ell)$, all without introducing any additional hyper-parameters. We empirically confirm that DPO-PG prevents LLD from Figure 12, and that it also optimizes the DPO loss to a comparable extent in Figure 13.

**Definition K.1.** DPO-Projected Gradient (DPO-PG): $\theta_{k+1} = \theta_k - \eta(\nabla L(y_w) - \frac{\alpha}{||\nabla L(y_\ell)||_2^2}\nabla L(y_\ell))$, where $\theta_k$ denotes the parameter at training step $k$, $\eta > 0$ is the step size, and $\alpha = \max(0, \nabla L(y_w) \cdot \nabla L(y_\ell))$.

Here, $L(y)$ is the **negative** log-likelihood loss: $-\frac{1}{M}\sum_{i=1}^{M}\log \pi(y^{(i)})$, where $M$ is the batch size, and $y^{(i)}$ is the $i$-th element in the batch. In practice, when using any non-SGD optimizer (*e.g.*, Adam (Kingma & Ba, 2015), RMSprop (Tieleman & Hinton, 2012)), we set the parameters' gradient as $\nabla L(y_w) - \frac{\alpha}{||\nabla L(y_\ell)||_2^2}\nabla L(y_\ell)$ and update its parameters following the optimizer's algorithm. For gradient-clipping, we clip the L2 norm of $\nabla L(y_w) - \frac{\alpha}{||\nabla L(y_\ell)||_2^2}\nabla L(y_\ell)$.

We now show that DPO-PG decreases $L(y_w)$ while maintaining or increasing $L(y_\ell)$, for sufficiently small step sizes. We begin with the definition of *descent direction* (Boyd & Vandenberghe, 2014):

**Definition K.2.** For some function $f : \mathbb{R}^D \to \mathbb{R}$, and a point $\theta \in \mathbb{R}^D$, a direction $\Delta\theta \in \mathbb{R}^D$ is called a *descent direction* if there exists $\bar{\alpha} > 0$ such that $f(\theta + \alpha\Delta\theta) < f(\theta), \forall \alpha \in (0, \bar{\alpha})$.

The following well-known lemma allows one to verify whether a direction is a descent direction of some differentiable objective function $f$ (Boyd & Vandenberghe, 2014).

**Lemma K.3.** *Consider a point $\theta \in \mathbb{R}^D$. Any direction $\Delta\theta \in \mathbb{R}^D$ satisfying $\Delta\theta \cdot \nabla f(\theta) < 0$ is a descent direction.*

We now analyze the properties of the update direction of DPO-PG: $\Delta\theta = \theta_{k+1} - \theta_k = -\eta\{\nabla L(y_w) - \frac{\max(0, \nabla L(y_w) \cdot \nabla L(y_\ell))}{||\nabla L(y_\ell)||_2^2}\nabla L(y_\ell)\}$. The following theorem states that DPO-PG increases the log-likelihood of $y_w$.

**Theorem K.4.** $\Delta\theta$ *is a descent direction of the negative log-likelihood of the chosen responses* $-\frac{1}{M}\Sigma_{i=1}^M \log \pi(y_w^{(i)}) = L(y_w)$.

*Proof.* Regardless of the sign value of $\nabla L(y_w) \cdot \nabla L(y_\ell)$, we can show that $\Delta\theta \cdot \nabla L(y_w) < 0$.

**Case 1:** If we have $\nabla L(y_w) \cdot \nabla L(y_\ell) > 0$, it follows that

$$\Delta\theta \cdot \nabla L(y_w) = -\eta\{||\nabla L(y_w)||_2^2 - \frac{\nabla L(y_w) \cdot \nabla L(y_\ell)}{||\nabla L(y_\ell)||_2^2}\nabla L(y_\ell) \cdot \nabla L(y_w)\}$$

$$= -\frac{\eta}{||\nabla L(y_\ell)||_2^2}\{||\nabla L(y_w)||_2^2 \cdot ||\nabla L(y_\ell)||_2^2 - (\nabla L(y_\ell) \cdot \nabla L(y_w))^2\} < 0,$$

where the last inequality follows from the Cauchy-Schwarz inequality: $||\nabla L(y_w)||_2^2 \cdot ||\nabla L(y_\ell)||_2^2 > ||\nabla L(y_w) \cdot \nabla L(y_\ell)||_2^2 > 0$.

**Case 2:** Otherwise, we have $\nabla L(y_w) \cdot \nabla L(y_\ell) \leq 0$ and it follows that $\Delta\theta \cdot \nabla L(y_w) = -\eta||\nabla L(y_w)||_2^2 < 0$. $\qquad\square$

Conversely, we can show that DPO-PG decreases or maintains the log-likelihood of $y_\ell$.

**Theorem K.5.** $\Delta\theta$ *is **not** a descent direction of the negative log-likelihood of the rejected responses* $-\frac{1}{M}\Sigma_{i=1}^M \log \pi(y_\ell^{(i)}) = L(y_\ell)$.

*Proof.* We have $\Delta\theta \cdot \nabla L(y_\ell) = -\eta\{\nabla L(y_w) \cdot \nabla L(y_\ell) - \max(0, \nabla L(y_w) \cdot \nabla L(y_\ell))\} \geq 0$. In other words, $\Delta\theta$ is either orthogonal or an ascent direction to the negative log-likelihood of the rejected responses $y_\ell$. $\qquad\square$

Meanwhile, various offline preference optimization methods can be characterized as solving the following objective (Tang et al., 2024):

$$\arg\min_\theta \mathbb{E}_{(y_w, y_\ell) \in \mathcal{D}} \left[ f(\beta \log \frac{\pi_\theta(y_w)}{\pi_{\text{ref}}(y_w)} - \beta \log \frac{\pi_\theta(y_\ell)}{\pi_{\text{ref}}(y_\ell)}) \right],$$

where $f$ denotes any valid supervised binary classification loss function (Hastie, 2009). As a consequence of Theorems K.4 and K.5, DPO-PG is able to optimize a wide variety of preference optimization objectives including DPO (Tang et al., 2024).

**Corollary K.5.1.** *For any valid supervised binary classification loss function $f$ with $f'(\cdot) < 0$, $\Delta\theta$ is a descent direction to the loss $f(\beta \cdot (\log \frac{\pi_\theta(y_w)}{\pi_{\text{ref}}(y_w)} - \log \frac{\pi_\theta(y_\ell)}{\pi_{\text{ref}}(y_\ell)}))$ where $\beta > 0$.*

*Proof.*

$$\Delta\theta \cdot \nabla f(\beta \log \frac{\pi_\theta(y_w)}{\pi_{\text{ref}}(y_w)} - \beta \log \frac{\pi_\theta(y_\ell)}{\pi_{\text{ref}}(y_\ell)})$$

$$= \Delta\theta \cdot \beta f'\left(\beta \log \frac{\pi_\theta(y_w)}{\pi_{\text{ref}}(y_w)} - \beta \log \frac{\pi_\theta(y_\ell)}{\pi_{\text{ref}}(y_\ell)}\right)(\nabla L(y_w) - \nabla L(y_\ell))$$

$$= \beta \underbrace{f'\left(\beta \log \frac{\pi_\theta(y_w)}{\pi_{\text{ref}}(y_w)} - \beta \log \frac{\pi_\theta(y_\ell)}{\pi_{\text{ref}}(y_\ell)}\right)}_{f'(\cdot)<0}(\underbrace{\Delta\theta \cdot \nabla L(y_w)}_{>0} - \underbrace{\Delta\theta \cdot \nabla L(y_\ell)}_{\leq 0}).$$

From Lemma K.4, we have $\Delta\theta \cdot \nabla L(y_w) > 0$, and from Lemma K.5, we have $\Delta\theta \cdot \nabla L(y_\ell) \leq 0$. Thus, we have $(\Delta\theta \cdot \nabla L(y_w) - \Delta\theta \cdot \nabla L(y_\ell)) > 0$. Since $\beta > 0$ and $\beta f'(\cdot) < 0$, it follows that $\Delta\theta \cdot \nabla f(\beta \log \frac{\pi_\theta(y_w)}{\pi_{\text{ref}}(y_w)} - \beta \log \frac{\pi_\theta(y_\ell)}{\pi_{\text{ref}}(y_\ell)}) < 0$. $\qquad\square$

To summarize, Lemma K.4 ensures that only $\log \pi(y_w)$ (and not $\log \pi(y_\ell)$) increases during training, for sufficiently small step sizes. This ensures policy reinforcement with respect to $\pi_{\text{ref}}$. Corollary K.5.1 further ensures that DPO-PG optimizes the DPO loss, too. We empirically validate that DPO prevents LLD in Figure 12, and also confirm that DPO-PG successfully optimizes the DPO loss in Figure 13.

## L    ADDITIONAL EXPERIMENTAL RESULTS

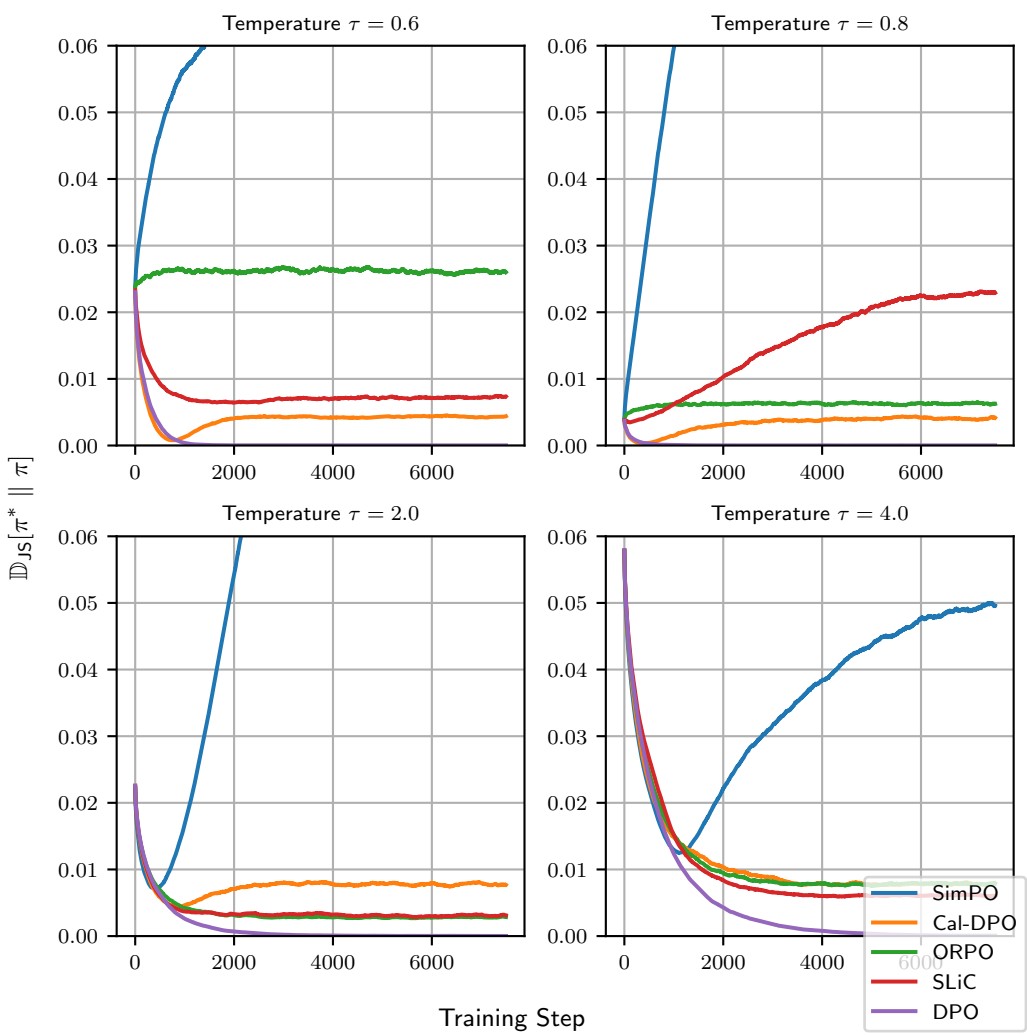

Figure 9: Validation of Theorem 3.2: Comparison of the Jensen-Shannon Divergence $\mathbb{D}_{\mathrm{JS}}[\pi^*\|\pi]$ during training using different objectives on the synthetic dataset of Section 3.3. Standard DPO ($r = \log(\pi/\pi_{\mathrm{ref}})$, purple) consistently minimizes the divergence to the target policy $\pi^*$. This demonstrates its optimality when preferences encode the Differential Information required to update the reference policy $\pi_{\mathrm{ref}}$ into the target policy $\pi^*$. The non-convergence of SimPO (Meng et al., 2024) to $\pi^*$ also follows from Theorem 3.2, which states that the DPO log-ratio reward *uniquely* recovers $\pi^*$ when preferences encode Differential Information (Theorem 3.1). We hypothesize that SimPO's trajectory exhibits a close "fly-by" of $\pi^*$ before settling in a different region, though a principled explanation for this specific trajectory remains open.

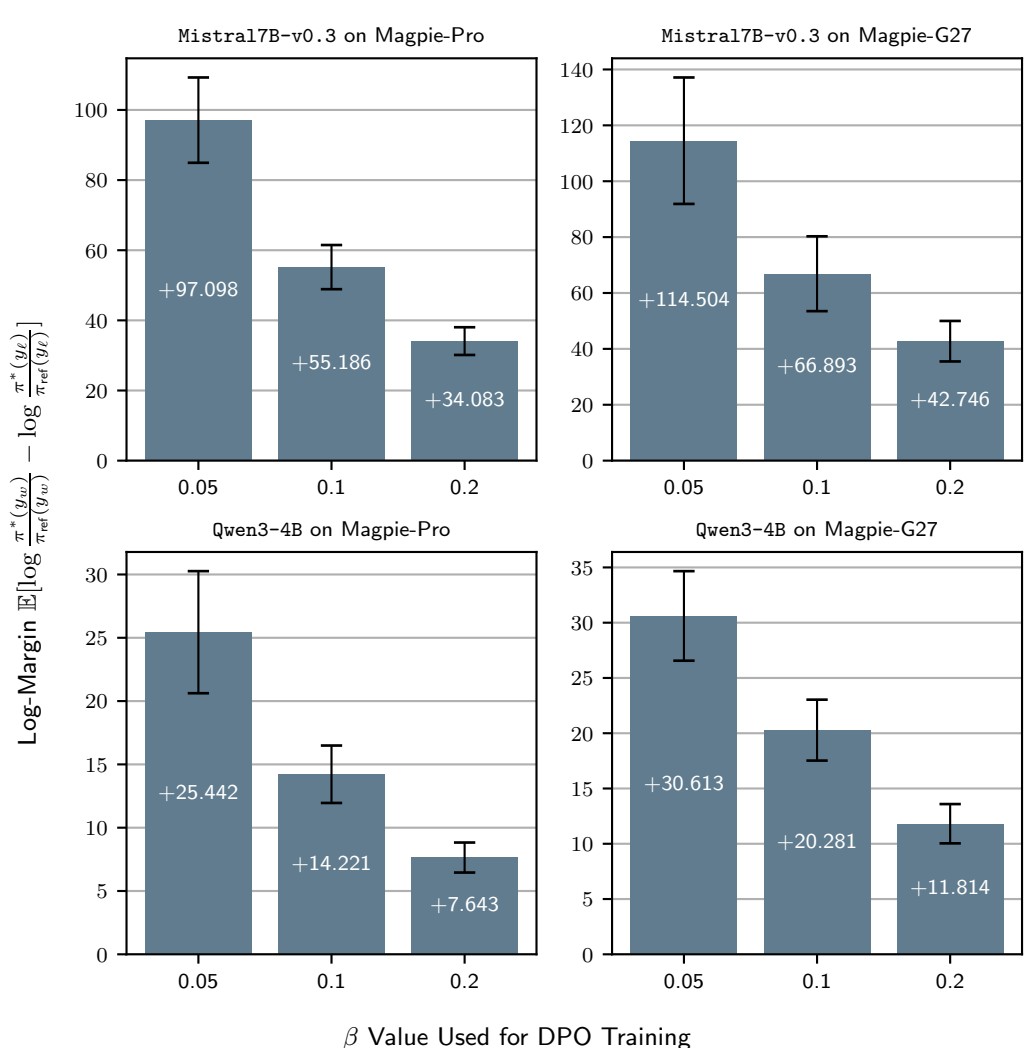

Figure 10: Log-margin [nats] of DPO-converged policies trained on real data, with error bars denoting $\pm 1$ standard deviation. The resulting values vary substantially with $\beta$, exhibiting statistically significant differences. In particular, the log-margin approximately scales proportionally to $1/\beta$, consistent with the DID power-law (Corollary 3.2.1).

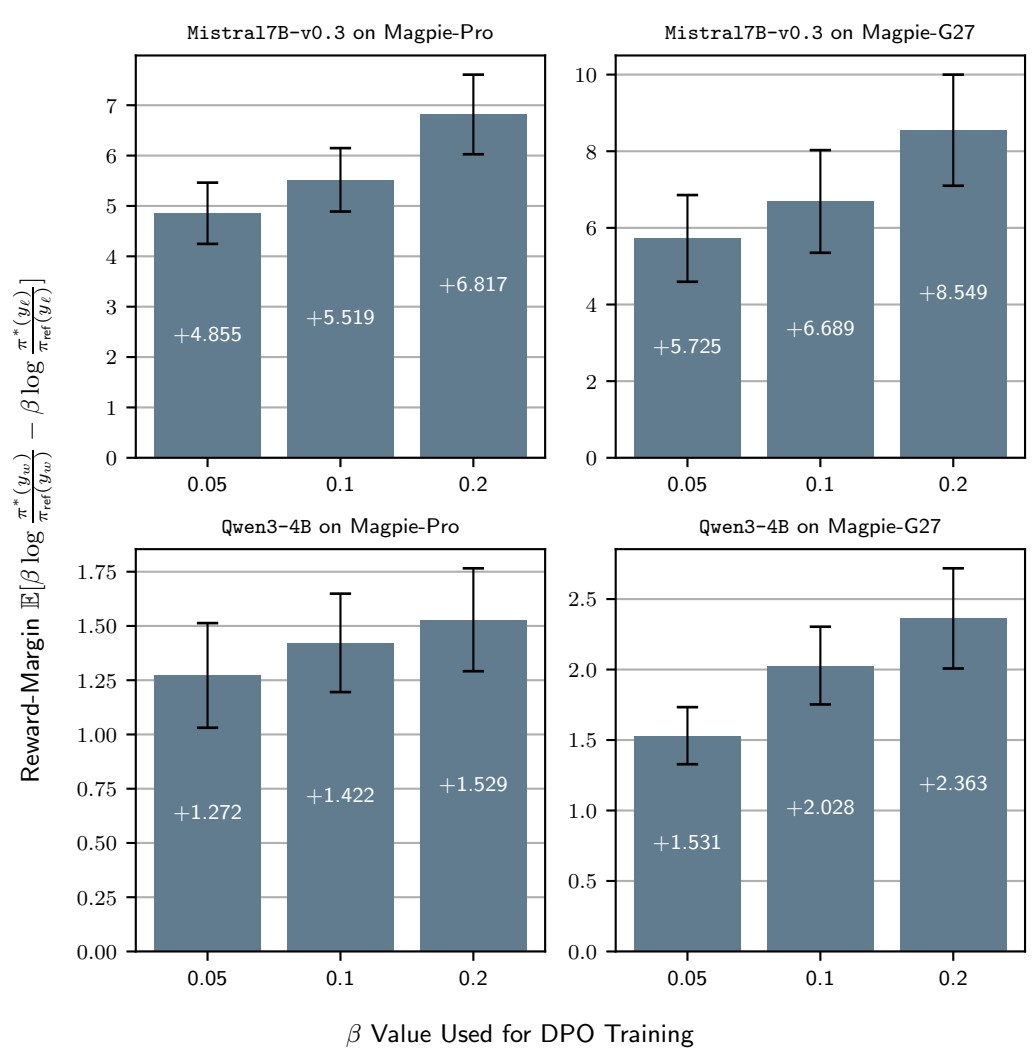

Figure 11: Reward-margin [nats] of DPO-converged policies trained on real data, with error bars denoting $\pm 1$ standard deviation. In contrast to the log-margin patterns from Figure 10, these values remain within a stable range across all $\beta \in \{0.05, 0.1, 0.2\}$. Although not perfectly identical due to finite data and imperfect optimization, the reward-margins exhibit strikingly low variation compared to the fluctuations observed in Figure 10, aligning with Corollary 3.2.1. Taken together with the log-margin trends, this $\beta$-invariant reward-margin behavior directly supports the DID power-law in real DPO training.

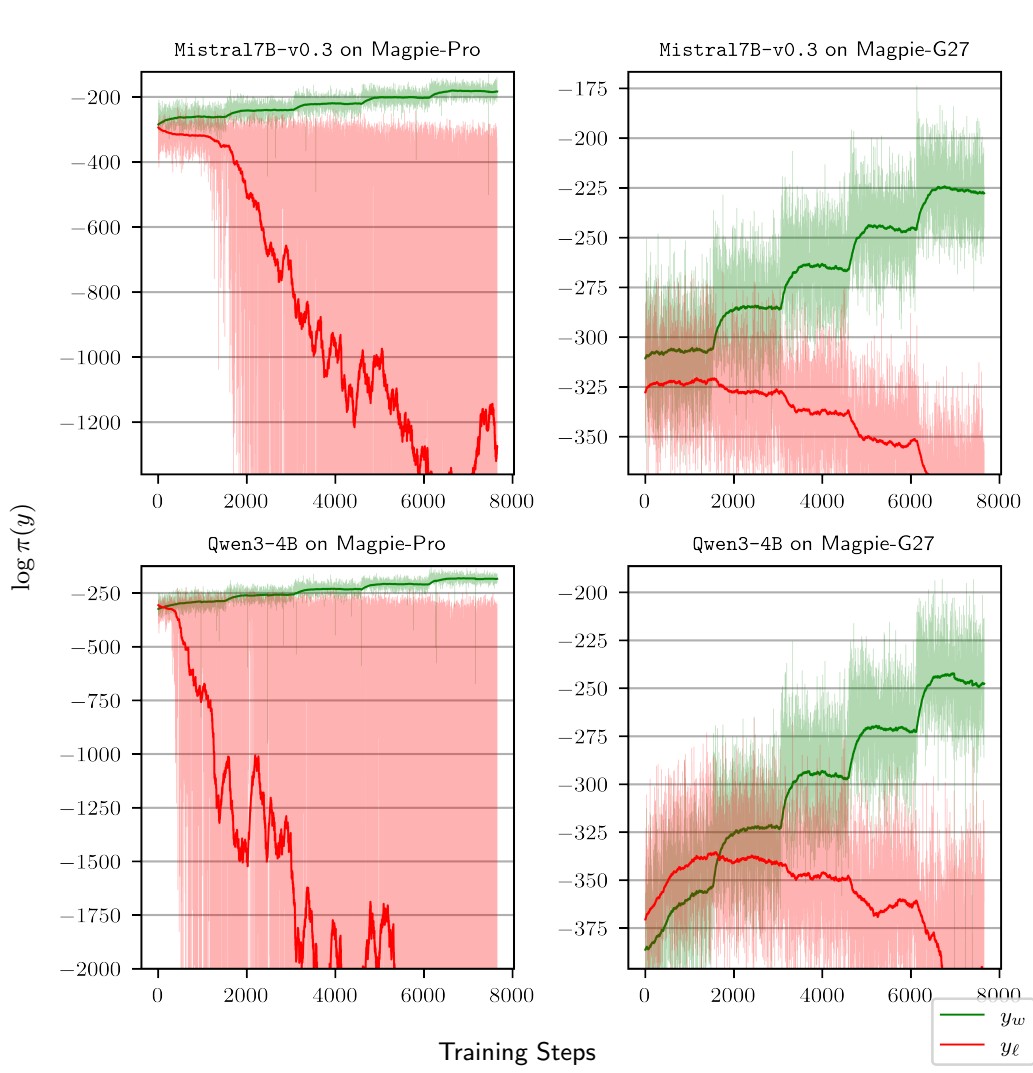

Figure 12: Log-likelihood change for DPO-PG across all experimental configurations. Overall, DPO-PG consistently increases the log-likelihood of $y_w$, while decreasing or maintaining the log-likelihood of $y_\ell$.

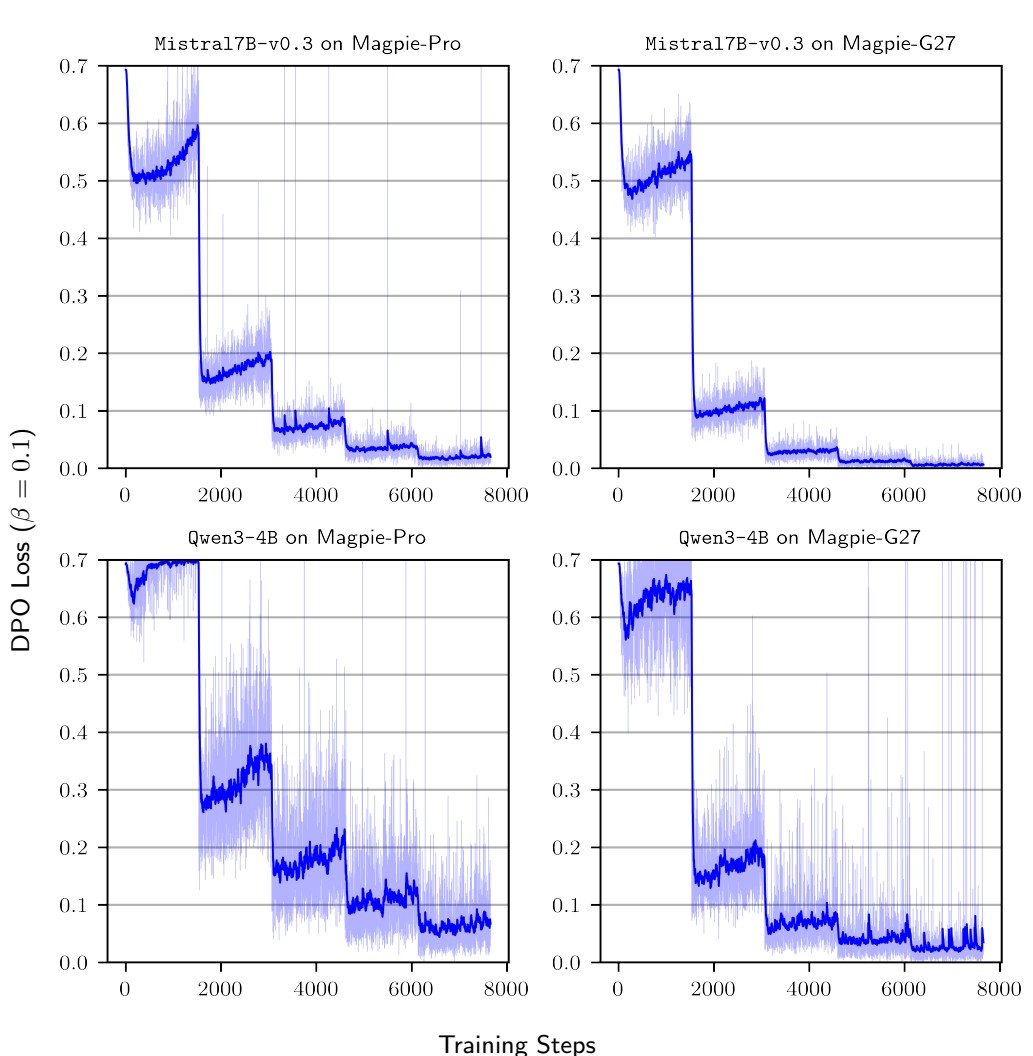

Figure 13: DPO loss for DPO-PG across all experimental configurations. The DPO loss is computed using $\beta = 0.1$. DPO-PG is able to optimize the DPO loss regardless of the model architecture or dataset, validating Corollary K.5.1. In conjunction with Figure 12, DPO-PG is able to prevent log-likelihood displacement while still optimizing the DPO objective.

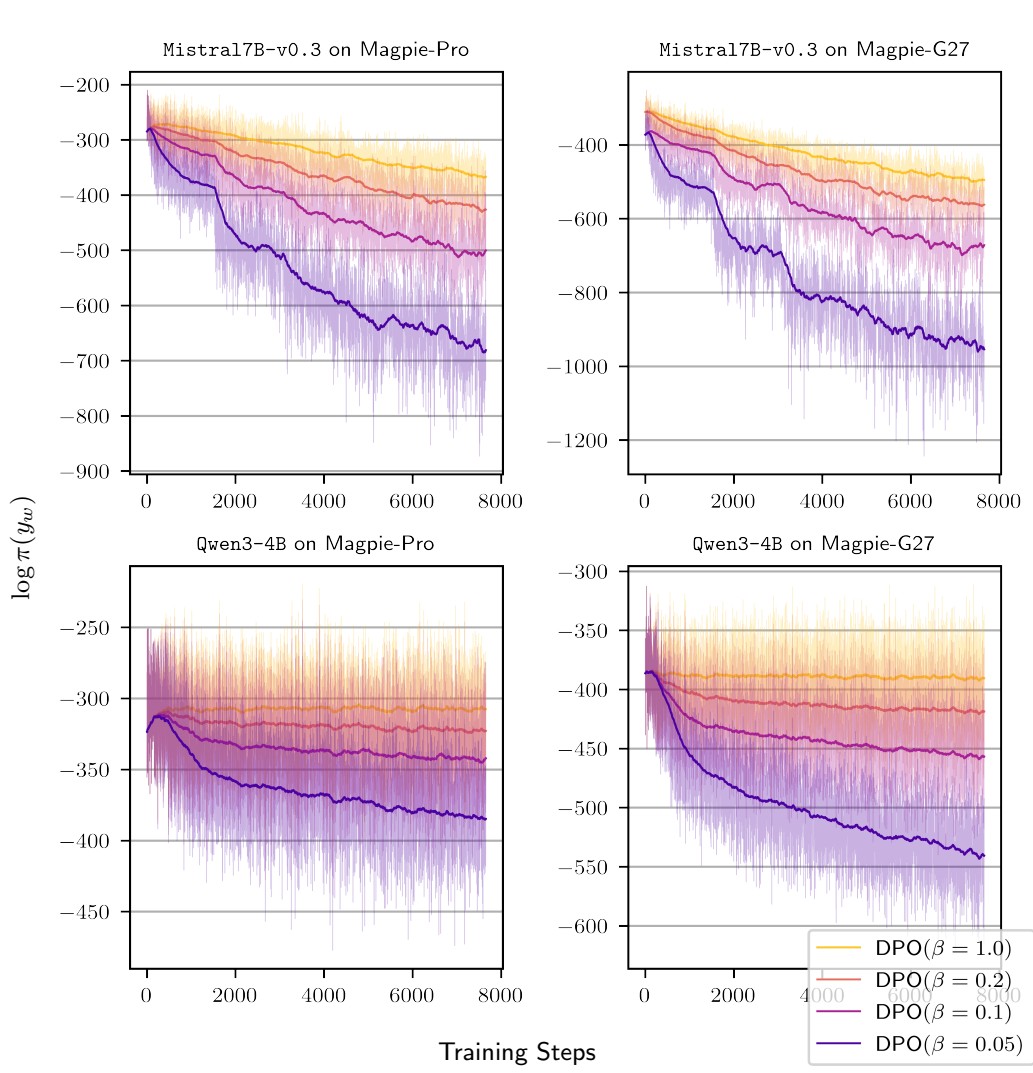

Figure 14: Log-likelihood change of $y_w$ for DPO across all experimental configurations. The log-likelihood of chosen responses decreases throughout the training process, indicating log-likelihood displacement.

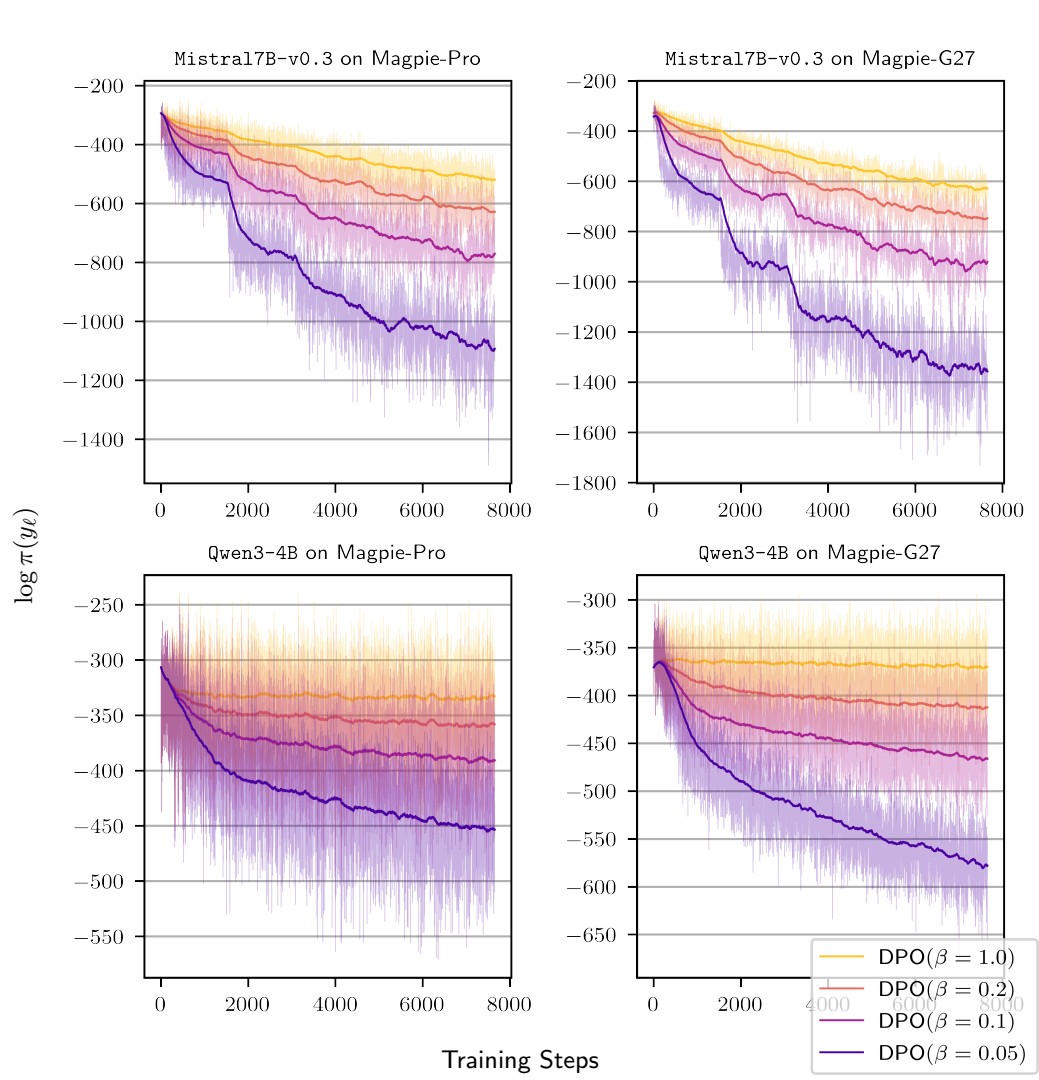

Figure 15: Log-likelihood change of $y_\ell$ for DPO across all experimental configurations.

Table 4: Evaluation results for open-ended instruction-following. We report the win-rate [%] for Arena-Hard-v0.1 and ELO score for Wild-Bench-v2, with the 95% confidence interval. We also specify the selected best epoch following the procedure in Appendix M.2, and highlight the model with the best Arena-Hard win-rate in **bold**. The DID entropy ($H(q_{\pi^*/\pi_{\mathrm{ref}}})$, [nats]) is estimated by importance sampling (Appendix D.5). Standard DPO, which exhibits LLD and learns a high-entropy DID, outperforms DPO-PG, which prevents LLD and learns a lower-entropy DID. This suggests that knowledge required for such open-ended tasks is associated with high-entropy DID.

(a) `Mistral7B-v0.3` trained on Magpie-Pro

| Method | Best Epoch | $H(q_{\pi^*/\pi_{\mathrm{ref}}})$ | Arena-Hard-v0.1 | Wild-Bench-v2 |
|---|---|---|---|---|
| DPO $\beta = 1.0$ | 3 | 1123.23 | 19.1 (-1.4, 2.0) | 1141.47 (-10.17, 10.36) |
| DPO $\beta = 0.2$ | 4 | 1303.44 | 18.5 (-1.4, 1.6) | 1145.34 (-9.79, 11.10) |
| DPO $\beta = 0.1$ | 1 | **1253.89** | **23.4 (-1.9, 2.0)** | **1146.63 (-11.99, 9.52)** |
| DPO $\beta = 0.05$ | 1 | 970.21 | 22.4 (-1.7, 1.9) | 1145.32 (-14.83, 12.52) |
| DPO-PG | 5 | 495.12 | 19.6 (-1.9, 1.6) | 1129.92 (-12.92, 11.83) |

(b) `Mistral7B-v0.3` trained on Magpie-G27

| Method | Best Epoch | $H(q_{\pi^*/\pi_{\mathrm{ref}}})$ | Arena-Hard-v0.1 | Wild-Bench-v2 |
|---|---|---|---|---|
| DPO $\beta = 1.0$ | 4 | **836.56** | **30.4 (-2.2, 2.0)** | **1140.56 (-12.72, 12.72)** |
| DPO $\beta = 0.2$ | 2 | 576.85 | 30.0 (-2.6, 1.9) | 1145.25 (-13.79, 13.36) |
| DPO $\beta = 0.1$ | 2 | 509.92 | 27.7 (-2.0, 2.4) | 1146.87 (-13.20, 10.76) |
| DPO $\beta = 0.05$ | 1 | 694.96 | 27.0 (-2.3, 2.0) | 1149.51 (-14.38, 14.72) |
| DPO-PG | 5 | 378.07 | 24.0 (-1.9, 1.4) | 1130.62 (-15.76, 15.07) |

(c) `Qwen3-4B` trained on Magpie-Pro

| Method | Best Epoch | $H(q_{\pi^*/\pi_{\mathrm{ref}}})$ | Arena-Hard-v0.1 | Wild-Bench-v2 |
|---|---|---|---|---|
| DPO $\beta = 1.0$ | 2 | 9158.43 | 30.7 (-1.4, 2.0) | 1134.39 (-15.73, 12.99) |
| DPO $\beta = 0.2$ | 3 | 4765.94 | 28.1 (-2.2, 2.1) | 1146.17 (-11.92, 11.28) |
| DPO $\beta = 0.1$ | 4 | 7663.28 | 27.5 (-1.9, 1.9) | 1148.22 (-8.99, 9.43) |
| DPO $\beta = 0.05$ | 4 | 6801.18 | **43.7 (-2.9, 2.2)** | **1164.49 (-9.68, 13.13)** |
| DPO-PG | 5 | 388.24 | 37.4 (-2.2, 2.4) | 1148.50 (-15.64, 11.74) |

(d) `Qwen3-4B` trained on Magpie-G27

| Method | Best Epoch | $H(q_{\pi^*/\pi_{\mathrm{ref}}})$ | Arena-Hard-v0.1 | Wild-Bench-v2 |
|---|---|---|---|---|
| DPO $\beta = 1.0$ | 2 | 6606.27 | 43.1 (-2.9, 2.5) | 1157.95 (-14.05, 16.92) |
| DPO $\beta = 0.2$ | 5 | 14744.58 | 48.8 (-2.5, 2.8) | 1165.78 (-11.16, 11.72) |
| DPO $\beta = 0.1$ | 3 | 3048.57 | 53.1 (-2.5, 2.3) | 1173.52 (-15.24, 13.22) |
| DPO $\beta = 0.05$ | 4 | 11705.65 | **54.0 (-2.7, 2.4)** | **1177.44 (-12.57, 13.84)** |
| DPO-PG | 4 | 400.75 | 40.0 (-2.8, 2.1) | 1160.97 (-11.34, 12.47) |

Table 5: 95% confidence interval (CI) for the estimated DID entropy ($H(q_{\pi^*/\pi_{\mathrm{ref}}})$, [nats]) values from Table 4. We compute the 95% bootstrap confidence intervals with 10,000 resamples. Overall, the DID entropy estimates are stable and the differences across methods are statistically significant.

(a) `Mistral7B-v0.3` trained on Magpie-Pro

| Method | $H(q_{\pi^*/\pi_{\mathrm{ref}}})$ | 95% Confidence Interval |
|---|---|---|
| DPO $\beta = 1.0$ | 1123.23 | (1065.74, 1186.46) |
| DPO $\beta = 0.2$ | 1303.44 | (1227.55, 1381.41) |
| DPO $\beta = 0.1$ | 1253.89 | (1172.91, 1337.54) |
| DPO $\beta = 0.05$ | 970.21 | (910.44, 1030.44) |
| DPO-PG | 495.12 | (477.12, 513.06) |

(b) `Mistral7B-v0.3` trained on Magpie-G27

| Method | $H(q_{\pi^*/\pi_{\mathrm{ref}}})$ | 95% Confidence Interval |
|---|---|---|
| DPO $\beta = 1.0$ | 836.56 | (818.00, 855.87) |
| DPO $\beta = 0.2$ | 576.85 | (563.49, 590.49) |
| DPO $\beta = 0.1$ | 509.92 | (495.02, 525.25) |
| DPO $\beta = 0.05$ | 694.96 | (664.22, 732.05) |
| DPO-PG | 378.07 | (367.27, 389.20) |

(c) `Qwen3-4B` trained on Magpie-Pro

| Method | $H(q_{\pi^*/\pi_{\mathrm{ref}}})$ | 95% Confidence Interval |
|---|---|---|
| DPO $\beta = 1.0$ | 9158.43 | (8753.45, 9577.95) |
| DPO $\beta = 0.2$ | 4765.94 | (4538.30, 5004.31) |
| DPO $\beta = 0.1$ | 7663.28 | (7515.45, 7810.67) |
| DPO $\beta = 0.05$ | 6801.18 | (6684.91, 6921.41) |
| DPO-PG | 388.24 | (374.38, 402.53) |

(d) `Qwen3-4B` trained on Magpie-G27

| Method | $H(q_{\pi^*/\pi_{\mathrm{ref}}})$ | 95% Confidence Interval |
|---|---|---|
| DPO $\beta = 1.0$ | 6606.27 | (6239.81, 6983.42) |
| DPO $\beta = 0.2$ | 14744.58 | (14546.10, 14948.17) |
| DPO $\beta = 0.1$ | 3048.57 | (2862.17, 3239.40) |
| DPO $\beta = 0.05$ | 11705.65 | (11468.98, 11946.71) |
| DPO-PG | 400.75 | (387.85, 413.72) |

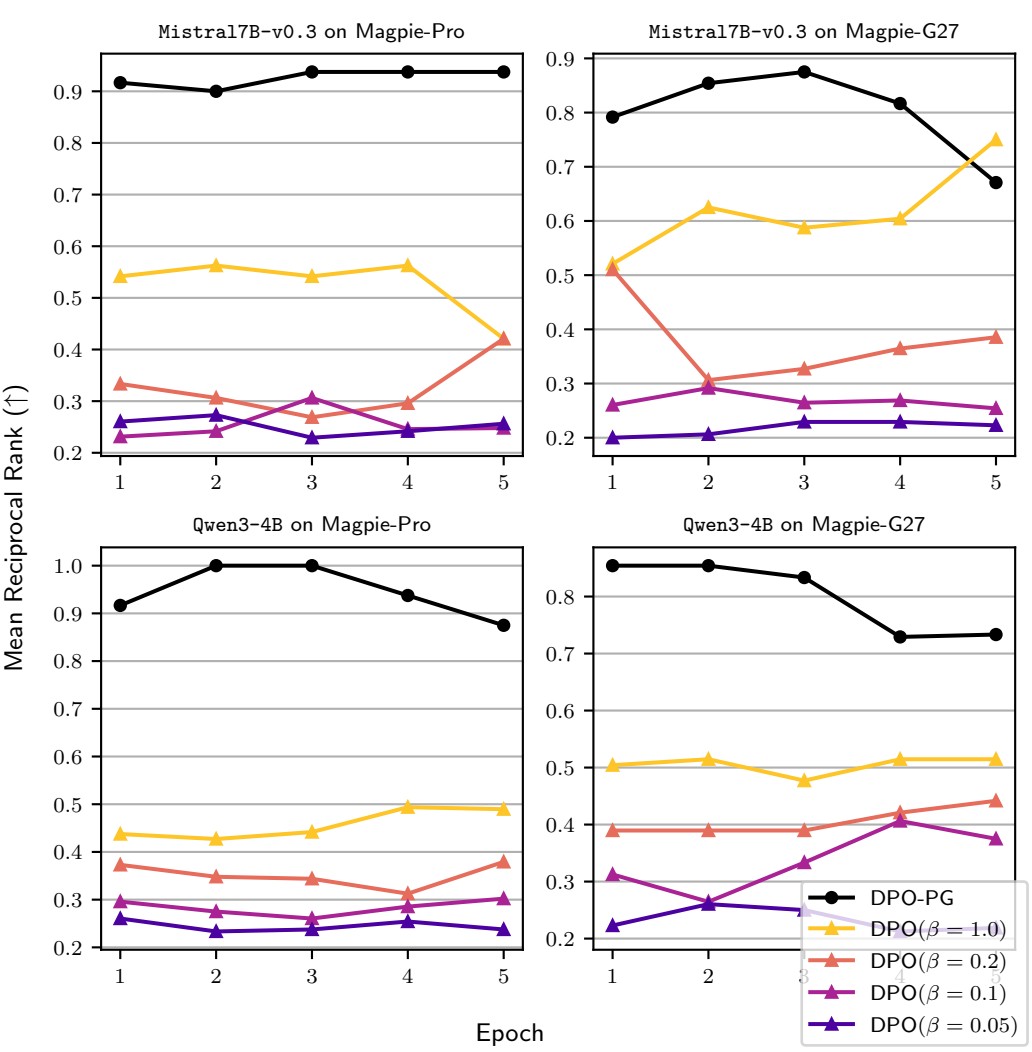

Figure 16: Mean reciprocal rank (MRR) across 8 knowledge-intensive QA benchmarks during training. The MRR is computed following the procedure in Appendix M.2. Preventing LLD (DPO-PG) outperforms standard DPO which exhibits LLD. As DPO-PG learns a low-entropy DID compared to standard DPO (Table 4), this suggests that the knowledge for factual QA is mainly associated with low-entropy DID.

# M EXPERIMENTAL SETUP

## M.1 CONTROLLED SETTING

We conduct controlled experiments involving Energy Based Models (EBMs) in a free-tier Google Colaboratory[8] CPU environment, using PyTorch (Paszke et al., 2019). We use `torch.float32` as the default data type. We set the total class size as 32, and use a batch size of 512 and fix the training seed to 42 for reproducibility. We utilize the RMSprop (Tieleman & Hinton, 2012) optimizer with gradient clipping at maximum norm of 1.0. We use a constant learning rate of 0.001.

**Figures 1 and 9.** For fair comparison, we follow Meng et al. (2024) in extensively searching the hyper-parameters for the following baseline methods:

- SLiC (Zhao et al., 2023b): $\beta \in \{0.1, 0.5, 1.0, 2.0\}, \lambda \in \{0.1, 0.5, 1.0, 10.0\}$.
- ORPO (Hong et al., 2024): $\beta \in \{0.1, 0.5, 1.0, 2.0\}$.
- SimPO (Meng et al., 2024): $\beta \in \{2.0, 2.5\}, \gamma \in \{0.3, 0.5, 1.0, 1.2, 1.4, 1.6\}$.
- Cal-DPO (Xiao et al., 2024a): $\beta \in \{0.001, 0.002, 0.003, 0.01, 0.1\}$.

The best hyper-parameter is chosen based on the minimum value of $\mathbb{D}_{JS}[\pi^* \| \pi]$ achieved through-out the training process.

For Figure 1 (left) we train for a total of 10,000 steps. For Figure 1 (right) and 9, we train for a total of 7,500 steps due to the large number of training configurations as listed above.

**Figure 2.** We train for a total of 7,500 training steps. The left plot tests $\beta \in \{0.05, 0.1, 0.5, 1, 2, 5\}$, and the right plot tests $\beta \in \{1, 2, 4, 8, 16, 32\}$ following the $\beta$ conditions in Theorem 4.1.

**Figure 4.** We train for a total of 20,000 training steps. For the baseline dataset $\mathcal{D}$ ($\alpha = 1$), we test $\beta \in \{0.25, 0.5, 1, 2, 4\}$, and for the strengthened dataset $\mathcal{D}'$ ($\alpha = 2$), we test $\beta \in \{0.5, 1, 2, 4, 8\}$.

**Figure 5.** We train for a total of 5,000 training steps, averaging over five training seeds: [42, 43, 44, 45, 46]. We measure the converged JS-divergence by averaging the $\mathbb{D}_{JS}[\pi^* \| \pi]$ of the last 50 training steps.

## M.2 REAL-WORLD SETTING

**Magpie-G27 dataset.** Magpie-G27 is an instruction-following preference dataset built from the prompts of Magpie-Air[9] and completed with responses generated by a stronger model (`google/gemma-2-27b-it`) (Team, 2024). Prompts in Magpie-G27 are disjoint from those in the Magpie-Pro dataset[10]. For each prompt, we sample five completions via vLLM (Kwon et al., 2023) using the following sampling configuration:

```
{n=5, temperature=0.9, top_p=1, max_tokens=4096, seed=42}.
```

We then score these completions with a strong off-the-shelf reward model `Skywork/Skywork-Reward-Gemma-2-27B-v0.2` (Liu et al., 2024a) and select the highest- and lowest-scoring responses as $y_w$ and $y_\ell$, respectively.

**Training setup.** To isolate the impact of alignment methods, we use pre-trained base models (*i.e.*, not instruction-tuned) paired with the official chat templates of their instruction-tuned counterparts. Specifically, we utilize the chat-template of `mistralai/Mistral-7B-Instruct-v0.3` for `Mistral7B-v0.3`, and the chat-template of `Qwen/Qwen3-4B-Instruct-2507` for `Qwen3-4B` with its thinking-tags removed.

---

[8] https://colab.google/

[9] https://huggingface.co/datasets/Magpie-Align/Magpie-Air-DPO-100K-v0.1

[10] https://huggingface.co/datasets/Magpie-Align/Magpie-Llama-3.1-Pro-DPO-100K-v0.1

**Reference policy setup.** To prepare the reference policy, we fine-tune the base model on the chosen responses, following standard practice (Rafailov et al., 2023; 2024). We train for one epoch with an effective batch size of 256, using the Adam optimizer (Kingma & Ba, 2015) (default $\beta_0, \beta_1$; weight decay $= 0$). Training proceeds with a constant learning-rate of $5 \times 10^{-6}$ and a linear warm-up over the first 10% of steps. The objective is standard cross-entropy loss applied to the full token sequence (including prompts and chat-template tokens). We fix the random seed to 0.

**Preference optimization.** During the alignment phase, we train for five epochs with an effective batch size of 64 using RMSprop (Tieleman & Hinton, 2012) with no weight decay. We adopt a constant learning rate of $1 \times 10^{-6}$, with 150-step linear warm-up and compute loss only over generated completions. For `Qwen3-4B` under DPO-PG, we increase the learning rate to $1 \times 10^{-5}$, as this setting leads to more effective optimization, preventing LLD while optimizing the DPO loss (Appendix K). We fix the random seed to 1. Models checkpoints are saved after each epoch and trained in `bfloat16` precision. For DPO trained models, we test $\beta \in \{0.05, 0.1, 0.2, 1.0\}$.

**Infrastructure and throughput.** All experiments use PyTorch FSDP (Zhao et al., 2023a) on NVIDIA A100 GPUs, with prompt lengths capped at 2,048 tokens and total sequence lengths at 4,096 tokens. Training `Mistral7B-v0.3` with DPO on 8 A100 GPUs takes approximately 3 hours for 1 Epoch on Magpie-Pro/G27, while `Qwen3-4B` on 4 A100 GPUs requires about the same time for the same data size.

**Evaluation.** We select the best checkpoint by absolute win-rate on Arena-Hard-v0.1 using `gpt-4.1-nano-2025-04-14` as the judge. Final performance on the Arena-Hard benchmark is reported using the judge `gpt-4.1-2025-04-14` to reduce evaluation costs, following Mao et al. (2024). For Wild-Bench-v2, we use `gpt-4o-2024-08-06` as recommended in the official repository.[11] During inference, we greedy-decode up to 4,096 tokens with vLLM. QA benchmarks are evaluated via the `lm-evaluation-harness` (Gao et al., 2024). The QA benchmarks consist of the following: PIQA (Bisk et al., 2020), SIQA (Sap et al., 2019), HellaSwag (Zellers et al., 2019), ARC-Easy/Challenge (Clark et al., 2018), MMLU (Hendrycks et al., 2021), GSM8k (Cobbe et al., 2021), and BoolQ (Clark et al., 2019).

The mean reciprocal rank (MRR) in Table 1 and Figure 16 provides a single aggregated metric for performance across the 8 QA benchmarks, each with its own primary metric. The procedure for measuring MRR is as follows.

1. For each of the 8 benchmarks, we evaluate all models using a pre-defined standard performance metric:
   - **ARC-Easy/Challenge, BoolQ, MMLU, PIQA, SIQA:** Accuracy.
   - **HellaSwag:** Normalized Accuracy.
   - **GSM8K:** Exact Match (flexible-extract).

2. Based on these scores, we rank the models for each benchmark.

3. We then calculate the reciprocal of each model's rank and average these reciprocal ranks across all 8 benchmarks to obtain the final MRR score.

---

[11] https://github.com/allenai/WildBench

