# OpenReview forum: "Differential Information Distribution: A Bayesian Perspective on Direct Preference Optimization"
_ICLR.cc/2026/Conference — Submitted to ICLR 2026_

### Official Review · Reviewer_TcgT · 2025-10-30

**Soundness:** 2
**Presentation:** 3
**Contribution:** 3
**Rating:** 4
**Confidence:** 4

**Summary:**

This paper presents a Bayesian reinterpretation of Direct Preference Optimization (DPO). It introduces the concept of Differential Information Distribution (DID), which represents the Bayesian evidence required to update a reference policy `π_ref` into a target policy `π*`. The authors show that DPO’s log-ratio reward can be derived as the unique form consistent with a power-law DID structure, and that training dynamics, specifically changes in log-likelihood and exploration, arise naturally from the properties of DID. Furthermore, the paper introduces DID entropy as a principled measure of the uncertainty of alignment information, showing that low-entropy DIDs improve factual QA while high-entropy DIDs improve open-ended generation. Theoretical claims are supported by formal derivations (e.g., the Likelihood Ratio Representation) and experiments on preference-based LLM fine-tuning.

**Strengths:**

1. Establishes a Bayesian formulation of preference optimization, linking DPO to information-theoretic evidence accumulation.
2. Explains DPO’s reward structure, dynamics, and task-dependent behaviors (open-ended vs factual) under one consistent lens.
3. Provides closed-form derivations (e.g., Likelihood Ratio Representation, Entropy of DID) that connect policy updates to Bayesian ratios.
4. Offers a principled way to reason about why different `β` or entropy configurations produce distinct behaviors in alignment training.

**Weaknesses:**

1. The conditional independence of `X` from the prior and the power-law DID assumption are not empirically testable or demonstrated to hold in real preference data.
2. DID “existence” is defined by construction, not derived, which weakens claims of theoretical generality.
3. DID entropy estimation uses a small sample (`K=32`) with potentially large variance; no confidence intervals or significance testing are reported.
4. Theorem 3.2’s “unique justification” of the log-ratio reward is contingent on strong assumptions and may not hold under alternative data generation processes.
5. Experiments lack ablations (e.g., sensitivity to `β`, prompt distribution, or entropy threshold) and multi-seed averages.
6. The analysis remains conceptual, real-world preference datasets are rarely power-law structured or independently sampled as assumed.

**Questions:**

1. Can the authors empirically validate that preference data follow a power-law DID or that `P(X|Y)` is conditionally independent of the prior?
2. How sensitive are the DID entropy and corresponding trends (open-ended vs factual tasks) to sample size and importance weighting variance?
3. Does Theorem 3.2’s “uniqueness” still hold if the DID deviates from power-law structure or when sampling dependencies between `(y_w, y_l)` exist?
4. Can the authors clarify whether the entropy results persist under token-level DID instead of sequence-level ratios?
5. Would alternative estimators (e.g., bootstrapped or token-normalized entropy) yield the same qualitative findings?

---

> ### Author Response · Authors · 2025-11-19
>
> We appreciate the reviewer’s recognition of the novelty of our Bayesian formulation for preference optimization. Below, we address the reviewer’s questions regarding the core assumptions underlying our framework.
>
> ---
> ### **[W2] Existence of DID**
> We would like to first respectfully clarify a potential misunderstanding. **The existence of DID is not defined by construction, instead, its existence follows from a formal result (Lemma D.1).**  In particular, the DID is well-defined and must exist whenever the prior and posterior have full support, which is true for modern softmax-based LLMs. Thus, the DID is a general and principled concept that can describe a wide-range of policy updates.
>
> ---
> ### **[W1,Q1] Justification of the Conditional Independence Assumption**
> While conditional independence cannot be directly validated on real data due to fundamental identifiability limits, **this assumption is conceptually justified, theoretically necessary, and well-established by prior work.**
>
> The conditional independence of $X$ from the prior $\pi_\text{ref}$ given $Y$ plays an essential role in defining the DID. This assumption is motivated by:
> 1. **Conceptual Intuition.** This assumption formalizes the idea that the properties we care about (e.g., safety) are inherent to the content of the sentence $Y$, not the model that produced it. For example, whether `"1+1=2"` is mathematically correct depends only on the sentence itself. Thus, **the probability that a sentence satisfies property $X$ should depend only on $Y$, not on the distribution from which $Y$ was drawn, which is exactly conditional independence (Appendix D)**.
> 2. **Analytical Tractability.** Without conditional independence, any analysis of DID would require modeling the full joint distribution $P(X, Y, \pi_{\text{ref}})$, which is unobservable and impossible to estimate. Conditional independence makes it possible to derive the closed-form DID (Theorem 2.3) and characterize DPO’s reward structure and training dynamics.
> 3. **Established Precedent.** Similar independence assumptions appear in Controlled Text Generation, where distributions are factored into a base LM and a control-dependent likelihood [1,2]. Our assumption also follows this widely adopted convention.
>
> Because the ground-truth joint distribution $P(X,Y,\pi_\text{ref})$ is unknown, conditional independence cannot be empirically validated. However, as detailed above, the conditional independence is a reasonable and well-established assumption that enables the subsequent analysis of preference optimization in Sections 3~5.
>
> ---
> ### **[W1,Q1] Empirical Validation of the Power-Law DID**
> Similarly, **directly testing the power-law DID on real data is infeasible without knowing the underlying distributions $\pi_w$, $\pi_\ell$, and the true target $\pi^\* $.** *This limitation is inherent to any theory involving latent data-generating distributions.* For example, the cross-entropy loss theoretically recovers the data distribution, yet no work empirically verifies this on real-world data due to the same identifiability issues.
>
> However, **when we do have access to the underlying distributions ($\pi_w,\pi_\ell,\pi^\* $), we can empirically demonstrate that the Power-law DID holds for DPO under realistic training settings (finite data and imperfect optimization) in Section 3.3 (Figure 1).**
>
> ---
> ### **[W6] IID as a Canonical Representation, not a Strict Requirement**
> We acknowledge the reviewer's concern that real-world datasets are often constructed in a non-iid manner (e.g., reward comparison). However, **our framework does not strictly require that the data *was originally generated* via iid sampling. It only requires the *existence* of two full-support distributions, $\pi_w$ and $\pi_\ell$,** such that the empirical loss over the dataset $\mathcal D$ can be expressed as the following canonical expectation over these distributions:
>
> $\mathbb E_{(y_w, y_\ell)\sim\mathcal D}[\mathcal L(y_w,y_\ell)]=\sum_{y_i, y_j}\pi_w(y_i)\pi_\ell(y_j)\mathcal L(y_i,y_j)$
>
> **Note that this is a canonical formulation adopted in prior analyses of preference optimization [3,4].**
>
> Furthermore, this does not strictly require the data to have been generated via iid sampling. It only requires that there *exists* two distributions that can generate the dataset as the above formulation. Once these distributions $(\pi_w,\pi_\ell)$ are identified, the findings of our framework can still be applied to those distributions. Therefore, our analysis remains useful for real-world data involving non-iid samples.
>
> ---
> ### **References**
> - [1] Holtzman, et al. "Learning to write with cooperative discriminators." ACL 2018.
> - [2] Krause, et al. "Gedi: Generative discriminator guided sequence generation." EMNLP 2021.
> - [3] Higuchi, et al. "Direct density ratio optimization: A statistically consistent approach to aligning large language models." ICML 2025.
> - [4] Pan, et al. "What Matters in Data for DPO?." NeurIPS 2025.

---

> ### Author Response · Authors · 2025-11-19
>
> ### **[W6] Validation of Training Dynamics on Real Data**
> As noted from our previous comment, we cannot directly observe the latent distributions $\pi_w$ and $\pi_\ell$ to test the power-law DID. However, **we can validate the necessary consequences of our framework.** Theorem 4.1 provides a specific, testable prediction: if the DPO-converged policy follows the DID power-law structure, specific asymmetric shifts in log-likelihood must occur regardless of the original data generation process.
>
> We tested this on three real-world datasets: Magpie-Pro, Magpie-G27, and UltraFeedback (50k samples each). We fine-tuned a reference policy (`Qwen3-4B-Base`) on either the chosen or rejected responses, then applied DPO ($\beta=1.0$) for 1 epoch.
>
> As predicted by Theorem 4.1, the dynamics on real data perfectly match the theoretical predictions of the DID framework:
> 1. When fine-tuning $\pi_\text{ref}$ on the chosen samples $y_w$, DPO consistently **decreases** $\log\pi(y_\ell)$.
> |Dataset|$\log\pi(y_\ell)-\log\pi_\text{ref}(y_\ell)$ at Final Training Step|
> |-|-|
> |Magpie-Pro|-1.1025|
> |Magpie-G27|-2.2510|
> |Ultra-Feedback|-1.2961|
> 2. When fine-tuning $\pi_\text{ref}$ on the rejected samples $y_\ell$, DPO consistently **increases** $\log\pi(y_w)$.
> |Dataset|$\log\pi(y_w)-\log\pi_\text{ref}(y_w)$ at Final Training Step|
> |-|-|
> |Magpie-Pro|+0.5532|
> |Magpie-G27|+0.8477|
> |Ultra-Feedback|+0.8113|
>
> This confirms that **even if real-world data contains sampling dependencies, DPO training dynamics behave as if the data were generated by distributions satisfying our framework.** This demonstrates that our framework can accurately describe DPO's training dynamics on real-world data, too.
>
> -----
> ### **[W4,Q3] Unique Optimality of DPO under Data Deviations**
> The reviewer asks if Theorem 3.2’s unique optimality holds if the data deviates from the power-law structure.
>
> First, it is important to clarify that in our framework, **$\pi^\* $ represents the target policy to be learned, not necessarily the globally *optimal* policy**. Corollary 3.2.1 establishes that the policy to which DPO converges must satisfy the DID power-law. Consequently, if the ground-truth optimal policy does not follow this power-law, DPO will not converge to it.
>
> However, it is worth mentioning that **our paper does not claim that DPO is optimal for arbitrary real-world datasets.** Rather, Theorem 3.2 shows that under a specific Bayesian setting where preferences encode the Differential Information required to update $\pi_\text{ref}$ into $\pi^\* $, the DPO log-ratio reward is the unique Bradley–Terry reward capable of recovering $\pi^\* $.
>
> While real-world preference data may admit an optimal policy that violates the power-law DID (a property difficult to verify without access to the global optimum), this does not contradict our findings. Even in such cases, the DPO-converged policy can provide a meaningful improvement over the reference policy. **Our core contribution is that we established the first fundamental, theoretical link between the unique Bayesian property of DPO in learning Differential Information and such empirical improvements.**
>
> ---
> ### **[W4,Q3] Evidence that Real Data Can Encode Differential Information**
> The empirical success of DPO suggests that real-world preferences **do** largely encode differential information, even for datasets involving non-iid samples.
>
> We performed an additional experiment on two real-world datasets (Magpie-Pro and Ultra-Feedback) which were constructed by reward comparisons. We compared the DPO log-ratio reward (which assumes preferences encode the Differential Information between $\pi^\* $ and $\pi_\text{ref}$) against a reference-free reward form (e.g., SimPO/CPO, which assumes preferences directly encode $\pi^\* $).
>
> We trained `Mistral-7B-v0.3` using both objectives across a sweep of $\beta \in\\{0.2, 0.1, 0.05, 0.02\\}$. We evaluated them on 1,000 unseen prompts by measuring the average reward of sampled completions using `Skywork-Reward-V2-Llama-3.1-8B`.
>
> **Table A: Comparison of Different Reward Parameterizations**
> |Dataset|Reward|Best Reward Improvement over $\pi_\text{ref}$|
> |-|-|-|
> |Ultra-Feedback|$r=\beta\log(\pi/\pi_\text{ref})$|**+8.20**|
> ||$r=\beta\log\pi$|+3.77|
> |Magpie-Pro|$r=\beta\log(\pi/\pi_\text{ref})$|**+7.41**|
> ||$r=\beta\log\pi$|+3.41|
>
> Overall, the DPO log-ratio reward consistently outperforms the reference-free formulation, improving upon the reference policy. This provides strong empirical evidence that **even for non-iid real data, its preferences often still encode differential information required to update a reference policy**, rather than encoding the standalone target policy.

---

> ### Author Response · Authors · 2025-11-19
>
> We will now address the questions regarding the reliability of our DID entropy estimation and the scope of our ablations.
>
> ---
> ### **[W3,Q2] Confidence Intervals for DID Entropy Estimation**
> We thank the reviewers for pointing out the importance of validating the stability of our entropy estimates. In response, we computed 95% bootstrap confidence intervals (10,000 resamples) for all DID entropy values reported in Table 1. The results below show that **the estimates are stable and the differences across methods are statistically significant**. In particular, the CIs for the low-entropy DPO-PG models are cleanly separated from those of the high-entropy DPO models, directly supporting our central claim.
>
> |Model|Method|DID Entropy|95% CI|
> |-|-|-|-|
> |`Mistral-7b-v0.3`|DPO ($\beta=1.0$)|1123.2|(1066.3, 1186.8)|
> ||DPO ($\beta=0.2$)|1303.4|(1229.0, 1380.6)|
> ||DPO ($\beta=0.1$)|1253.9|(1172.9, 1337.2)|
> ||DPO ($\beta=0.05$)|970.2|(910.6, 1032.4)|
> ||DPO-PG|495.1|(477.8, 513.2)|
> |`Qwen3-4B`|DPO ($\beta=1.0$)|9158.4|(8746.5, 9566.7)|
> ||DPO ($\beta=0.2$)|4765.9|(4543.9, 4997.9)|
> ||DPO ($\beta=0.1$)|7663.3|(7518.2, 7813.4)|
> ||DPO ($\beta=0.05$)|6801.2|(6686.4, 6922.7)|
> ||DPO-PG|388.2|(374.9, 402.6)|
>
> ---
> ### **[Q4,Q5] On Alternative Entropy Estimators (Token-level, Length-normalized, etc.)**
> We believe that sequence-level DID entropy is the most principled and theoretically grounded metric for our setting. The DID distribution $q_{\pi^\* /\pi_\text{ref}}$ is defined over **entire sequences** $y$, not individual tokens. Since a sequence log-probability is the sum of token log-probabilities, the ratio $\frac{\pi^\* (y)}{\pi_\text{ref}(y)}$ and its entropy must also be defined at the sequence level to correctly capture the total differential information involved in the Bayesian update: $\pi_\text{ref}\rightarrow\pi^\* $.
>
> For this reason, **it is unclear how a “token-level DID entropy” could be coherently defined or empirically estimated.** A similar issue arises for length-normalized entropy. Normalizing the $\log Z$ term or the log-ratio term $\mathbb E_{y\sim q_{\pi/\pi_\text{ref}}}[\log\frac{\pi(y)}{\pi_\text{ref}(y)}]$ by token length has no clear interpretation within the DID framework and, most crucially, does not measure the uncertainty in the DID.
>
> **Given that (1) the sequence-level entropy is the correct theoretical object, and (2) our SNIS-based estimator is principled and empirically stable (as confirmed by the confidence intervals above), we believe this is the appropriate metric for our analysis.**
>
> ---
> ### **[W5] Ablation on Hyper-parameters and Seeds**
> We appreciate the reviewer's concern about ablations. We believe our experiments already include extensive ablations, which we are happy to highlight more clearly.
>
> * **Controlled Experiments (Sec 3.3 & 4.3, Details in Appendix K.1):**
>     * **Methods:** We compare DPO against 4 different methods (SimPO, Cal-DPO, ORPO, SLiC) across 4 different preference data generation setups (Figures 1 & 7). We searched for the best hyper-parameter for each baseline following standard practice [5,6]
>     * **Hyper-parameters :** We explicitly test over a wide range of $\beta$ values to verify our theoretical predictions (Figures 2 & 3).
>     * **Multi-Seed:** Our EBM experiment in Figure 4 was averaged over 5 random seeds across 200 different $\beta$ configurations ($\beta_\ell$ vs $\beta_r$), confirming the robustness of our findings.
>
> * **Real-World Experiments (Sec 5, Details in Appendix K.2.):**
>     * **Models & Data:** We ablate across 2 different model architectures (`Mistral7B-v0.3`, `Qwen3-4B`) and 2 different large-scale datasets (Magpie-Pro, Magpie-G27).
>     * **Hyper-parameters:** We test 4 different $\beta$ values for standard DPO in all real-world setups (Tables 1 & 2, Figure 12). We also selected the best checkpoint from 5 epochs, which is more than the standard setup of 3 training epochs as in [7,8].
>     * **Evaluation:** We evaluate on a comprehensive suite of 10 different tasks (2 open-ended, 8 QA) to ensure our conclusions are not based on a single benchmark.
>
> We therefore believe these comprehensive tests sufficiently address the concerns about ablations and robustness.
>
> ---
> ### **References**
> - [5] Meng, et al. "Simpo: Simple preference optimization with a reference-free reward." NeurIPS 2024.
> - [6] Xiao, et al. "Cal-dpo: Calibrated direct preference optimization for language model alignment." NeurIPS 2024.
> - [7] Tunstall, et al. "Zephyr: Direct distillation of lm alignment." COLM 2024.
> - [8] Hong,  et al. "Orpo: Monolithic preference optimization without reference model." EMNLP 2024.

---

> ### Author Response · Authors · 2025-11-25
>
> Dear Reviewer TcgT,
>
> Thank you once again for the time and effort you’ve dedicated to reviewing our work.
>
> As the discussion period nears its end, we wanted to kindly follow up on our rebuttal.  We believe that some of the concerns raised may stem from a potential misunderstanding of our main findings, which we have aimed to clarify in our response. **In the common response, we have also provided further empirical evidence that the DID power-law (Corollary 3.2.1) holds on real data.**
>
> If there are any remaining questions or concerns, we would be grateful for the opportunity to address them.
>
> Thank you again for your time and consideration.

---

### Official Review · Reviewer_KUMF · 2025-10-31

**Soundness:** 3
**Presentation:** 2
**Contribution:** 3
**Rating:** 6
**Confidence:** 4

**Summary:**

This paper introduces Differential Information Distribution (DID), which represents the distribution over samples that carry the Bayesian evidence required to update policies. The paper provides theoretical insights to prove that the log-ratio reward of DPO is the only optimal solution when preferences encode the differential information. Through DID analysis of policy training dynamics, the DID entropy is empirically linked to the performance of the model on open-ended instruction-following and knowledge-intensive QA tasks.

**Strengths:**

+ The paper introduces the Differential Information Distribution (DID), providing a deep understanding of how Bayesian evidence drives the updating of policies in DPO.

+ The paper demonstrates that the reward parameterization, training dynamics, and learned capabilities in DPO emerge naturally by analyzing DID.

+ By analyzing the Shannon entropy of the DID, this paper demonstrates how DID entropy influences the trade-off between factual accuracy and open-ended task performance via a real LLM experiment in Section 5.

**Weaknesses:**

+ The controlled Energy-Based Model experiments and empirical tests (Figure 1, Figure 2, Figure 3) in Section 3 and 4 are built around strong assumptions of matched data generating processes and synthetic settings where DIDs align almost perfectly. While some results (Table 1) use real-world LLMs and datasets, they are limited. It's uncertain whether the findings from synthetic setups apply to LLMs on real tasks.

+ There is the assumption that $\pi_{w} = \pi_{\text{ref}}$ in Section 4, along with the assumption that $\pi_{\text{ref}}$ is the unoptimized policy. However, the initial policy is not necessarily fine-tuned on $y_w$, and there is research focusing on applying different $\pi_{\text{ref}}$ strategies, which limits the applicability of the theory.

+ There are too many theorems, and many key proofs are in the appendix, which increases the difficulty of reading and hampers the flow.

**Questions:**

+ Can the authors explain why SimPO's JS divergence exhibits a distinct behavior in Figure 1 (Right)?

+ How about the performance of other baselines like SimPO in the experiment of Table 1? Can the same phenomenon be observed?

---

> ### Author Response · Authors · 2025-11-21
> **[W1] Generalizability of EBM Experiments to Real-World LLMs**
>
> We thank the reviewer for their insightful comment. We are encouraged that the reviewer found our work to provide "*a deep understanding of how Bayesian evidence drives the updating of policies in DPO*".
>
> Regarding W1, we would like to thank the reviewer for suggesting that we extend our analysis to real-world tasks. To address the reviewer's request, during the rebuttal period, **we conducted additional experiments on three real-world datasets and found that the predicted log-likelihood changes of Theorem 4.1 hold consistently. We also compared our theoretical predictions on DPO policy exploration (Theorem 4.2) with recent empirical findings and observed strong agreements.**
>
> ---
> ### **1. Role of EBM Experiments**
> First, we wish to clarify that the EBM experiments in Sections 3.3 and 4.3 were not intended solely as "perfectly ideal" synthetic settings. Rather, they served to bridge the gap between our theoretical derivations (which assume infinite samples and perfect optimization) and real-world conditions (finite samples and stochastic gradient descent). They confirm that **the DID framework remains valid under realistic training conditions involving finite samples and noisy optimization.**
>
> ---
> ### **2. New Experiment: Verifying Log-Likelihood Shifts on Real LLMs**
> To directly address the reviewer's concern about generalizability, we extended the empirical validation of **Theorem 4.1 (Log-Likelihood Change of DPO)** to LLMs trained on real-world data.
>
> We used `Qwen3-4B` on three distinct real datasets: Magpie-Pro, Magpie-G27, and Ultra-Feedback [1] (50k samples each). Unlike the synthetic setting, **preference pairs in these datasets were *not* generated by IID sampling but were selected based on reward comparisons,** which is common in real-world settings [1,2]. For each dataset, we constructed the reference policy $\pi_\text{ref}$ by fine-tuning the base model on either the chosen or rejected responses, then applied DPO for one epoch ($\beta=1.0$).
>
> **The empirical results on these real-world tasks consistently match the predictions of Theorem 4.1:**
> 1. When fine-tuning the reference policy on the chosen samples, DPO training **decreases** the average log-probability of the rejected samples $y_\ell$ across all datasets.
>
>     |Dataset|$\log\pi(y_\ell)-\log\pi_\text{ref}(y_\ell)$ at Final Training Step|
>     |-|-|
>     |Magpie-Pro| -1.1025|
>     |Magpie-G27| -2.2510 |
>     |Ultra-Feedback| -1.2961 |
>
> 2. When fine-tuning the reference policy on the rejected samples, DPO training **increases** the average log-probability of the chosen samples $y_w$ across all datasets.
>
>     |Dataset|$\log\pi(y_w)-\log\pi_\text{ref}(y_w)$ at Final Training Step|
>     |-|-|
>     |Magpie-Pro| +0.5532|
>     |Magpie-G27| +0.8477 |
>     |Ultra-Feedback| +0.8113 |
>
> **These results demonstrate that the log-likelihood dynamics predicted by our Bayesian framework extend to real-world training environments and data.**
>
> ---
> ### **3. External Validation: Policy Exploration (Theorem 4.2)**
> Furthermore, we additionally verified that **the predictions of Theorem 4.2 closely align with empirical behaviors observed in DPO training on real data.**
>
> Our framework predicts that policy exploration is jointly determined by the KL-penalty ($\beta$) and the sharpness (quality) of preference data (Theorem 4.2). Specifically, stronger preference signals drive the policy further from the reference, requiring a higher $\beta$ to maintain the same KL-constraint.
>
> **This theoretical prediction is strongly supported by independent empirical findings in $\beta$-DPO (Wu et al., NeurIPS 2024) [3].** Their extensive experiments across model sizes (410M to 2.8B) show that easily discriminated preference pairs (high sharpness) require higher $\beta$ values for optimal alignment, while noisy pairs require lower $\beta$ values. This aligns with our Theorem 4.2, which predicts strong preference signals to require higher KL-penalties than weaker preference labels.
>
> To the best of our knowledge, **our work provides the first principled theoretical explanation for such real-world trends in DPO.** The DID power-law explains how preference data quality and $\beta$ value jointly determines the effective KL-budget of DPO.
>
> ---
> ### **Conclusion**
> Our additional experiments on `Qwen3-4B` and the strong alignment with concurrent empirical studies confirm that our Bayesian framework provides valid and actionable insights for real-world LLM alignment, extending beyond synthetic settings.
>
> To address the reviewer's concerns, we will add the real-data experiments of Theorem 4.1 and the independent empirical support of Theorem 4.2 in the camera-ready version.
>
> ---
> ### **References**
> - [1] Cui, et al. "Ultrafeedback: Boosting language models with scaled ai feedback." ICML 2024.
> - [2] Xu, et al. "Magpie: Alignment data synthesis from scratch by prompting aligned llms with nothing." ICLR 2025.
> - [3] Wu, et al., "β-DPO: Direct Preference Optimization with Dynamic β" NeurIPS 2024.

---

> ### Author Response · Authors · 2025-11-21
>
> ### **[W2] Assumptions of the Reference Policy**
> We appreciate the reviewer for raising the concern regarding the assumption that the reference policy $\pi_\text{ref}$ is fine-tuned on either the chosen or rejected response, and how this might limit the generality of our analysis.
>
> To clarify the applicability of our theory, it is important to distinguish between the general theoretical framework in Section 3, which holds **universally without specific assumptions of $\pi_\text{ref}$**, and the analysis of training dynamics in Section 4, where we do apply the assumption ($\pi_\text{ref} = \pi_w$ or $\pi_\text{ref} = \pi_\ell$) to analytically prove DPO's training properties in **standard settings**.
>
> **1. The general framework in Section 3 holds for any $\pi_\text{ref}$ with full-support.**
>
> Our core theoretical contribution of interpreting preference optimization as learning Differential Information does *not* rely on the assumption that $\pi_\text{ref} = \pi_w$ or $\pi_\text{ref} = \pi_\ell$. In particular, **the entire analysis of Section 3 holds regardless of how $\pi_\text{ref}$ is initialized.** Therefore, our Bayesian formulation of DPO presented in Section 3 is general and applicable to various reference policy configurations.
>
> **2. The assumption in Section 4 follows standard practice.**
>
> While Section 3 provides the general framework, Section 4 aims to explain the specific training dynamics observed in standard practice. In particular, Section 4 assumes sampling either the chosen or rejected response from the reference policy. We would like to note that **this reference policy assumption has been widely adopted in the RLHF and preference optimization literature:**
> * A large body of prior work fine-tunes the reference policy on chosen samples before applying DPO, which is in principle equivalent to drawing chosen samples from the reference policy [4,5].
> * Likewise, the opposite configuration where rejected samples come from the reference policy has also appeared in prior work [6].
> * Furthermore, the vast majority of works on "*On-Policy Preference Optimization*" deals with preference pairs sampled from the current reference policy [7].
>
> For this reason, **the reference-policy setup used in Section 4 is a common and practically important regime in preference optimization.** Although this assumption may not hold universally, our aim is to theoretically formalize and explain several empirically observed behaviors within this widely used setting, which underscores the value of our analysis. We believe that exploring alternative reference policy formulations is an interesting direction for future work.
>
> ---
> ### **[W3] Presentation of Technical Details**
> We thank the reviewer for this helpful feedback. Due to the 9-page submission limit, we had to move the proofs to the appendix. **For the camera-ready version, we will utilize the additional page to bring the general outline of the proofs back into the main text to enhance readability.**
>
> We are also happy to make specific revisions to address any areas the reviewer found reduced readability. **If the reviewer believes certain theorems would be clearer if presented as lemmas or supporting propositions, we are also ready to reorganize the presentation accordingly.**
>
> ---
> ### **References**
> - [4] Rafailov, et al. "Direct preference optimization: Your language model is secretly a reward model." NeurIPS 2023.
> - [5] Tunstall, et al. "Zephyr: Direct distillation of lm alignment." COLM 2024.
> - [6] Li, et al. "Direct preference knowledge distillation for large language models." arXiv:2406.19774.
> - [7] Guo, et al. "Direct language model alignment from online ai feedback." arXiv:2402.04792.

---

> ### Author Response · Authors · 2025-11-21
>
> ### **[Q1] SimPO's Behavior in Figure 1**
> Thank you for pointing out this interesting observation.
>
> **The non-convergence of SimPO to the target policy can be understood through Theorem 3.2.** According to Theorem 3.2, the log-ratio form is the *unique* BT reward capable of recovering the target policy when preferences encode Differential Information (Theorem 3.1). Consequently, the reference-free reward used by SimPO is theoretically guaranteed to **not** recover the target policy.
>
> Compared to other objectives, SimPO does initially move toward the target policy but then converges to a policy different from the target policy. We suspect this occurs because the optimization trajectory passes near the target policy before settling elsewhere, i.e., the converged policy lies in a region such that the path makes a close “fly-by.” **To the best of our knowledge, the current literature does not have a principled explanation for this specific trajectory shape.**
>
> As a supplementary note, we evaluated **12 different hyper-parameter configurations** for SimPO following the recommendations of the original work [8], and selected the setting with the smallest JS-divergence to the target policy. However, across all configurations, we observed similar patterns indicating that this trend is **not** due to limitations in our experimental setup but is instead characteristic of the SimPO objective itself.
>
> ---
> ### **[Q2] Testing SimPO for Table 1**
> We appreciate the reviewer for suggesting additional comparisons (SimPO) for Table 1. **To address this request, we conducted additional experiments on `Qwen3-4B` and found that the trends of DID entropy and downstream performance remain consistent for SimPO as well.**
>
> Before discussing SimPO, we wish to clarify that we attempted to benchmark against other LLD-mitigation objectives (DPOP, Cal-DPO). However, as detailed in Appendix I, we found these methods to be highly sensitive to hyper-parameters at the scale of our setup (4B & 7B models, 100K samples, 5 epochs). We could not identify configurations that successfully mitigated LLD without causing mode collapse. This instability was the core motivation for DPO-PG, which utilizes projected gradient descent to stably prevent LLD without introducing any new hyper-parameters.
>
> To address the reviewer’s request, we ran additional experiments on `Qwen3-4B` using SimPO under the Table 1 setup. SimPO was also shown to be sensitive to hyper-parameters and prone to over-optimization on our datasets. To address this training difficulty and high computational costs, we trained for 2 epochs, saving checkpoints every 20k samples (10 checkpoints total), and selected the best checkpoint following the same procedure of DPO and DPO-PG (Appendix K.2).
>
> **We found that SimPO results in LLD, and its DID entropy and downstream performance trends closely mirror those observed in DPO, which also exhibit LLD.**
>
> * **Table A: Performance Trends of `Qwen3-4B` on Magpie-Pro Dataset.**
> |Method|DID Entropy|AH (↑)|WB (↑)|QA (↑)|
> |-|-|-|-|-|
> |DPO ($\beta=1.0$)|9158.4|30.7|1134.4|0.37|
> |DPO ($\beta=0.2$)|4765.9|28.1|1146.2|0.30|
> |DPO ($\beta=0.1$)|7663.3|27.5|1148.2|0.23|
> |DPO ($\beta=0.05$)|6801.2|**43.7**|1164.5|0.20|
> |**SimPO** ($\beta=10,\gamma=5$)|1071.4|42.8|**1177.5**|0.50|
> |DPO-PG|388.2|37.4|1148.5|**0.85**|
>
> *Note: QA MRR values differ slightly from Table 1 because MRR is a relative metric which shifts when additional methods (SimPO) are included.*
>
> As shown in Table A, SimPO follows the same trade-off observed in DPO. SimPO learns a high-entropy DID which improves performance on open-ended tasks (AH, WB) at the cost of degraded accuracy in factual QA tasks. These results confirm that **the DID entropy and downstream performance trends extend to SimPO as well.**
>
> This demonstrates our Bayesian interpretation linking training dynamics (LLD) and downstream performance is robust across diverse preference-optimization objectives.
>
> ---
> ### **References**
> - [8] Meng, et al. "Simpo: Simple preference optimization with a reference-free reward." NeurIPS 2024.

---

> ### Author Response · Authors · 2025-11-25
>
> Dear Reviewer KUMF,
>
> Thank you again for your thoughtful review, as well as for recognizing our theoretical contribution in establishing a Bayesian framework for preference optimization. We would like to briefly follow up on our rebuttal.
>
> In our response, we provided additional evidence demonstrating that our findings extend to real-world tasks, clarified the applicability of the reference policy assumption, and included new empirical results using SimPO in Table 1 as requested. **In the common response, we also supplied further empirical evidence showing that the DID power-law (Corollary 3.2.1) holds on real data.**
>
> We would be very grateful if you could take these clarifications and new results into consideration when evaluating our work. Please feel free to reach out if you have any remaining questions.
>
> Thank you again for your time and for engaging with our work.

---

### Official Review · Reviewer_9kxT · 2025-10-31

**Soundness:** 2
**Presentation:** 3
**Contribution:** 2
**Rating:** 2
**Confidence:** 4

**Summary:**

The paper introduces an idea of differential information which captures the information needed to update a reference policy to a target policy. They utilize this framework to show that when the differential information distribution between preferred and dispreferred is related to the DID between the target and reference policy by a power law, the preference data contains the information needed to learn the target policy. They also demonstrate how DPO learns the optimal policy under this framework and setting. They provide validation on synthetic data and show that high and low DID entropy is correlated with different model behavior.

**Strengths:**

Framework - The paper provides a thorough analysis under the DID framework and show conditions under which the DPO reward is optimal. The framework utilizes a Bayesian perspective and brings a new approach to analyzing the behavior of DPO.  The analysis leads to potential new insights on the effects of likelihood displacement and the structure of preference data.

**Weaknesses:**

Justification - The framework while interesting and novel lacks justifications for key assumptions and definitions. For example, in definition 2.2, it is unclear why should conditional independence and a bayesian update model preference data and learning well. The framework relies on this definition, so it is important that justification and empirical support is provided. Furthermore, in Theorem 4.1, it is assumed that either preferred or dispreferred responses are sampled from the reference model which is in many settings not the case. Lastly, there is a lack of empirical evidence on real-world data that these assumptions hold or that the results do closely align with practice.

Experimental results - Following up on previous comments, the experiments do not provide strong support for the claims. In particular, a demonstration of the power law on real-world data should be provided as well as a comparison across different rewards. Other results that would support claims are verifying Theorem 4.1 which can be directly verified on real-world data. Currently, the primary results on real-world data are a comparison between DPO and DPO-PG which was not a central part of the earlier results.

Direct verification of results and justification of key assumptions would greatly improve the paper.

**Questions:**

- Can you provide justification for Definition 2.2?
- How might Theorem 4.1 generalize to other datasets?
- Can you directly verify the power-law relationship, the optimality of DPO, or the likelihood change on real-world data?

---

> ### Author Response · Authors · 2025-11-19
> **[W1,Q1] Justification of the DID Definition**
>
> We would like to thank the reviewer for the detailed feedback and for recognizing that our analysis provides a new lens for understanding preference optimization and the structure of preference data.
>
> Regarding Q1, we appreciate the opportunity to elaborate on the motivation and justification behind Definition 2.2. The two assumptions, Bayesian update and Conditional Independence, provide a principled and tractable way to model the information learned during preference optimization.
>
> ---
> ### **1. Justification for the Bayesian Update**
> The core goal of preference optimization is to update an initial reference policy ($\pi_\text{ref}$) into an improved target policy ($\pi^\* $) using a dataset of pairwise preferences. We propose that this process is most naturally and formally modeled as a **Bayesian belief update**.
>
> * In this view, the reference policy $\pi_\text{ref}$ represents the **prior belief** (the model's state before learning from preferences).
> * The target policy $\pi^\* $ represents the **posterior belief** (the desired state after learning).
> * The preference data, therefore, must contain the **Bayesian evidence** (which we denote as $X$, or Differential Information) required to drive the update from the prior to the posterior.
>
> **The Bayesian update rule ($\pi^\* (y)=P(Y=y|\pi_\text{ref},X)$) is the canonical framework for describing this transformation of beliefs, and has been widely used in prior work [1,2].** It allows us to characterize the abstract information $X$ that the model must learn to recover the target policy.
>
> ---
> ### **2. Justification for Conditional Independence**
> The conditional independence assumption is motivated by three key factors:
>
> 1.  **Conceptual Intuition:** This assumption formalizes the idea that the properties we care about (e.g., helpfulness, safety) are **inherent to the content of the sample $Y$** (the sentence), not the model that produced it. For example, the mathematical correctness of the sentence ``"1+1=2"`` is an intrinsic property of that sentence; it doesn't depend on whether it was sampled from $\pi_\text{ref}$ or any other model. This translates to the statement that the probability of $Y=y$ having $X$ does not depend on whether it was sampled from $\pi_\text{ref}$, which is essentially conditional independence ($P(X|Y=y)=P(X|Y=y,\pi_\text{ref})$). Our framework is aimed to analyze this type of sample-inherent information. We have provided a detailed explanation with illustrative examples in **Appendix D**.
>
> 2.  **Analytical Tractability:** This assumption enables the subsequent closed-form analysis in Sections 3~5. **Without conditional independence, we would need to model the full joint distribution $P(X, Y, \pi_\text{ref})$, which is unobservable and empirically unverifiable.** Conditional independence allows us to derive the closed-form expression for the DID in Theorem 2.3, enabling the subsequent analysis of DPO's reward structure and training dynamics. Without this assumption, it is simply impossible to characterize the evidence distribution using only the prior and posterior distributions.
>
> 3.  **Established Precedent:** Similar independence assumptions are standard in adjacent areas. For example, in Controlled Text Generation, “weighted decoding” methods [3,4,5,6] factor the controlled distribution into the base model’s prior distribution and a control-dependent likelihood, effectively mirroring our conditional independence formulation. Therefore, the conditional independence assumption can be considered as a well-established notion accepted by prior relevant works.
>
> ---
> In summary, **the Bayesian update rule provides a principled framework for characterizing the change in beliefs, while the conditional independence assumption provides a tractable formulation that makes subsequent analysis possible, all while aligning with the intuition that information is an inherent property of the data samples themselves.** These assumptions justify Definition 2.2 as a principled and meaningful way to characterize the Bayesian evidence that drives policy updates in preference optimization.
>
> We once again thank the reviewer for raising these important questions about the core concepts underlying our framework. If any aspects remain unclear, we would be more than happy to provide further clarifications.
>
> ---
> ### **References**
> - [1] Korbak, et al. "RL with KL penalties is better viewed as Bayesian inference." EMNLP 2022.
> - [2] Chaudhari, et al. "Rlhf deciphered: A critical analysis of reinforcement learning from human feedback for llms." ACM Computing Surveys 2025.
> - [3] Holtzman, et al. "Learning to write with cooperative discriminators." ACL 2018.
> - [4] Krause, et al. "Gedi: Generative discriminator guided sequence generation." EMNLP 2021.
> - [5] Liu, et al. "DExperts: Decoding-time controlled text generation with experts and anti-experts." ACL 2021.
> - [6] Yang, et al. "FUDGE: Controlled text generation with future discriminators." NAACL 2021.

---

> ### Author Response · Authors · 2025-11-19
> **[W2,Q2] How Theorem 4.1 Can Generalize to Other Datasets**
>
> We would like to thank the reviewer for raising important questions about the core assumptions of Theorem 4.1. Below, we clarify why this assumption aligns with standard practice in the literature and how it can be extended and empirically examined on real-world datasets.
>
> In particular:
>
> * The assumption in Theorem 4.1 that one side of the preference pair is sampled from the reference policy is a standard and widely adopted setup in RLHF.
> * Our analysis relies only on the *existence* of two underlying distributions, independent of the true specific data-generation process.
> * New experiments on real-world preference datasets show that the theorem’s conclusions hold under realistic, non-IID conditions.
>
> ---
> ### **1. The Standard Assumption of Reference-Sampled Pairs**
> We first note that sampling either the chosen or rejected response from the reference policy has been widely adopted in the RLHF and preference optimization literature.
> - A large body of prior work fine-tunes the reference policy on chosen samples before applying DPO, which is in principle equivalent to drawing chosen samples from the reference policy [7,8].
> - Likewise, the opposite configuration where rejected samples come from the reference policy has also appeared in prior work [9].
> - Furthermore, the whole field of on-policy preference optimization deals with preference pairs sampled from the reference policy [10].
>
> As such, **the setup in Sections 4~5 is a common and practically important regime in which preference optimization is typically performed.** While this assumption may not hold universally, our goal is to theoretically formalize and explain several empirically observed behaviors in this widely used setup, which we believe makes our analysis valuable.
>
> ---
> ### **2. Why our theoretical framework still applies beyond this assumption**
> Our theoretical results require only that preference pairs **can be expressed** as iid samples from two underlying distributions. Formally, we only assume that there exists two distributions $\pi_w,\pi_\ell$ with full-support such that the following holds:
>
> $\mathbb E_{(y_w,y_\ell) \sim \mathcal{D}}[\mathcal{L}(y_w,y_\ell)]=\sum_{i,j}\pi_w(y_i)\pi_\ell(y_j)\mathcal{L}(y_i,y_j).$
>
> **Note that this is a canonical formulation adopted in prior analyses of preference optimization [11,12].**
>
> Crucially, this does not require the data to have been generated via iid sampling. It only requires that there **exists** two distributions that can generate the dataset as the above formulation. Once these distributions $(\pi_w,\pi_\ell)$ are identified, our framework can still be applied. **Thus, the applicability of Theorem 4.1 can be extended beyond the original generative assumption.**
>
> In the following real-world experiments, we demonstrate this applicability of Theorem 4.1.
>
> ---
> ### **3. Empirical validation on real-world preference data**
> To directly address the reviewer’s request, we conducted new experiments verifying Theorem 4.1 on three real-world preference datasets: Magpie-Pro, Magpie-G27, and Ultra-Feedback (Cui, et al., 2024), 50k samples each. In these datasets, preference pairs were not generated by iid sampling from two distributions; instead, they were selected based on estimated reward scores.
>
> For each dataset, we formed the reference policy by fine-tuning `Qwen3-4B-Base` on either the chosen or rejected responses, and then applied DPO for one epoch with $\beta=1.0$. The empirical results consistently match the predictions of Theorem 4.1:
>
> 1. When fine-tuning the reference policy on the chosen samples, DPO training **decreases** the average log-probability of the rejected samples across all datasets.
> |Dataset|$\log\pi(y_\ell)-\log\pi_\text{ref}(y_\ell)$ at Final Training Step|
> |-|-|
> |Magpie-Pro|-1.1025|
> |Magpie-G27|-2.2510|
> |Ultra-Feedback|-1.2961|
> 2. When fine-tuning the reference policy on the rejected samples, DPO training **increases** the average log-probability of the chosen samples across all datasets.
> |Dataset|$\log\pi(y_w)-\log\pi_\text{ref}(y_w)$ at Final Training Step|
> |-|-|
> |Magpie-Pro|+0.5532|
> |Magpie-G27|+0.8477|
> |Ultra-Feedback|+0.8113|
>
> These results show that **the predicted log-likelihood change of Theorem 4.1 holds on real-world preference data,** which demonstrates the practical relevance of our framework.
>
> ---
> ### **References**
> - [7] Rafailov, et al. "Direct preference optimization: Your language model is secretly a reward model." NeurIPS 2023.
> - [8] Tunstall, et al. "Zephyr: Direct distillation of lm alignment." COLM 2024.
> - [9] Li, et al. "Direct preference knowledge distillation for large language models." arXiv:2406.19774.
> - [10] Guo, et al. "Direct language model alignment from online ai feedback." arXiv:2402.04792.
> - [11] Higuchi, et al. "Direct density ratio optimization: A statistically consistent approach to aligning large language models." ICML 2025.
> - [12] Pan, et al. "What Matters in Data for DPO?." NeurIPS 2025.

---

> ### Author Response · Authors · 2025-11-19
> **[W2,Q3] Real-World Validation of the DID**
>
> We would like to clarify what can and cannot be verified using real-world preference data, and provide additional empirical results as requested. Regarding Theorem 4.1, as noted from our earlier response, we were able to empirically validate the log-likelihood change on three real-world preference datasets.
>
> ---
> ### **1. Verifying the Power-Law DID Relationship**
> We would like to first acknowledge that directly testing the DID power-law on real data is fundamentally impossible without access to the true underlying distributions ($\pi_w,\pi_\ell$) and the target policy $\pi^\* $. **This limitation is inherent to any learning theory involving unknown data-generating distributions.**
>
> For instance, although minimizing the cross-entropy loss recovers the data-generating distribution in principle, no prior work empirically verifies this on real data due to finite samples, noisy optimization, and, most critically, the lack of access to the ground-truth data distribution. Nevertheless, the cross-entropy loss remains a principled and widely-used objective for distribution matching.
>
> As such, **the inability to test the DID power law on real data does not necessarily undermine its theoretical validity or practical relevance.**
>
> ---
> ### **2. Verifying the Optimality of DPO**
> Likewise, verifying the optimality of DPO on real data is infeasible without access to the global optimal policy. However, **our paper does not claim that DPO is optimal for arbitrary real-world datasets**. In our framework, $\pi^\* $ is a target policy we aim to learn, not necessarily the globally optimal one.
>
> Our contribution is that **under a specific Bayesian setting where preferences encode the Differential Information required to update $\pi_\text{ref}$ into $\pi^\* $, the DPO log-ratio reward is the *unique* Bradley–Terry reward that recovers $\pi^\* $**. When real data deviates from this setting, DPO is not expected to learn a globally optimal policy. Instead, it will converge to a policy satisfying the power-law DID relationship (Corollary 3.2.1). If this converged policy empirically improves over $\pi_\text{ref}$, we can infer that preference data encodes the differential information necessary for policy improvement. DPO’s strong performance across various domains [13] suggests that this condition often holds in real-world preference datasets.
>
> ---
> ### **3. Verifying the DID of Real-World Preferences**
> Following the reviewer’s request, we conducted an additional experiment to test whether real-world preference datasets align more closely with the DPO log-ratio reward $r=\beta\log(\pi/\pi_\text{ref})$ or with an alternative, reference-free reward $r=\beta\log\pi$ commonly used in SimPO or CPO (Xu, et al., 2024).
>
> For a Bradley-Terry preference distribution $p^\* $ and the target policy $\pi^\* $, the reference-free reward assumes that preferences directly encode the target policy ($p^\* (y)\propto\pi^\* (y)^\beta$). Meanwhile, the DPO log-ratio reward assumes that preferences encode *Differential Information* ($p^\* (y)\propto q_{\pi^\* /\pi_\text{ref}}(y)^\beta$).
>
> On the Magpie-Pro and Ultra-Feedback datasets, we trained `Mistral-7B-v0.3` under both reward forms and evaluated them on 1,000 unseen prompts by measuring the average reward of sampled completions. Across a broad range of hyper-parameters ($\beta\in\\{0.2,0.1,0.05,0.02\\}$ and three training epochs), **DPO consistently produces responses with higher reward than $\pi_\text{ref}$, even outperforming the reference-free reward.**
>
> **Table A: Comparison of Expected Reward Using `Skywork-Reward-V2-Llama-3.1-8B`**
> |Dataset|Reward|Best Reward Improvement over $\pi_\text{ref}$|
> |-|-|-|
> |Ultra-Feedback|$r=\beta\log(\pi/\pi_\text{ref})$|**+8.20**|
> ||$r=\beta\log\pi$|+3.77|
> |Magpie-Pro|$r=\beta\log(\pi/\pi_\text{ref})$|**+7.41**|
> ||$r=\beta\log\pi$|+3.41|
>
> This shows that real-world preference data often encodes the *differential information* required to improve $\pi_\text{ref}$ into $\pi^\* $, rather than directly encoding the target policy $\pi^\* $ itself.
>
> ---
> ### **4. Summary**
> We were able to verify Theorem 4.1 on three real-world datasets. While the DID power-law and the global optimality of DPO cannot be confirmed on real data due to fundamental identifiability limitations, this does not necessarily invalidate our framework. Empirical results from the Magpie-Pro and Ultra-Feedback datasets indicate that real-world preference often encodes the differential information needed to improve upon $\pi_\text{ref}$.
>
> **We emphasize that our goal is to develop a principled Bayesian framework explaining when and why preference-optimization objectives such as DPO arise from the structure of preference data. Our experiments validate this perspective and show that the framework’s predictions align with empirical behavior.**
>
> ---
> ### **References**
> - [13] Xiao, et al. "A comprehensive survey of direct preference optimization: Datasets, theories, variants, and applications." arXiv:2410.15595.

---

> ### Author Response · Authors · 2025-11-25
>
> Dear Reviewer 9kxT,
>
> Thank you again for your thoughtful and constructive review. We're writing to kindly follow up on our rebuttal.
>
> In our response, we’ve addressed your questions regarding the definition of DID, the reference policy assumption, and the validation of Theorem 4.1 on real data. **In the common response, we have provided further empirical evidence that the DID power-law (Corollary 3.2.1) holds on real data.**
>
> If you have a moment to review our reply, we would greatly appreciate it. Please don’t hesitate to reach out if you have any remaining questions.
>
> Thank you again for your time and consideration.

---

### Author Response · Authors · 2025-11-25
**Empirical Evidence of DID Power-Law on Real Data**

We sincerely thank all the reviewers for their comments. Below, we address the common question on whether the DID power-law of Corollary 3.2.1 can be validated on real data.

Directly verifying this on real data is fundamentally difficult without access to the ground-truth distributions ($\pi_w, \pi_\ell$). Nevertheless, empirical evidence strongly supports that the DID power-law holds in practice. **Our experiments show that it accurately predicts the invariance of reward margins across $\beta$, and explains the log-likelihood shifts in DPO which cannot be accounted for from prior work.**

---
### **1. Evidence from Reward Margins**
**On two LMs (`Mistral7B-v0.3`, `Qwen3-4B`) and two real datasets (Magpie-Pro, Magpie-G27), we observe that the implicit reward margins of DPO-trained policies strongly align with Corollary 3.2.1.**

The DID power-law predicts that if $\pi_\text{ref}$ is fine-tuned on chosen responses, the average reward margin of the converged policy $\pi^\* $ should be invariant to the choice of $\beta>0$:

$\mathbb E_{y_w\sim\pi_\text{ref}}[\beta\log\frac{\pi^\* (y_w)}{\pi_\text{ref}(y_w)}]-\mathbb E_{y_\ell\sim\pi_\ell}[\beta\log\frac{\pi^\* (y_\ell)}{\pi_\text{ref}(y_\ell)}]=\mathbb D_\text{KL}[\pi_\text{ref}\parallel\pi_\ell]+\mathbb D_\text{KL}[\pi_\ell\parallel\pi_\text{ref}].$

*(Proof in Appendix F, Lines 1096–1104.)*

To test this, we revisited the training setup of Section 5.1. On Magpie-Pro and Magpie-G27 datasets, we fine-tuned `Mistral7b-v0.3` and `Qwen3-4B` on chosen responses, then DPO trained with $\beta\in\\{0.2, 0.1, 0.05\\}$ for 1 epoch (1530 steps). The final reward and log-margins were computed over the last 50 steps.

**Results:**
First, observe that the log-margins ($\mathbb E[\log\frac{\pi^\* (y_w)}{\pi_\text{ref}(y_w)}-\log\frac{\pi^\* (y_\ell)}{\pi_\text{ref} (y_\ell)}]$) vary drastically across $\beta$, scaling inversely to $\beta$ as expected. The differences are statistically significant and large:

|Dataset|Model|$\beta$|Log-Margin Avg.|Log-Margin Stddev.|
|-|-|-|-|-|
|Magpie-Pro|`Mistral7b-v0.3`|0.2|34.0834|3.9517|
|||0.1|55.1857|6.2964|
|||0.05|97.0979|12.1632|
||`Qwen3-4B`|0.2|7.6429|1.1854|
|||0.1|14.2212|2.2680|
|||0.05|25.4419|4.8218|
|Magpie-G27|`Mistral7b-v0.3`|0.2|42.7456|7.2483|
|||0.1|66.8927|13.3918|
|||0.05|114.5042|22.6283|
||`Qwen3-4B`|0.2|11.8139|1.7770|
|||0.1|20.2807|2.7566|
|||0.05|30.6132|4.0508|

However, the reward margins remain in a stable range, consistent with Corollary 3.2.1. While not completely identical due to finite data and noisy optimization, **the reward margins are remarkably similar compared to the variance seen from the previous log-margins:**
|Dataset|Model|$\beta$|Reward-Margin Avg.|Reward-Margin Stddev.|
|-|-|-|-|-|
|Magpie-Pro|`Mistral7b-v0.3`|0.2|6.8167|0.7903|
|||0.1|5.5186|0.6296|
|||0.05|4.8549|0.6082|
||`Qwen3-4B`|0.2|1.5286|0.2371|
|||0.1|1.4221|0.2268|
|||0.05|1.2721|0.2411|
|Magpie-G27|`Mistral7b-v0.3`|0.2|8.5491|1.4497|
|||0.1|6.6893|1.3392|
|||0.05|5.7252|1.1314|
||`Qwen3-4B`|0.2|2.3628|0.3554|
|||0.1|2.0281|0.2757|
|||0.05|1.5307|0.2025|

**This invariance of the reward margin across varying $\beta$ provides strong empirical support that the DID power-law holds in real-data settings.**

---
### **2. Evidence from Likelihood Shifts**
**The specific shift in log-likelihood of real-world DPO training offers further validation, distinguishing our framework from prior work.**

As noted from our comments, we observe likelihood shifts consistent to Theorem 4.1 on real data (Magpie-Pro, Magpie-G27). When $\pi_\text{ref}$ is fine-tuned on rejected responses and DPO trained with $\beta=1$, we consistently observe an **increase** in $\log\pi(y_w)$. This is a direct consequence of the DID power-law: under these conditions, Corollary 3.2.1 predicts that DPO with $\beta=1$ should converge to $\pi_w$.

To our knowledge, **prior work cannot explain this behavior without the DID power-law.** Existing analyses [1, 2, 3, 4] focus on DPO's tendency to *reduce* both $\log \pi(y_w)$ and $\log \pi(y_\ell)$, but do not account for conditions where $\log \pi(y_w)$ reliably *increases*. However, our experiments demonstrate this increase even at relatively high learning rates (1e-6), beyond the small-step assumption required by prior gradient-based explanations.

**In contrast, our framework accurately predicts *both* the increase and decrease of log-likelihoods depending on the reference policy, providing stronger explanatory power than prior work.**

---
### **References**
- [1] Pal, et al. "Smaug: Fixing failure modes of preference optimisation with dpo-positive." arXiv:2402.13228.
- [2] Feng, et al. "Towards analyzing and understanding the limitations of dpo: A theoretical perspective." arXiv:2404.04626.
- [3] Razin, et al. "Unintentional unalignment: Likelihood displacement in direct preference optimization." ICLR 2025.
- [4] Mao, et al. "As simple as fine-tuning: Llm alignment via bidirectional negative feedback loss." ICLR 2025.

---

### Author Response · Authors · 2025-12-03
**Final Global Response [1/2]**

## **Paper Summary**
We analyze the theoretical properties of Direct Preference Optimization (DPO) from a Bayesian perspective, using the concept of Differential Information Distribution (DID). We present the first unified explanation for the log-ratio reward of DPO, its training dynamics (likelihood change and policy exploration), and the performance trade-off between open-ended tasks and factual accuracy.

## **Acknowledged Strengths**
We appreciate the reviewers’ recognition of the contributions of our work:
- **A novel Bayesian framework for analyzing policy updates in preference optimization** (Reviewers 9kxT, KUMF, TcgT).
- **New insights on how the statistical structure of preference data shapes the training dynamics of DPO** (Reviewers 9kxT, TcgT).
- **DID entropy as a new, principled metric for understanding learned capabilities in preference optimization** (Reviewers KUMF, TcgT).

## **Discussion Overview**
While all the reviewers could not respond to our rebuttal due to the OpenReview bug, we were able to successfully address the reviewers' concerns with new real-world experiments and further clarifications on our theoretical findings.

**1. Validation on Real-World Data** (Reviewers 9kxT W3/Q3, KUMF W1, TcgT W6/Q1)
* Q: Requested real-world validation of our theoretical predictions involving **likelihood shifts and DID power-law**.
* A1: Conducted new experiments using Qwen3-4B on three datasets (Magpie-Pro, Magpie-G27, Ultra-Feedback), and confirmed that the **likelihood shifts** of DPO ($\beta=1.0$) match our predictions (Lines 371-373, Figure 3).
* A2: Conducted new experiments on two datasets  (Mapgie-Pro, Magpie-G27) and two LLMs (Mistral7B-v0.3, Qwen3-4B), and verified that the log- and reward-margin trends of DPO strongly support the **DID power-law** (Lines 273-291, Figures 10-11).

**2. Justification of Assumptions** (Reviewers 9kxT W1-2/Q1-2, KUMF W2, TcgT W1/Q1)
* Q: Asked justification for the **Conditional Independence** assumption in DID definition, and the **reference policy sampling assumption** from our analysis.
* A1: Clarified that **Conditional Independence** is a conceptually justified assumption that models sample-inherent information, and has also been widely adopted in related fields (e.g., Controlled Text Generation) (Lines 1028-1032).
* A2: Highlighted that the general Bayesian framework in Section 3 holds for any **reference policy** with full-support, while the reference policy assumption in Section 4 is a standard setup in the RLHF literature (Lines 934-947).

**3. DPO Optimality & Reward Design** (Reviewers 9kxT Q3, TcgT W4/Q3)
* Q: Asked **whether DPO’s optimality extends to real data** that deviate from theoretical assumptions and requested a **comparison with alternative reward forms**.
* A1: Clarified that **we do *not* claim DPO's optimality on all real data**; rather, its optimality holds for learning Differential Information, a distinct Bayesian property that may explain the wide adoption of DPO across domains (Lines 950-962).
* A2: Conducted new experiments on real datasets (Magpie-Pro and Ultra-Feedback) and found that **the DPO log-ratio reward consistently outperforms the reference-free reward (SimPO/CPO)**, supporting that real-world preferences are often better modeled as encoding the Differential Information needed to improve the reference policy (Appendix E).

**4. Robustness of Experimental Results** (Reviewers KUMF Q2, TcgT W3/Q2,4-5)
* Q: Requested validation on **alternative methods (SimPO)** and confirmation of **the stability of our DID entropy estimates**.
* A1: Conducted new experiments on SimPO that confirms our DID entropy analysis: **SimPO mirrors DPO in exhibiting likelihood displacement, similar performance trade-offs, and a tendency to learn high-entropy DID** (Appendix I).
* A2: Provided **95% confidence intervals for all DID entropy estimates**, showing that the reported differences between methods are statistically significant and robust (Table 5).

## **Conclusion**
Our work offers a principled Bayesian framework for understanding preference optimization, through the concept of Differential Information Distribution. New experiments and clarifications fully address the reviewers’ concerns and strongly support the correctness, robustness, and practical relevance of our theoretical contributions. ***As such, our framework delivers concrete, practical predictions while uncovering new theoretical insights into the behavior and design of DPO.***

---

> ### Author Response · Authors · 2025-12-03
> **Final Global Response [2/2]**
>
> We sincerely thank all reviewers for their constructive feedback. Below, we detail how our revision addresses the weaknesses and questions raised by each reviewer.
>
> *Note:* All changes from the revision are highlighted in $\color{blue}\text{blue}$.
>
> ---
> ## **Reviewer 9kxT**
> * **[W1-2,Q1-2] Justification of Assumptions**
> 	* **Line 113**: Clarified that the **IID assumption** is a standard setup adopted in prior analyses.
> 	* **Lines 934-947:** Clarified the scope of the **reference policy assumptions**, and included a discussion on how our framework can extend to other setups.
> 	* **Lines 1028-1032**: Clarified the conceptual basis for the **Conditional Independence** assumption and its established use in prior literature (e.g., Controlled Text Generation).
> * **[W3,Q3] Validation on Real-World Data**
> 	* **Lines 273-291, Figures 10-11:** Included new experiments on Mistral7B-v0.3 and Qwen3-4B using Magpie-Pro and Magpie-G27 datasets, which shows that the log/reward-margin of DPO directly supports the **DID power-law.**
> 	* **Lines 370-373, Figure 3:** Included new experiments on three datasets (Magpie-Pro, Magpie-G27, Ultra-Feedback), which demonstrate that the **log-likelihood shifts** of DPO follow the predictions of Theorem 4.1.
> * **[Q3] DPO Optimality**
> 	* **Lines 950-962:** Clarified that **we do *not* claim DPO's universal optimality on all real data**; rather, its optimality in learning Differential Information may help explain its robust performance in various domains.
> 	* **Appendix E:** Added new experiments on Magpie-Pro and Magpie-G27 datasets where **DPO's log-ratio reward consistently outperforms the reference-free reward (SimPO/CPO),** supporting that real-world preferences are better viewed as encoding Differential Information.
>
> ## **Reviewer KUMF**
> * **[W1] Validation on Real-World Data**
> 	* **Lines 273-291, Figures 10-11:** Included new real-data experiments that verify the **DID power-law** of DPO-converged policies using two datasets (Mapgie-Pro, Magpie-G27) and two LLMs (Mistral7B-v0.3, Qwen3-4B).
> 	* **Lines 370-373, Figure 3:** Included new real-data experiments that confirm the **likelihood shifts** of DPO on three datasets (Magpie-Pro, Magpie-G27, Ultra-Feedback).
> 	* **Line 377:** Highlighted that our predictions of **DPO policy exploration** (Theorem 4.2) strongly align with recent empirical studies (Wu, et al., 2024).
> * **[W2] Justification of Assumptions**
> 	* **Lines 934-947:** Clarified that the general Bayesian framework in Section 3 holds for any **reference policy** with full-support, while the reference policy assumption in Section 4 is a standard setup in the RLHF literature.
> * **[W3] Presentation of Technical Details**
> 	* **Lines 864-912:** Included a **Table of Contents for the Appendix** to help readers locate detailed proofs and additional discussions.
> * **[Q1] SimPO's Behavior in Figure 1**
> 	* **Lines 1984-1988:** Included an explanation of **SimPO's non-convergence to the target policy** based on Theorem 3.2.
> * **[Q2] Robustness of Experimental Results**
> 	* **Appendix I:** Extended our **DID entropy analysis to SimPO** which shows that its trend also mirrors DPO in exhibiting LLD and learning high-entropy DID.
>
> ## **Reviewer TcgT**
> * **[W1,Q1] Justification of Assumptions**
> 	* **Lines 1028-1032:** Included further justification for the **Conditional Independence** assumption and its wide adoption in prior work (e.g., Controlled Text Generation).
> * **[W3,Q2,4-5] Robustness of Experimental Results**
> 	* **Table 5:** Provided **95% confidence intervals for all DID entropy estimates** which confirms statistical significance.
> 	* **Lines 1214-1219:** Clarified why **alternative entropy estimators** (e.g., token-normalized DID entropy) fail to capture the uncertainty in the total Differential Information, and how they are difficult to evaluate efficiently in practice.
> * **[W4,Q3] DPO Optimality & Reward Design**
> 	* **Lines 950-962:** Clarified that **we do *not* claim DPO's optimality on all real data**; rather, Theorem 3.2 serves to establish DPO's optimality in learning *Differential Information*.
> 	* **Appendix E:** Added new experiments on Magpie-Pro and Magpie-G27 datasets showing that **the DPO log-ratio reward consistently outperforms the SimPO/CPO reference-free reward**.
> * **[W6,Q1] Validation on Real-World Data**
> 	* **Lines 273-291, Figures 10-11:** Included new experiments that verify the **DID power-law** on two datasets (Mapgie-Pro, Magpie-G27) and two LLMs (Mistral7B-v0.3, Qwen3-4B).
> 	* **Lines 370-373, Figure 3:** Included new experiments that confirm the **likelihood shifts** of DPO on three datasets (Magpie-Pro, Magpie-G27, Ultra-Feedback).
> 	* **Line 377:** Highlighted that our formal characterization of **DPO policy exploration** aligns with empirical trends (Wu, et al., 2024).
> ---
> We hope these revisions clarify the contributions and implications of our work. We once again thank the reviewers for their thoughtful feedback, which helped strengthen the paper.

---

### Meta-Review · Area_Chair_rNbz · 2026-01-08

**Summary:**

The reviewers share several common concerns about this submission:

1. The framework relies heavily on key assumptions and definitions, especially Definition 2.2 and the power-law DID assumption. The analysis remains conceptual, and real-world preference datasets are rarely power-law structured or independently sampled as assumed. Justification and empirical support in real preference data should be provided.

2. Theorem 4.1 assumes that either preferred or dispreferred responses are sampled from the reference model, which is often not true in practice and restricts the theorem's relevance.

3. The empirical support for the claims remains weak. Experiments on real LLMs are limited and do not directly test the key assumptions or predictions (power-law DID, DPO optimality, log-likelihood change).

4. The controlled Energy-Based Model experiments and empirical tests are built upon strong assumptions and synthetic settings where DIDs align perfectly. Results using real-world LLMs and datasets are limited.

5. There is the assumption that $\pi\_w = \pi\_{ref}$ in Section 4, along with the assumption that $\pi\_{ref}$ is the unoptimized policy. However, the initial policy is not necessarily fine-tuned on $y_w$, and there is research focusing on applying different $\pi\_{ref}$ strategies, which limits the applicability of the theory.

6. There are too many theorems, and many key proofs are in the appendix, which increases the difficulty of reading and hampers the flow.

7. DID existence is defined by construction, not derived, which weakens claims of theoretical generality. DID entropy estimation uses a small sample with potentially large variance. No confidence intervals or significance testing are reported.

8. Theorem 3.2's unique justification of the log-ratio reward is contingent on strong assumptions and may not hold under alternative data generation processes.

9. Experiments lack ablations and multi-seed averages.

**Reviewer Concerns:**

I appreciate that the authors have addressed many of the concerns with extensive experiments and clearer explanations. These clarifications have also been incorporated into the revised submission. However, several important reviewer concerns remain only partially addressed. As Reviewer 9kxT and TcgT point out, the conditional independence assumption is strong and cannot be fully justified or empirically tested. The provided examples are helpful, but do not rule out realistic scenarios where conditional independence fails, so the assumption of conditional independence appears mainly motivated by theoretical considerations. In addition, Reviewer 9kxT and TcgT also pointed out that the DID power-law is not directly empirically testable, which was also mentioned in the authors' rebuttal. Finally, as Reviewer KUMF mentions, the paper contains many theorems, which makes the paper theory-dense and impacts the overall readability and flow.

**Reviewer Scores:**

Reviewer KUMF has already given a score of 6 and is thus positive about the paper. Reviewers 9kxT and TcgT gave scores of 2 and 4, respectively, and they have concerns about several key assumptions in the paper, because these assumptions do not appear to be directly empirically testable. The authors' rebuttal clarifies some points but does not fully resolve these concerns. Then, Reviewers 9kxT and TcgT may still be inclined to remain negative.

---

### Decision · Program_Chairs · 2026-01-26

Reject